# The roles of resuspension, diffusion and biogeochemical processes on oxygen dynamics offshore of the Rhône River, France: a numerical modeling study

Julia M. Moriarty[1], Courtney K. Harris[1], Katja Fennel[2], Marjorie A. M. Friedrichs[1], Kehui Xu[3,4], Christophe Rabouille[5]

[1]Virginia Institute of Marine Science, College of William & Mary, Gloucester Point, Virginia, 23062 USA
[2]Department of Oceanography, Dalhousie University, Halifax, Nova Scotia B3P 2A3, Canada
[3]Department of Oceanography and Coastal Sciences, Louisiana State University, Baton Rouge, Louisiana 70803, USA
[4]Coastal Studies Institute, Louisiana State University, Baton Rouge, Louisiana 70803, USA
[5]Laboratoire des Sciences du Climat et de l'Environnement, UMR CEA-CNRS-UVSQ and IPSL, Gif sur Yvette 91198, France

*Correspondence to*: Julia M. Moriarty (moriarty@vims.edu)

**Abstract.** Observations indicate that resuspension and associated fluxes of organic material and porewater between the seabed and overlying water can alter biogeochemical dynamics in some environments, but measuring the role of sediment processes on oxygen and nutrient dynamics is challenging. A modeling approach offers a means of quantifying these fluxes for a range of conditions, but models have typically relied on simplifying assumptions regarding seabed-water-column interactions. Thus, to evaluate the role of resuspension on biogeochemical dynamics, we developed a coupled hydrodynamic, sediment transport, and biogeochemical model (HydroBioSed) within the Regional Ocean Modeling System (ROMS). This coupled model accounts for processes including the storage of Particulate Organic Matter (POM) and dissolved nutrients within the seabed; fluxes of this material between the seabed and the water column via erosion, deposition, and diffusion at the sediment-water interface; and biogeochemical reactions within the seabed. A one-dimensional version of HydroBioSed was then implemented for the Rhône subaqueous delta, France. To isolate the role of resuspension on biogeochemical dynamics, this model implementation was run for a two-month period that included three resuspension events; also, the supply of organic matter, oxygen and nutrients to the model was held constant in time. Consistent with time-series observations from the Rhône Delta, model results showed that erosion increased the diffusive flux of oxygen into the seabed by increasing the vertical gradient of oxygen at the seabed-water interface. This enhanced supply of oxygen to the seabed, as well as resuspension-induced increases in ammonium availability in surficial sediments, allowed seabed oxygen consumption to increase via nitrification. This increase in nitrification compensated for the decrease in seabed oxygen consumption due to aerobic remineralization that occurred as organic matter was entrained into the water column. Additionally, entrainment of POM into the water column during resuspension events, and the associated increase in remineralization there, also increased oxygen consumption in the region of the water column below the pycnocline. During these resuspension events, modeled rates of oxygen consumption increased by up to factors of ~2 and ~8 in the seabed and

below the pycnocline, respectively. When averaged over two months, the intermittent cycles of erosion and deposition led to a ~16 % increase of oxygen consumption in the seabed, as well as a larger increase of ~140 % below the pycnocline. These results imply that observations collected during quiescent periods, and biogeochemical models that neglect resuspension or use typical parameterizations for resuspension, may underestimate net oxygen consumption at sites like the Rhône delta.

Local resuspension likely has the most pronounced effect on oxygen dynamics at study sites with a high oxygen concentration in bottom waters, only a thin seabed oxic layer, and abundant labile organic matter.

## 1 Introduction

Understanding and quantifying the role that physical processes play on coastal water quality remains a scientific and management concern. Management solutions to hypoxia, the occurrence of low oxygen concentrations, as well as other

water quality issues, have focused on reducing riverine delivery of nutrients and sediments (Bricker et al., 2007). Yet temporal lags between these reductions and water quality improvements (Kemp et al., 2009), and increased cycling of nutrients within coastal systems (e.g. Testa and Kemp, 2012), indicate that temporary storage of nutrients in the seabed and subsequent release to the water column via diffusion and/or resuspension can affect water quality in some coastal environments. Neglecting these processes impairs managers' ability to develop and evaluate strategies for improving coastal

water quality (e.g. Artioli et al., 2008).

Resuspension-induced fluxes of sediment, Particulate Organic Matter (POM), and dissolved chemical species between the seabed and water-column can significantly affect biogeochemistry in coastal waters, including oxygen dynamics (Glud, 2008). Entrainment of seabed organic matter and reduced chemical species into the water-column can increase

remineralization and oxidation rates, thereby decreasing oxygen concentrations in bottom-waters (BW) in some environments. For example, Abril et al. (1999) observed that oxygen concentrations were inversely correlated with tidal fluctuations of suspended particulate matter concentrations in the Gironde Estuary, France. Recently, Toussaint et al. (2014) collected high-resolution time-series of microelectrode oxygen profiles on the Rhône River subaqueous delta that showed resuspension may also increase oxygen consumption in the seabed. This experiment revealed increases in diffusive fluxes of

oxygen from the water-column to the seabed during erosional events. Other observational studies have estimated resuspension-induced increases in oxygen consumption within the seabed and bottom-waters using measurements of turbulent oxygen fluxes (Berg and Huettel, 2008) and erodibility experiments (e.g., Sloth et al., 1996). Yet, it remains difficult to distinguish and quantify the relative influences of different biogeochemical (e.g. remineralization, oxidation) and physical (e.g. diffusion, resuspension) processes on oxygen dynamics in both the seabed and bottom-waters.

Hydrodynamic-biogeochemical models often complement observational studies of water quality (e.g. Moll and Radach, 2003; Aikman et al., 2014), but these simulations usually neglect or simplify seabed-water-column fluxes. Water quality

models often assume that organic matter and nutrients reaching the seabed are permanently buried, instantaneously remineralized, resuspended without remineralization, or a combination thereof (e.g. Cerco et al., 2013; Fennel et al., 2013; Druon et al., 2010; Bruce et al., 2014; Liu et al., 2015). Yet, numerical experiments showed that switching among relatively simple parameterization methods for seabed-water-column fluxes can alter the estimated area of low-oxygen regions by

about -50 % to +100 % in the Gulf of Mexico (Fennel et al., 2013). This sensitivity of modeled oxygen concentrations to the choice of parameterization, as well as the observations of temporally variable oxygen fluxes discussed above, motivate development of a process-based model for seabed-water-column fluxes.

We therefore developed a modeling approach that accounts for physical and biogeochemical processes at the seabed-water

interface, including resuspension of POM and porewater, and implemented it for the dynamic Rhône Delta. Previously, one-dimensional box models with a few vertical levels have been used to study the role of organic matter resuspension on oxygen (Wainright and Hopkinson, 1997) and contaminant levels (Chang and Sanford, 2005). Additionally, three-dimensional circulation models have been coupled to biogeochemical models with a single seabed layer and implemented to investigate the role of POM resuspension on Baltic Sea carbon budgets (Almroth-Rosell et al., 2011) and Black Sea biogeochemistry

(Capet et al., 2016). To the best of our knowledge, however, no previously existing models have sufficient vertical resolution to resolve changes in the vertical biogeochemical profiles that drive diffusive seabed-water-column fluxes, or the ability to account for the entrainment of reduced chemical species into the water column.

This paper presents a model called *HydroBioSed* that can reproduce the mm-scale changes in seabed profiles of oxygen,

nitrogen and carbon, as well as the resuspension-induced changes in seabed-water-column fluxes observed on the Rhône River subaqueous delta, by coupling hydrodynamic, biogeochemical and sediment transport modules. This process-based numerical model was implemented for the Rhône River subaqueous delta and used to evaluate how episodic erosion and deposition affect millimeter-scale seabed biogeochemistry and overall oxygen consumption in a dynamic coastal environment. Specific research questions for this paper include: (1) How do erosion and deposition affect the timing and

magnitude of seabed and bottom-water oxygen consumption? (2) What are the relative roles of local resuspension, organic matter remineralization, and oxidation of reduced chemical species in controlling oxygen consumption in the seabed and bottom waters? (3) How sensitive is oxygen consumption to resuspension frequency and magnitude, sedimentation rate, organic matter lability and availability, rate of diffusion within the seabed, and seabed nitrification rate? (4) What characteristics of the study site lead to the dependence of oxygen dynamics on local resuspension?

**2 Methods**

This section describes the Rhône Delta (Sec. 2.1), and HydroBioSed (Sect. 2.2), before explaining how the model was implemented to address the research questions (Sect. 2.3). Tables 1 and 2 list related symbols and vocabulary.

**2.1 Study site**

Located in the Gulf of Lions at the northwest end of the Mediterranean Sea, the Rhône River subaqueous delta in France is an excellent study site for these research questions in part because of the available observations (Fig. 1). Our study is co-located with the site from Toussaint et al. (2014) at the "Mesurho" station (Pairaud et al., 2016) and is only a few km away

from Site A in Pastor et al. (2011a); both locations are at ~25 m water depth and are characterized by similar biogeochemical characteristics (e.g. Rassmann et al., 2016), and so data from both sites were used for model input, validation and evaluation. Importantly, data from Toussaint et al. (2014) included a time-series of oxygen profiles with sub-millimeter scale resolution within the seabed and bottom centimeter of the water column. By resolving changes that occurred during resuspension events, Toussaint et al. (2014) showed that diffusion of oxygen into the seabed increased during resuspension events.

This site experiences frequent seabed disturbance due to centimeters of erosion superimposed on rapid fluvial deposition. Over timescales of decades, due to its proximity to the Rhône River (Fig. 1), accumulation rates at this site are ~10 cm $y^{-1}$ for sediment and 657 g $m^{-2}$ $y^{-1}$ of carbon (Radakovitch et al., 1999; Pastor et al., 2011a), although deposition varies in response to seasonal and episodic changes in river discharge and wave energy (Pont, 1997; Miralles et al., 2006; Ulses et al., 2008;

Cathalot et al., 2010). Deposition is punctuated by erosional events, and our study period, April-May 2012, included three instances when wave energy resuspended 1-2 cm of material from the seabed (Toussaint et al., 2014). At this site, erosion and deposition are the main sources of seabed disturbance; little bioturbation has been observed (Pastor et al., 2011b).

The delivery of organic matter to the shelf drives oxygen consumption directly via aerobic remineralization, and indirectly,

as reduced chemical species produced during remineralization are oxidized (Lansard et al., 2009). Organic material comprises about 2-12 % and <1-5 % of water-column and seabed particulate matter, respectively, and about four-fifths of it originates from a terrestrial source, with little marine influence at the study site (Bourgeois et al., 2011; Pastor et al., 2011a; Lorthiois et al., 2012; Cathalot et al., 2013). Yet, the material settling to the seabed at this site is relatively labile, and has been estimated to have remineralization rate constants of 11 - 33 $y^{-1}$ in the water column (Pinazo et al., 1996) and 0.31–11 $y^{-1}$

in the seabed (Pastor et al., 2011a). Despite the large input of organic matter to the Gulf of Lions, oxygen concentrations remain near saturation and hypoxia has not been reported, likely because the system is physically dynamic (Rabouille et al., 2008), suggesting that most organic matter is aerobically remineralized. In contrast, ~85% of seabed organic matter remineralization is anaerobic at our study site (Pastor et al., 2011a). This remineralization produces high ammonium concentrations that diffuse upwards and cause nitrification to account for an unusually large amount (over half) of the site's

seabed oxygen consumption, which is about 10-30 mmol $O_2$ $m^{-2}$ $d^{-1}$ in the prodelta where our site is located (Lansard et al., 2009; Pastor et al., 2011a, Toussaint et al., 2014). Yet, seabed fluxes of oxygen, carbon, and dissolved nutrients vary during resuspension events, complicating efforts to quantify the importance of different biogeochemical processes at this site (Lansard et al., 2009; Toussaint et al., 2014) and motivating this study.

**2.2 Model development**

The fully coupled HydroBioSed numerical model was developed within the Regional Ocean Modeling System (ROMS), a community-based and well-utilized ocean modeling framework (Haidvogel et al., 2000, 2008; Shchepetkin, 2003; Shchepetkin and McWilliams, 2009). In addition to its core hydrodynamic components, ROMS includes widely-used
modules for sediment transport (CSTMS; Community Sediment Transport Modeling System; Warner et al., 2008), and water-column biogeochemistry (e.g. Fennel et al., 2006, 2013). We built on those previous studies by coupling the sediment transport and water-column biogeochemistry components (Fig. 2a), enabling the model to account for storage of POM and nutrients in the seabed, and subsequent resuspension and redistribution of the organic matter and nutrients. As part of the coupling, we also incorporated aggregation of detritus, seabed-water-column diffusion, and a multi-layer seabed
biogeochemical model based on Soetaert et al. (1996a, 1996b). Below, we briefly describe the sediment transport and water-column biogeochemistry modules used, highlighting differences from standard ROMS implementations and the addition of the seabed biogeochemistry model.

**2.2.1 Sediment transport module**

Suspended sediment tracers in the ROMS-CSTMS module are transported by ocean currents, experience downward settling,
may be deposited and eroded from the multi-layer seabed model, and are subject to source and sink terms such as river discharge (Warner et al., 2008). As discussed in Warner et al. (2008), the rates of deposition, $D_{ised}$, and erosion, $E_{ised}$, for each sediment class $ised$, are calculated as follows (parameters are defined in Table 1):

$$D_{ised} = -\frac{\partial(w_{s,ised}C_{ised,z=1})}{\partial z_{w1}} \tag{1}$$

$$E_{ised} = M(1-\Phi)f_{ised}\left(\frac{\tau_{bed}-\tau_{crit,ised}}{\tau_{crit,ised}}\right) \qquad when\ \tau_{bed} \geq \tau_{crit,ised} \tag{2}$$

$$= 0 \qquad when\ \tau_{bed} < \tau_{crit,ised}$$

Resuspension from the seabed is parameterized such that erosion may only occur when the modeled bed stress, $\tau_{bed}$, exceeds the critical shear stress, $\tau_{crit,ised}$. Because erosion and deposition can co-occur, "erosional" and "depositional" time periods refer to times of net erosion, i.e. when $E_{ised}$ - $D_{ised} > 0$, and net deposition, i.e. when $F_{ised}$ - $D_{ised} < 0$, respectively. Previous CSTMS applications accounted only for inert particulates; however, here we adapted the model to link sediment transport
and biogeochemical processes. In HydroBioSed, POM from the water-column biogeochemical module provides an additional source of particulates to the seabed (Sect. 2.2.3), and POM can be deposited, eroded, and buried along with the sediment in its seabed layer. Note that POM comprises only ~3 % of the seabed by mass on the Rhône Delta and so it was considered negligible for calculating fluxes within the seabed layering scheme. Additionally, the seabed layering scheme of Warner et al. (2008) was modified so that the seabed has sufficient resolution (<1 mm) near the seabed-water interface where
vertical gradients in biogeochemical constituents such as dissolved oxygen can be high (see Appendix A). Finally, while

versions of CSTMS already accounted for diffusion of sediment within the seabed (Sherwood et al., in prep), HydroBioSed uses the same methods to accounts for the diffusion of porewater and POM.

### 2.2.2 Water-column biogeochemistry module

ROMS water-column biogeochemistry modules have typically included variables for multiple nutrient, plankton and detrital classes and accounted for processes such as growth, grazing and remineralization (e.g. Fennel et al., 2006). Here, the ROMS biogeochemical model from Fennel et al. (2013) was modified so that HydroBioSed converts some of the large detritus into faster-sinking aggregates in the water column. In Fennel et al. (2013), small detritus and phytoplankton in the water column may coagulate to form large detritus. HydroBioSed builds on the Fennel et al. (2013) framework by partitioning coagulated material into three types of particulate matter: (1) large detritus, (2) labile aggregates, and (3) refractory aggregates (Fig. 2b). Based on estimates that roughly half of the deposited particulate organic matter is refractory in the Gulf of Lions (Tesi et al., 2007; Pastor et al., 2011a), the model partitions coagulated material into 50 % refractory aggregates and 50 % labile material ($f_{lab} = 0.5$), which is divided evenly ($f_{ldet} = 0.5$) between labile aggregates (25 %) and large detritus (25 %):

$$Agg_{ref} = (1 - f_{lab}) \times (L_{det} + Agg_{lab} + Agg_{ref}) \tag{3}$$

$$Agg_{lab} = (f_{lab}) \times (1 - f_{ldet}) \times (L_{det} + Agg_{lab} + Agg_{ref}) \tag{4}$$

$$L_{det} = (f_{lab}) \times (f_{ldet}) \times (L_{det} + Agg_{lab} + Agg_{ref}) \tag{5}$$

Aggregates, similar to phytoplankton and detritus, are assigned settling velocities, remineralization rate constants and partitioning coefficients (Table 3; Fennel et al., 2006), and are transported within the water column by the hydrodynamic module. Upon sinking to the bed, aggregates, as well as phytoplankton and detritus, are added to the pool of seabed organic matter within the seabed module, as described in the next section.

### 2.2.3 Seabed biogeochemistry module

A seabed biogeochemistry module (Soetaert et al., 1996a, 1996b) was added to ROMS to account for changes in oxygen, dissolved nitrogen, and POM due to remineralization, oxidation of reduced chemical species, and diffusion across the seabed-water interface. This model has performed well in many environments including areas near river deltas (Wijsman et al., 2002; Pastor et al., 2011a), on the continental shelf and slope (Soetaert et al., 1998; Epping et al., 2002), and in the deep ocean (Middelburg et al., 1996). To incorporate the Soetaert et al. (1996a, 1996b) model into HydroBioSed, we used the code developed by Wilson et al. (2013), and adapted it for the ROMS framework and the Rhône Delta. Calculations use the first-order accurate Euler method.

This seabed biogeochemistry model specifically tracks degradable particulate organic carbon (POC), oxygen, nitrate, ammonium, and oxygen demand units (ODUs), defined as the moles of reduced chemical species that react with one mole of

$O_2$ when oxidized. Like Soetaert et al.'s early diagenetic model (1996a, 1996b), HydroBioSed uses ODUs to represent a combination of reduced chemical species that are produced during anoxic remineralization, including iron and manganese ions, sulfide, and methane. Modeled POC includes both labile and refractory (or semi-labile) classes. For a full model description, see Soetaert et al. (1996a, 1996b), but here we present the rate equations for oxic remineralization (Eq. 6),

denitrification (Eq. 7), anoxic remineralization (Eq. 8), nitrification (Eq. 9) and oxidation of ODUs (Eq. 10) to provide context for the Results and Discussion (see Table 1 for parameter definitions):

$$R_{aerobic} = POC \times R_{POC} \left( \frac{O_2}{O_2 + k_{O_2}} \frac{1}{L_{tot}} \right) \tag{6}$$

$$R_{DNF} = POC \times R_{POC} \left( \frac{l_{O_2}}{O_2 + l_{O_2}} \frac{NO_3}{NO_3 + k_{NO_3}} \frac{1}{L_{tot}} \right) \tag{7}$$

$$R_{anoxic} = POC \times R_{POC} \left( \frac{l_{O_2\_anoxic}}{O_2 + l_{O_2\_anoxic}} \frac{l_{NO_3\_anoxic}}{NO_3 + l_{NO_3\_anoxic}} \frac{1}{L_{tot}} \right) \tag{8}$$

$$R_{nit} = NH_4 \times R_{nit,max} \left( \frac{O_2}{O_2 + k_{O_2\_nit}} \right) \tag{9}$$

$$R_{oduox} = ODU \times R_{odu,max} \left( \frac{O_2}{O_2 + k_{O_2\_oduox}} \right) \tag{10}$$

$L_{tot}$, the non-dimensional sum of the limitation factors on remineralization processes, is:

$$L_{tot} = \left( \frac{O_2}{O_2 + k_{O_2}} \right) + \left( \frac{l_{O_2}}{O_2 + l_{O_2}} \frac{NO_3}{NO_3 + k_{NO_3}} \right) + \left( \frac{l_{O_2\_anoxic}}{O_2 + l_{O_2\_anoxic}} \frac{l_{NO_3\_anoxic}}{NO_3 + l_{NO_3\_anoxic}} \right) \tag{11}$$

Adaptations to the Soetaert et al. (1996a, 1996b) early diagenesis model that were made to merge it with the CSTMS and

Fennel modules include neglecting seabed consolidation and temperature-induced changes to biogeochemical rates. Specifically, HydroBioSed neglects changes in porosity with depth in the sediment bed because this study focused on the frequently resuspended surficial centimeter of the seabed and seabed-water-column interactions. Also, we neglected the effect of temperature on remineralization and diffusion because temperature was held constant for this implementation of HydroBioSed (see Sect. 2.3).

Merging the Soetaert et al. (1996a, 1996b) seabed biogeochemical model with the sediment transport and water-column biogeochemistry modules allows HydroBioSed to account for exchanges of biogeochemical tracers across the seabed-water interface due to deposition, erosion, and diffusion (Fig. 2b). Upon settling to the seabed, phytoplankton, detritus, and labile aggregates are incorporated into labile seabed organic matter in the surficial seabed layer. Refractory aggregates are added to

the pool of refractory seabed organic matter in that layer. Porewater in newly deposited sediments is assumed to initially have concentrations of dissolved nutrients and oxygen equal to those in the overlying water column. This material may be re-entrained into the water column when bed shear stress exceeds the critical shear stress of the seabed. Specifically, any POM or dissolved chemical species in the porewater within an eroded layer(s) of sediment is also entrained into the bottom water-column layer. The flux of sediment entrained into the water column is determined by the CSTMS module (see Sect. 2.2.1).

In addition to erosion and deposition, dissolved oxygen and nutrients may be transported across the seabed-water interface by diffusion as described in Appendix A.1.

During erosional periods, resuspended labile and refractory seabed organic matter is incorporated into the pools of labile or refractory aggregates suspended in the water column, respectively. Like other coagulated material in the water column, this material may be repartitioned based on Eqs. (3–5). Usually, the seabed organic matter is enriched in refractory material compared to the water column. Thus, this repartitioning reclassifies a fraction of the resuspended refractory organic matter, i.e. refractory aggregates, into the labile organic matter classes, i.e. large detritus, and labile aggregates. This modeling approach is supported by laboratory experiments by Stahlberg et al. (2006) indicating that organic matter remineralization rates increased during and in the days following resuspension events, and that changes in remineralization rates were not only due to changes in oxygen availability. Due to the limited availability of pertinent research, we also considered literature related to the effect of redox oscillations on organic matter remineralization (e.g. Gilbert et al., 2016; Sun et al., 2003; Caradec et al., 2004; Aller, 1994; Wakeham and Canuel, 2006; Arzayus and Canuel, 2004). Yet, because guidance from this literature was inconclusive, we chose the simple approach described above for the partitioning of organic matter that mimics the changes in remineralization described in Stahlberg et al. (2006). We also tested an alternative, 'no-repartitioning' approach that did not repartition resuspended organic matter, but this approach caused decreases in oxygen gradients across the seabed-water interface during depositional periods, inconsistent with observations from Toussaint et al. (2014) (Fig. 2c).

Overall, HydroBioSed represents POM in the seabed until it is resuspended, remineralized, or buried. Similarly, dissolved chemical species in the porewater may undergo biogeochemical transformations, diffuse into or out of the seabed, or be exchanged with the water column during periods of erosion and deposition. Thus, unlike Soetaert et al. (1996a, 1996b) and other classical seabed biogeochemistry models (e.g. Berner, 1980; Boudreau, 1997; Soetaert et al., 2000; DiToro, 2001), HydroBioSed can quantify the effect of resuspension on biogeochemical dynamics (Fig. 2).

**2.3 Model implementation and sensitivity tests**

To evaluate the coupled model and explore the role of local resuspension on oxygen dynamics, we implemented a one-dimensional version of HydroBioSed for the Rhône Delta. This section describes the standard model run and sensitivity tests, and summarizes our methods for model evaluation and analysis. See Table 3 for a list of model input and parameters.

*"Standard" Model Run:* A one-dimensional (vertical) version of HydroBioSed was implemented for a 24-m deep site on the Rhône subaqueous delta (Fig. 1) for April-May 2012. This time period coincided with Toussaint et al. (2014)'s observational study and included three resuspension events as well as quiescent periods characterized by low bed stress. To implement a quasi one-dimensional model within the ROMS framework, a 5-cell x 6-cell model grid with spatially uniform forcing and periodic open boundary conditions was used. Vertical stratification in the model was maintained by strongly nudging

temperature and salinity to climatological values; a pycnocline at 4 m above the seabed separated the colder saltier bottom waters from the warmer fresher upper water column. Wave- and current-induced bed stresses were estimated using the Sherwood, Signell and Warner (SSW) bottom boundary layer parameterization based on Madsen (1994) and described in Warner et al., (2008).

To isolate the effect of resuspension on seabed-water-column fluxes, water-column concentrations of oxygen, nitrogen, and ODU, as well as the supply of POM (excluding that from resuspension) were strongly nudged to temporally constant values. Hourly to daily oxygen observations from the bottom boundary layer (Toussaint et al., 2014) were used to constrain modeled concentrations in the water-column. These observations indicated that oxygen concentrations 1 m above the bed varied

between 216 - 269 mmol $O_2$ m$^{-3}$, but that resuspension events did not appear to impact near-bed $O_2$ fluctuations. A constant value of 253 mmol $O_2$ m$^{-3}$ was therefore used for water-column $O_2$ concentrations (Pastor et al., 2011a). Values for water-column nitrate, ammonium, and ODU concentrations were chosen based on Pastor et al. (2011a)'s Site A data because no observations were available from our study site (Fig. 1). Additionally, small detritus concentrations were strongly nudged to provide a constant supply of degradable POM to the water-column equivalent to 657 gC m$^{-2}$ y$^{-1}$, based on Pastor et al.

(2011a)'s estimate for organic sedimentation rate, $S_{organic}$. Nudging the small detritus concentrations did not affect those of the large detritus and aggregates that were resuspended from and deposited onto the seabed.

Model forcing and parameters were chosen based on a combination of observed values (wave height, bottom-water oxygen concentrations), climatology (inorganic sedimentation rate, salinity, temperature), and values used in previously

implemented models (fraction of labile material, nitrification rate, rates of diffusion within the seabed). See Table 3 for more details. A few parameters, i.e. critical shear stress for erosion and erosion rate parameter, were tuned to reproduce the 1-2 cm of observed erosion. For initialization, the model was run without resuspension until it reached steady state. As the biogeochemical profiles reached a state of quasi-equilibrium within days following perturbations, using alternative initialization techniques primarily affected estimates for the first resuspension event and did not have a large effect on our

results. The model used a 30 second time-step, the MPDATA advection scheme (Smolarkiewicz and Margolin, 1998), the Generic Length Scale turbulent closure (Umlauf and Burchard, 2009), and a Piecewise Parabolic Method (Colella and Woodward, 1984) with a weighted essentially non-oscillatory scheme (Liu et al., 1994) to estimate particle settling. It saved output in three-hour increments, and took ~6 hours to run on a single processor for a 2-month simulation.

*Sensitivity Tests:* In addition to the standard model run, seven sets of sensitivity tests examined the response of oxygen consumption to different parameters and processes (Table 4). These tests modified parameters related to resuspension and seabed processes, including the critical shear stress for erosion ($\tau_{crit}$), erosion rate parameter (*M*), inorganic and organic sedimentation rates ($S_{inorganic}$ and $S_{organic}$), lability of aggregated organic matter ($f_{lab}$) and the partitioning of organic matter (see Fig. 2b), rate of diffusion within the seabed ($D_i$), and nitrification rate in the seabed ($R_{nit,max}$). Additional tests modifying

the ODU oxidation rate and the parameterization scheme for seabed-water-column diffusion had a negligible effect on model results and so are not presented here.

Additionally, "no-resuspension" model runs were completed to evaluate the role of cycles of erosion and deposition on biogeochemical dynamics. Specifically, for each sensitivity test and the standard model run, a corresponding simulation was conducted that was identical to the original, except that erosion was prevented by increasing the critical shear stress to $\tau_{crit} =$ 10 Pa and decreasing the erosion rate parameter to $M= 0$ kg m$^{-2}$ s$^{-1}$. For conciseness, however, references to the "no-resuspension" model run refer to the no-resuspension version of the standard model, unless otherwise noted.

*Model Analysis:* We focused on seabed and bottom-water oxygen consumption and on fluxes of oxygen at the seabed-water interface. Bottom water was defined as the region of the water column within 4 m of the seabed, i.e. below the pycnocline, where suspended sediment concentrations were high during resuspension events. Concentrations and rates for analyses were saved in the model output. The fraction of oxygen consumption due to resuspension was calculated by dividing the difference between each sensitivity test and its no-resuspension model run by the value from the sensitivity test. Additionally, note that all POM estimates presented in this paper are for degradable organic matter. Although some studies add concentrations of inert POM to model estimates of degradable POM for comparison to observations, we plot only degradable POM for simplicity. Finally, depths of erosion into the seabed, which depend on both the duration of the event and bed stresses, were calculated by comparing the thickness of the seabed before versus during a time period of net erosion.

## 3 Results

This section evaluates the skill of the standard model run by comparing it to observations (Sect. 3.1), analyzes the effect of resuspension on oxygen dynamics (Sect. 3.2), and evaluates the results' sensitivity to model parameters (Sect. 3.3).

### 3.1 Model evaluation

Comparison of the standard version of HydroBioSed to Toussaint et al. (2014)'s time-series of oxygen profiles showed that model results were consistent with measured concentrations, and changed during resuspension events in a manner similar to the observations (Fig. 3). During quiescent conditions when bed shear stress was low, modeled and observed oxygen concentrations decreased with depth into the seabed, falling from about 250 mmol $O_2$ m$^{-3}$ in the bottom water column to 0 mmol $O_2$ m$^{-3}$ within 1-2 mm below the seabed surface. Similarly, both the modeled and observed oxygen penetration depths decreased to about <1 mm in the seabed during times of erosion, before returning to a quasi-steady state within hours of bed stresses returning to background values.

To quantify the changes in seabed oxygen profiles, the oxygen gradient near the seabed-water interface was calculated from both the observed and modeled profiles (Table 5). Specifically, the slope of the oxygen profile was averaged over the oxygen penetration depth (OPD; variables are defined in Table 1):

$$\frac{dO_2}{dz_{OPD}} = -\frac{O_{2,SWI} + O_{2,OPD}}{z_{SWI} + z_{OPD}} \tag{13}$$

Overall, $dO_2/dz_{OPD}$ increased during erosional periods (Fig. 3). During times when the seabed was not mobilized, $dO_2/dz_{OPD}$ maintained a baseline of ~100 mol $O_2$ m$^{-4}$, in both the modeled results and the observed values. In contrast, resuspension decreased the oxygen penetration depth, increasing $dO_2/dz_{OPD}$ to about 500 mol $O_2$ m$^{-4}$ (observed by Toussaint et al., 2014) and 900 mol $O_2$ m$^{-4}$ (modeled).

Differences in the modeled and observed oxygen profiles derive at least partially from differences in estimating seabed elevation (i.e. erosion and deposition). As a one-dimensional vertical model, HydroBioSed assumed uniform conditions in the horizontal, so that all resuspended material was re-deposited in the same location within a few days following an event. Yet, at the actual study site, it is likely that some material was carried out of the area and that deposition following the erosional periods was more gradual than estimated in the model (e.g. see the late April/early May event in Fig. 3c). Also, the model provided higher temporal resolution than possible with the sampling gear, and may capture peaks in $dO_2/dz_{OPD}$ that are missed by the sampling frequency (Fig. 3d). Yet, in spite of these differences, HydroBioSed reproduced the general behavior of oxygen profiles as observed on the Rhône subaqueous delta (Fig. 3e,f,g). In contrast to previous models that could not account for resuspension-induced temporal variations (Pastor et al., 2011a), both observed and modeled $dO_2/dz_{OPD}$ increased by factors of approximately 4-9 during erosional periods.

**3.2 Response of oxygen dynamics to resuspension**

Overall, the combined seabed-bottom-water oxygen consumption increased from ~40 mmol $O_2$ m$^{-2}$ d$^{-1}$ to over 200 mmol $O_2$ m$^{-2}$ d$^{-1}$ during resuspension events (Fig. 4b,c). Averaged over two months, resuspension roughly doubled the combined seabed-bottom-water oxygen consumption to >70 mmol $O_2$ m$^{-2}$ d$^{-1}$. Although the seabed and bottom waters contributed about equally to oxygen consumption during quiescent periods, the large increase in combined seabed-bottom-water oxygen consumption during resuspension events was primarily driven by remineralization of POM in bottom waters (Table 6). For both the seabed and bottom waters, resuspension added variability to oxygen dynamics, so that about one-half of the total oxygen consumption occurred within the 30 % of the two-month study period that included the resuspension events.

The cycles of erosion and deposition that affected biogeochemical cycles are illustrated by time-series of seabed profiles (Fig. 5). Before resuspension events, the porewater in surface sediments was typically equilibrated with the overlying water column, with oxygen penetrating ~1-2 mm into the seabed (Fig. 5a). As energetic waves increased bed stresses, however, particulate matter from the seabed was eroded into overlying water, with typical erosion depths of ~5-20 mm. This erosion of

the surficial seabed exposed low-oxygen, high-ammonium, high-ODU porewater to the sediment-water interface. This exposure changed profiles by, for example, sharpening the oxygen and ammonium gradients at the seabed-water interface and resuspending POM (Fig. 5b,h,k). As wave energy subsided and bed stresses decreased hours to a few days later, previously resuspended sediment and POM was re-deposited on the seabed (Fig. 5l). This re-deposited organic matter was particularly enriched in labile organic matter compared to the material that had remained on the seabed, due to repartitioning in the water column (Fig. 2b). As new seabed layers formed from re-deposited sediments, dissolved constituents from the overlying water were incorporated into the porewater of these new layer(s). This altered profiles by, for example, briefly increasing the thickness of the oxic layer up to ~5 mm during depositional periods.

The next two sections provide a more detailed and quantitative analysis of how these exchanges of porewater and particulate matter between the seabed and the overlying water increased oxygen consumption and affected related biogeochemical processes within the seabed (Sect. 3.2.1) and bottom waters (Sect. 3.2.2).

### 3.2.1 Seabed oxygen consumption

Resuspension directly altered the supply of oxygen to the seabed. In this environment, where oxygen penetration was limited to the top few millimeters of the seabed, resuspension events typically removed the entire seabed oxic layer; the oxygen that had been in the porewater was entrained into the water column. Similarly during deposition, incorporation of oxygen within the porewater of newly deposited sediment provided a source of oxygen to the seabed, accounting for up to a quarter of oxygen input to the seabed on a timescale of hours to days. Overall, this "pumping" of oxygen into and out of the seabed when sediments were deposited or eroded provided a small net source of oxygen to the seabed during a typical resuspension cycle; based on time-integrated fluxes of oxygen across the seabed-water interface for the two-month period (Fig. 6a), these exchanges accounted for 4 % of the net oxygen supply to the seabed.

The remaining supply of oxygen (96 %) was delivered to the seabed via diffusion across the seabed-water interface. Although these diffusive fluxes of oxygen were always directed into the seabed, erosion and deposition caused fluctuations in the rate of diffusion. During periods of resuspension, erosion of the oxic layer sharpened the oxygen gradient at the seabed-water interface, thus increasing diffusion of oxygen into the seabed by about 77 % (Fig. 6a). In contrast, during periods of deposition, incorporation of oxygen-rich porewater into newly deposited surficial seabed layers reduced the oxygen gradient at the seabed-water interface, decreasing diffusion of oxygen into the seabed by about 71 %. However, "erosional oxygen profiles" with thin oxygen penetration depths persisted longer and induced larger changes in the rate of diffusion, compared to "depositional oxygen profiles" with thick oxygen penetration depths. This imbalance occurred because the additional oxygen available in the seabed during periods of re-deposition (i.e., oxygen available due to the incorporation of oxic water into the porewater of newly-deposited sediments) was rapidly consumed by aerobic organic matter remineralization and nitrification, and so oxygen profiles returned to their quasi-steady state condition within hours to

~1 day after a resuspension event. In contrast, during erosional periods, steep oxygen gradients and increased rates of diffusion into the seabed persisted for ~2-5 days because of high nitrification rates (Fig. 6). Overall, averaged over two months, these resuspension-induced variations increased the rate of oxygen diffusion into the seabed by 12 %.

In addition to impacting the supply of oxygen to the seabed, resuspension altered the magnitude of various biogeochemical oxygen sinks within the seabed (Table 6, Fig. 6b). For example, erosion of organic matter, and labile organic matter in particular, decreased rates of oxic remineralization in the seabed from about 5 to <1 mmol $O_2$ $m^{-2}$ $d^{-1}$ (e.g. compare the mid-April quiescent period to the late April resuspension event). This decrease was offset by nitrification, which increased from ~10-15 to ~30 mmol $O_2$ $m^{-2}$ $d^{-1}$ during resuspension events. Nitrification rates increased because of the greater supply of

oxygen to the seabed from erosion-enhanced diffusion. Nitrification also increased due to the larger ammonium concentrations in surficial sediments that occurred as erosion exposed relatively ammonium-rich seabed layers and due to the erosion-induced increase in the rate of diffusion of ammonium from deeper regions of the seabed towards the seabed-water interface. Overall, these changes increased the fraction of oxygen consumed via nitrification from about 60-70 % during quiescent periods to ~85 % during erosional periods. At the same time, the fraction of oxygen consumed via aerobic

remineralization decreased from about 30-40 % during quiescent periods to 15 % during erosion. In contrast, following resuspension events, remineralization of redeposited organic matter, especially labile organic matter, briefly increased oxic remineralization rates. Also, low ammonium concentrations in newly deposited sediments limited nitrification during depositional periods. Together, these changes briefly altered the fraction of oxygen consumed via nitrification vs. remineralization to about 17 % and 83 %, respectively, during periods of re-deposition. Averaged over two months,

however, resuspension-induced changes in the availability of oxygen, organic matter, and nutrients had little effect on the fraction of oxygen consumption due to nitrification (74 %) and remineralization (26 %).

### 3.2.2 Bottom-water oxygen consumption

Resuspension primarily affected oxygen dynamics within the water column by entraining POM into the layer of water below the pycnocline, i.e. bottom waters, which increased remineralization rates there (Table 6). Turbulence entrained this material

as high as ~3-4 m above the seabed during resuspension events, with near-bed concentrations of POM reaching up to 5 x $10^4$ mmol C $m^{-3}$ in the model. Aerobic remineralization of resuspended material consumed up to 170 mmol $O_2$ $m^{-2}$ $d^{-1}$, although the average rate during erosional periods was 63 mmol $O_2$ $m^{-2}$ $d^{-1}$.

In addition to entraining POM into the water column, resuspension increased fluxes of reduced chemical species from the

seabed into bottom waters, further increasing oxygen consumption in the water column (Table 6). During quiescent periods, oxidation of ammonium (nitrification) resulted in a background level of oxygen consumption of ~23 mmol $O_2$ $m^{-2}$ $d^{-1}$ in bottom waters. During erosion, the steepening of gradients at the seabed-water interface increased the diffusive flux of ammonium from the seabed to bottom waters from near zero to up to about 25 mmol $m^{-2}$ $d^{-1}$ of $NH_4$. Direct entrainment of

ammonium into the water column provided an additional ~5-10 mmol m$^{-2}$ d$^{-1}$ of NH$_4$. The greater supply of NH$_4$ increased bottom-water nitrification rates to up to ~34 mmol O$_2$ m$^{-2}$ d$^{-1}$ during resuspension events, with an average of 26 mmol O$_2$ m$^{-2}$ d$^{-1}$ during erosional periods. Comparing this oxygen demand with the estimates of remineralization-related demand calculated above, nitrification accounted for ~30 % of oxygen consumption in bottom waters during erosional periods. The remaining ~70 % percent came from the remineralization of organic matter.

### 3.3 Sensitivity tests

Like the standard model run, results from every sensitivity test showed that resuspension increased bottom-water oxygen consumption during both individual resuspension events and when estimates were averaged over two months (Fig. 7d). All sensitivity tests except one showed that resuspension also increased seabed oxygen consumption (Fig. 7b). In all model runs, oxygen consumption in bottom waters was larger than that in the seabed for every sensitivity test by at least a factor of ~5 during resuspension events and ~2 when results were averaged over two months. However, altering various parameters affected the model estimates of oxygen consumption in both the seabed and bottom waters, as explored below. This analysis focuses on the two-month average of oxygen consumption rate and the maximum rate of oxygen consumption from erosional periods (Fig. 7a,c). For both of these quantities we also computed the fraction of oxygen consumption induced by resuspension (Fig. 7b,d).

### 3.3.1 Seabed oxygen consumption: Sensitivity tests

Over timescales ranging from hours to two months, seabed oxygen consumption was more sensitive to changes in the rate of diffusion within the seabed ($D_i$, Cases B1 and B2; Fig. 7a) than any other parameter considered in the sensitivity tests (Table 4). Halving and doubling the diffusion coefficients changed the seabed oxygen consumption by -28 % and 39 %, respectively, when integrated over the two-month model run, and by -22 % and 24 % during individual resuspension events. These changes occurred because faster diffusion rates within the seabed more quickly transported oxygen deeper into the seabed, reducing oxygen levels in surface sediments, and thereby increasing the diffusion of oxygen through the seabed-water interface. Additionally, faster diffusion rates within the seabed transported ammonium upwards, toward the seabed-water interface. Increasing $D_i$ thus increased the amount of oxygen and ammonium at the oxic-anoxic interface within the seabed, allowing for more seabed oxygen consumption via nitrification. In contrast, lower diffusion rates within the seabed lowered the supply of oxygen and ammonium to this region of the seabed, reducing seabed oxygen consumption.

Within the standard model run and most sensitivity tests, resuspension accounted for about 14 % of the cumulative seabed oxygen consumption when integrated over two months. The role of resuspension, however, was especially sensitive to the partitioning and delivery of organic matter because POM entrained into the water column was subject to repartitioning (see Sect. 2.2.3; Fig. 2b) and so resuspension increased the amount of labile material available to re-deposit on the seabed. This additional source of seabed labile organic matter increased seabed oxygen consumption directly, due to oxic

remineralization, and indirectly, as ammonium produced during this process was oxidized via nitrification. Overall, altering the partitioning of organic matter between labile and refractory classes changed the effect of resuspension on seabed oxygen consumption by up to 60 % over two months (Cases L1 and L2; Fig. 7b). Specifically, decreasing (increasing) the fraction of organic matter that is labile, $f_{lab}$, by 30 % decreased (increased) the resuspension-induced fraction of the seabed oxygen consumption to 5 % (22 %), compared to 14 % in the standard model run. Furthermore, the sensitivity test without repartitioning of POM in the water column was the only sensitivity test for which resuspension caused a marginal (negative) effect on seabed oxygen consumption when results were averaged over two months (Case C1; Fig. 2c, 7b). In this case, resuspension-induced increases in the supply of oxygen and seabed nitrification were about equal to the decrease in oxic remineralization that occurred when POM was entrained into the water column.

### 3.3.2 Bottom-water oxygen consumption: Sensitivity tests

Oxygen consumption in bottom waters averaged over two months was more sensitive to changes in the critical shear stress for erosion, $\tau_{crit}$, than other parameters (Fig. 7c; Cases T1 and T2). Halving and doubling the critical shear stress changed time-averaged bottom-water oxygen consumption by 50 % and -35 %, respectively. During individual resuspension events, the effect of halving and doubling this parameter was more moderate and resulted in 7 % and -20 % changes, respectively. These changes in oxygen consumption occurred because halving and doubling the critical stress for erosion changed the frequency of resuspension, i.e. the amount of time that $\tau_{bed} > \tau_{crit}$, from 36 % of the time in the standard model run to 53 % and 15 %, respectively. Thus, decreasing the critical shear stress prolonged resuspension events, which caused more seabed organic matter and porewater to be entrained into the water column, increasing oxygen consumption in bottom waters. In contrast, a larger critical shear stress shortened resuspension events, decreasing oxygen consumption there.

Within the standard model run and most sensitivity tests, resuspension accounted for about 57 % of bottom-water oxygen consumption when averaged over two months (Fig. 7d). Similar to the above analysis, the extent to which resuspension affected oxygen consumption was especially sensitive to the critical shear stress (Cases T1, T2). Over the two-month model run, halving (doubling) the critical shear stress changed the fraction of bottom-water oxygen consumption that occurred due to resuspension to 34 % (71 %).

### 4 Discussion

This discussion focuses on the importance of resuspension-induced changes in oxygen budgets in different environments (Sect. 4.1); compares our approach to other modeling techniques (Sect. 4.2); and suggests future research (Sect. 4.3).

**4.1 Resuspension-induced increases in oxygen consumption**

Resuspension-induced oxygen consumption that occurred during short time periods (hours to days) increased model estimates of oxygen consumption integrated over longer timescales of weeks to months for all model runs (Fig. 7, 8). In other words, erosion and deposition did not just add variability to the time-series of oxygen consumption; resuspension impacted the oxygen budget of the Rhône subaqueous delta. This section discusses the environmental conditions that caused this effect and the extent to which we expect resuspension to increase oxygen consumption in other coastal systems (Sect. 4.1.1); and the importance of these changes relative to seasonal variability (Sect. 4.1.2).

**4.1.1 Why does resuspension change oxygen consumption on the Rhône Delta?**

Several characteristics of the Rhône subaqueous delta favor the increased rates of oxygen consumption due to local resuspension. First, frequent resuspension, e.g. three events in two months (Fig, 3c), ensures that the entrainment of seabed organic matter into the water column and erosional seabed profiles occur often, increasing resuspension-induced oxygen consumption in both bottom waters and the seabed. Second, oxygen concentrations in bottom waters and near the seabed-water interface are relatively high, i.e. over 200 mmol $O_2$ $m^{-3}$ (Fig. 3e,f,g), ensuring that oxygen is available to be consumed. Third, the seabed at this site on the Rhône Delta experiences little biological mixing (Pastor et al., 2011a). This encourages the formation of a relatively thin oxic layer that can be completely resuspended, allowing erosional seabed profiles that increase seabed oxygen consumption to form frequently. Fourth, organic matter and/or reduced chemical species concentrations are high in surficial sediments relative to the water column (e.g. Pastor et al., 2011a,b; Cathalot et al., 2010). This ensures that erosion provides a significant supply of organic matter to the water column for remineralization, increasing oxygen consumption in bottom waters during resuspension. Also, the large amount of labile organic matter and reduced chemical species in the seabed facilitates resuspension-induced seabed oxygen consumption by quickly consuming oxygen via remineralization or oxidation during resuspension events. The speed of oxygen consumption is important for the maintenance of erosional seabed profiles and destruction of depositional profiles throughout the entire resuspension event. Fifth, remineralization rates in bottom waters are fast compared to the residence time of suspended particles in the water column, ensuring oxygen can be consumed in bottom waters before organic matter settles back to the seabed. The rates used in the model imply that as much as 170 mmol $O_2$ $m^{-2}$ $d^{-1}$ is consumed via organic matter remineralization during resuspension events, which often last for days on the Rhône Delta (Table 6, Fig. 4). Finally, resuspension can increase rates of organic matter remineralization during and following resuspension events due to changes in redox conditions and other processes, increasing oxygen consumption (e.g. Stahlberg et al., 2006). Such changes can increase aerobic remineralization rates, and were particularly important for enhancing time-averaged seabed oxygen consumption.

We expect that the effect of local resuspension on oxygen dynamics in other systems that share characteristics of the Rhône subaqueous delta would be similar to our results. For seabed oxygen dynamics, this implies that the importance of local

resuspension increases in energetic, oxic, coastal areas with high organic matter input, but relatively little bioturbation, including other river deltas (Aller, 1998; e.g. Amazon Delta, Brazil: Aller et al., 1996). For water-column oxygen dynamics, the above criteria suggest that local resuspension is most important in similar coastal areas with organic-rich, muddy seabeds, but relatively low background concentrations of organic matter in the water column. These characteristics may be found in regions with historically high nutrient loading and where organic matter has accumulated in the seabed (e.g. Gulf of Finland: Almroth et al., 2009). In sites that meet some, but not all of the above criteria, local resuspension may have a reduced effect on oxygen dynamics compared to the Rhône subaqueous delta.

### 4.1.2 How does resuspension-induced $O_2$ consumption compare to seasonal variability?

The model estimated that resuspension increased seabed and bottom-water oxygen consumption by about 16 % and 140 %, respectively, when integrated over April-May 2012 (Fig. 7); however, seasonal variations in environmental conditions such as temperature may change the importance of resuspension for oxygen dynamics. The two-month model run presented here assumed a constant bottom-water temperature of 15°C, but observed values vary from ~12–20 °C over the course of a year on the Rhône Delta (Millot, 1990; Fuchs and Pairaud, 2014; Rabouille, pers. comm.). A common method for estimating temperature-induced changes in biogeochemical processes is the "$Q_{10}$ rule" (van't Hoff, 1898), which predicts that oxygen consumption increases by a factor of ~2-3 for each temperature increase of 10°C in coastal areas (e.g. Thamdrup et al., 1998; Dedieu et al., 2007; Cardoso et al., 2014). Based on the 16±4 °C temperature range expected at this site over a year, this suggests that resuspension-induced changes in oxygen consumption are as important as the factor of 2 change estimated due to temperature-induced variability. Thus, although temperature effects have been widely studied, resuspension can cause similar variations in oxygen consumption.

Seasonal variations in resuspension frequency and magnitude may have a similarly large effect on oxygen consumption. During the winter when easterly storms are more frequent (Guillén et al., 2006; Palanques et al., 2006), resuspension-induced oxygen consumption could be more important than was estimated for the April-May period in this study. At the 32 m deep "Sète" site in the central coastal region of the Gulf of Lions, significant wave heights exceeding 2 m were observed an average of 3.5, 1 and 2 times per month in November-December 2003, January-February 2004, and March-April 2004, respectively (Ulses et al., 2008). Approximately doubling the resuspension frequency during the winter storm season could roughly double resuspension-induced oxygen consumption, counteracting reductions in wintertime oxygen consumption due to colder temperatures. Overall, accounting for the effect of erosional and depositional cycles on oxygen consumption may vary in importance throughout the year on the Rhône subaqueous delta, but it is likely more important during Fall compared to the Springtime period that was analyzed for this study.

Finally, oxygen dynamics may vary in response to seasonal or episodic variations in organic matter availability and lability. Following a flood in 2008, seabed oxygen consumption on the Rhône Delta decreased by one-third to one-half when riverine

inputs of relatively refractory organic matter lowered remineralization rates in surficial seabed sediments, reducing seabed oxygen consumption (Cathalot et al., 2010). This result is consistent with results from our L1 sensitivity test indicating that reducing the ratio of labile to refractory organic matter lowered seabed oxygen consumption (Fig. 7a). Thus, although variability in the amount and quality of organic matter delivered to the delta could be episodic, it may also substantially affect estimates of seabed oxygen consumption oxygen, similar to temperature and resuspension.

## 4.2 Modeling resuspension-induced changes in oxygen dynamics

HydroBioSed differs from other models by accounting for resuspension-induced changes in millimeter-scale biogeochemistry, a feature that was necessary to reproduce Toussaint et al. (2014)'s observed temporal variations in seabed oxygen consumption on the Rhône subaqueous delta. In contrast, other models neglect resuspension-induced changes in biogeochemical dynamics or assume that increases in water-column oxygen consumption due to remineralization of resuspended organic matter during erosion are at least partially offset by decreases in remineralization and associated oxygen consumption in the seabed (e.g. Druon et al., 2010; Capet et al., 2016). Results from these model parameterizations therefore conflict with our HydroBioSed results that show that both water-column and seabed oxygen consumption *increase* during resuspension events (Fig. 4, 6), consistent with observations for the Rhône subaqueous delta (Fig. 4, 6; Toussaint et al., 2014). This implies that the parameterizations from other models such as those cited above underestimate oxygen consumption during resuspension events when applied to environments with similar characteristics to the Rhône Delta, as described in Sect. 4.1.1. The remainder of this section explores which sediment processes were most critical for modeling the effect of resuspension on Rhône Delta oxygen dynamics.

First, resuspension increased the importance of bottom waters relative to the seabed for oxygen consumption. During quiescent conditions, bottom waters and the seabed each accounted for similar rates of oxygen consumption. However, when POM and porewater were entrained into the water column via resuspension, bottom-water oxygen consumption increased by a factor of 8, while seabed oxygen consumption only doubled. This disproportionate increase of oxygen consumption within bottom waters affirmed the importance of observing and modeling oxygen dynamics within bottom waters during resuspension events. Also, only accounting for quiescent time periods would underestimate the role of bottom waters, which accounted for 75 % of the total oxygen consumption over the two-month model run for the Rhône Delta site, but only accounted for about 50 % when resuspension was neglected.

Second, diffusion of oxygen across the sediment-water interface dominated the supply of oxygen to the seabed in the model, regardless of the timescale or time period considered. The other transport mechanism, the "pumping" of oxygen into and out of the seabed as layers of sediment were deposited or eroded, provided at most a third of the instantaneous flux to the seabed (during depositional time periods; Fig. 5). Also, pumping contributed much less to seabed oxygen supply over time, primarily because the entrainment of porewater from the seabed into the water column during erosional periods partially

offset the depositional flux of oxygen (Fig. 5). Over the two-month simulation, diffusion across the seabed-water interface accounted for 96 % of the seabed oxygen supply, whereas pumping via erosion and deposition accounted for only 4 % of seabed oxygen fluxes. Thus, for environments like the Rhône Delta, future observational and modeling efforts should include resuspension-induced changes to diffusive fluxes across the seabed water interface (Jørgensen and Revsbech, 1985).

Although resuspension can affect oxygen dynamics in coastal environments, the large spatial or temporal scale of some biogeochemistry models may make incorporating a full sediment model undesirable. For environments similar to the Rhône Delta, we suggest parameterizations for bottom-water and seabed oxygen consumption that focus on the role of resuspended organic matter and seabed-water-column diffusion. For example, various approaches have been used to parameterize the

effect of resuspension on particulate organic matter fluxes (e.g. Cerco et al., 2013; Druon et al., 2010). Approaches accounting for temporal lags between deposition and re-entrainment of organic matter into the water column seem especially promising for modeling oxygen dynamics in episodically energetic environments like the Rhône Delta (e.g. Almroth-Rosell, 2011; Capet et al., 2016). In addition, future parameterizations for seabed-water-column fluxes should focus on diffusion of oxygen across the seabed-water interface as well as the supply of organic matter and reduced chemical species (e.g. Findlay

and Watling, 1997; De Gaetano et al., 2008; Hetland and DiMarco, 2008; Murrell and Lehrter, 2011; Testa et al., 2013; Laurent et al., 2016). Methods combining parameterizations for seabed-water-column fluxes and seabed resuspension may be particularly helpful for environments similar to the Rhône Delta where erosion and deposition may affect these processes.

### 4.3 Implications of model development & future work

This study focused on oxygen dynamics while holding the supply of organic matter and sediment; water-column

concentrations of nutrients and oxygen; and temperature constant in time based on conditions observed on the Rhône subaqueous delta. Future work should therefore include analyzing the role of resuspension on oxygen dynamics for a variety of environmental conditions and investigating how temporal variability in environmental conditions affects the relative importance of resuspension for oxygen dynamics. Additionally, applying HydroBioSed for a three-dimensional system would further facilitate its application to additional scientific and water quality concerns. For example, transport of organic

matter from regions near the Mississippi and Atchafalaya river mouths, shallow autotrophic waters, and wetlands to "Dead Zones" has been speculated to encourage the depletion of oxygen in bottom waters there (Bianchi et al., 2010). However, the importance of organic matter transport within a single season of hypoxia, and on inter-annual timescales, is difficult to quantify with observations and has been debated on the northern shelf of the Gulf of Mexico (Rowe and Chapman, 2002; Boesch, 2003; Turner et al., 2008; Forrest et al., 2012; Eldridge and Morse, 2008) and other locations (Kemp et al., 2009 and

references therein). Modeling efforts that account for resuspension of organic matter, as well as oxygen and nutrients, can help quantify the extent to which organic matter supply, resuspension and transport affect biogeochemistry in these dynamic coastal environments (e.g. Almroth-Rosell et al., 2011; Capet et al., 2016).

Our analysis focused on oxygen, but resuspension also affected model estimates of nitrogen dynamics. For example, during quiescent periods, nitrification roughly balanced production of ammonium from remineralization of organic matter in the seabed, consistent with Pastor et al. (2011a). Yet, during erosional periods, the exposure of ammonium-rich porewater to oxygen increased seabed nitrification, enhancing fluxes of nitrate out of the seabed, consistent with observations from other

systems (e.g. Fanning et al., 1982; Sloth et al., 1996; Tengberg et al., 2003). Overall, resuspension roughly doubled nitrate fluxes out of the seabed during resuspension, which led to about a 10 % increase overall for the two-month model run.

HydroBioSed did not represent all processes that occur near the seabed-water-column interface. For example, future work could include accounting for turbulence-induced changes in diffusion, advective fluxes through the seabed, and variations in

seabed porosity; as well as improving the model's representation of organic matter. Within HydroBioSed, for example, the steepening of the oxygen gradient at the seabed-water interface occurred because of changes in oxygen concentrations within the seabed and bottom waters (Fig. 3). HydroBioSed did not account for the thinning of the viscous layer at the seabed-water interface in response to wave-induced turbulence, which would act to further increase the oxygen gradient during erosional time periods (Gundersen and Jorgensen, 1990; Chatelain and Guizien, 2010; Wang et al., 2013). This implies that our current

model estimates of oxygen diffusion into the seabed during resuspension events are conservative. Additionally, the model could be adapted for locations where waves and currents drive flows of water through non-cohesive seabeds, stimulating biogeochemical reactions (Huettel et al., 2014), or to account for vertical gradients in seabed porosity (Soetaert et al., 1996a, 1996b). Finally, the uncertainty about how to partition organic matter into classes for numerical modeling efforts and the effect of resuspension on remineralization rates, as noted in Sect. 2.2.3, both have a large effect on model estimates (Fig. 7,

Cases L1, L2, C1) and deserves attention from both the modeling and observational research communities.

Finally, this modeling effort incorporated time-dependent reactions into the ROMS sediment transport module and could be adapted for other research applications for which both resuspension and time-dependent tracers are important. For example, the model has been adapted to account for short-lived radioisotopes (Birchler, 2014) and could be adapted to include:

particle-reactive nutrients and contaminants (Wiberg and Harris, 2002; Chang and Sanford, 2005); other "particulates" such as harmful algal blooms (HAB) cysts (Beaulieu et al., 2005; Giannakourou et al., 2005; Butman et al., 2014; Kidwell, 2015) or fecal pellets (Gardner et al., 1985; Walsh et al., 1988); and temporal variability in organic matter lability, oxygen exposure time and carbon budgets (Aller, 1998; Hartnett et al., 1998; Burdige, 2007).

## 5 Summary and conclusions

A model called HydroBioSed was developed that couples hydrodynamics, sediment transport, and both water-column and seabed biogeochemistry. A one-dimensional (vertical) version of the model was then implemented for the Rhône River subaqueous delta. This work expanded on the commonly used ROMS framework by accounting for non-conservative

tracers, the resuspension of organic matter and entrainment of porewater into the water column, diffusion of dissolved tracers across the seabed-water interface, and feedbacks between resuspension and diffusion across the seabed-water interface. Including these processes created a new model capable of reproducing previously observed changes in seabed profiles that occurred during resuspension events on the Rhône River subaqueous delta.

Resuspension increased model estimates of oxygen consumption over the range of timescales considered (hours to two months). In the seabed, resuspension increased the exposure of anoxic, ammonium-rich sediment to oxic, ammonium-poor bottom waters, thus stimulating seabed oxygen consumption via nitrification during erosional periods. This oxygen consumption compensated for or exceeded the decrease in oxic remineralization rates that occurred as organic matter was

resuspended into the water column. Additionally, entrainment of seabed organic matter and reduced chemical species from the porewater into the bottom portion of the water column, i.e. below the pycnocline, increased oxygen consumption there. Overall, resuspension increased peak oxygen consumption rates more in bottom waters (factor of 8) than in the seabed (factor of 2). When averaged over a two-month period that included intermittent periods of erosion and deposition, accounting for resuspension increased oxygen consumption by ~16 % in the seabed and ~140 % in bottom waters. Overall,

the combined seabed and bottom-water oxygen consumption increased by a factor of ~5 during wave resuspension events and roughly doubled the two-month average.

These results imply that observations collected during quiescent periods, and models based on steady-state assumptions, may underestimate net oxygen consumption. This finding is consistent with results from laboratory erodibility experiments (e.g.

Sloth et al., 1996), observations using eddy correlation techniques (Berg and Huettel, 2008), and microelectrode profiles (Toussaint et al., 2014). While all of these studies showed increased oxygen consumption during resuspension events, they each had limitations; i.e., erodibility experiments are limited to low levels of erosion and timescales of hours, eddy-correlation methods can only be used for time periods without abrupt shifts in hydrodynamic and oxygen conditions (Lorrai et al., 2010), and microelectrodes can only be deployed in soft muddy seabeds. Thus, models like HydroBioSed that resolve

both biogeochemical processes and resuspension may help observational studies quantify oxygen dynamics over longer time periods, during storms, and in a variety of environments.

Certain characteristics of the Rhône subaqueous delta study site, including its oxic water column, shallow oxygen penetration into the seabed compared to the thickness of eroded layers, fast rates of oxygen consumption, and the high

concentrations of labile seabed organic matter, enhance the effect of resuspension on oxygen dynamics. Together, these characteristics ensure that: oxygen consumption in bottom waters is limited by the supply of organic matter and reduced chemical species, as opposed to oxygen availability; resuspended material is rich in organic matter and reduced chemical species that increases oxygen demand in the water column; oxygen consumption in the seabed is dependent on the supply of oxygen, as opposed to the rate of consumption; oxygen is available to be supplied to the seabed during resuspension; and

erosion exposes anoxic regions of the seabed to oxic regions of the water column. The dependence of oxygen dynamics on those environmental conditions caused modeled estimates of oxygen consumption to be particularly sensitive to the supply and lability of organic carbon, rates of diffusion within the seabed, nitrification rate, and the frequency of resuspension. Our results imply that local resuspension may affect oxygen dynamics in other environments with similar characteristics.

**Appendix A**

This study modified the seabed layering scheme from Warner et al. (2008) to include biogeochemical tracers and diffusion of dissolved tracers between the seabed and water column (A.1), and to resolve millimeter-scale processes in surficial sediments while maintaining centimeter-scale resolution deeper in the seabed (A.2).

**A.1 Inclusion of biogeochemical tracers and seabed-water-column diffusion**

To couple the sediment transport and biogeochemical modules, we incorporated tracers representing particulate organic carbon and dissolved chemical species including oxygen and nutrients into the seabed module. To elaborate on the information presented in the Methods (Sect. 2.2), this section details how the sediment transport module was adapted from Warner et al. (2008) to account for them. The inclusion of particulate organic carbon was relatively straightforward because

the model treats it similarly to sediment classes, except that it decays in time. Inclusion of dissolved oxygen, nitrogen and ODU in the model, however, necessitated accounting for the formation of porewater within newly deposited layers and the entrainment of porewater into the water column during erosion, as described in Sect. 2.2.3, as well as diffusion of dissolved chemical constituents across the seabed-water interface, which is described below.

Our model parameterizes diffusion across the seabed-water interface by assuming that concentrations of dissolved tracers in the bottom water column and surficial seabed layer are equal. At each step, dissolved tracers move into or out of the seabed so that concentrations in the surficial seabed layer match those in the bottom water-column cell, while conserving tracer concentrations (symbols defined in Table 1):

$$C_{w\_tnew} = \frac{z_{w1}}{z_{w1}+z_a\times\Phi}\times\left(C_{w\_told}\times z_{w1} + C_{s\_told}\times z_a\times\Phi\right) \tag{A1}$$

$$C_{s\_tnew} = \left(1 - \frac{z_{w1}}{z_{w1}+z_a\times\Phi}\right)\times\left(C_{w\_told}\times z_{w1} + C_{s\_told}\times\Phi\right) \tag{A2}$$

Note that we also tested a second approach relying on a Fickian diffusion law with a diffusion coefficient of 1.09 x $10^9$ m$^2$ s$^{-1}$ based on Boudreau (1997) and Toussaint et al. (2014) to more directly account for diffusion across the seabed-water interface. Yet, both approaches yielded nearly identical results at the Rhône study site, and so we kept the simpler approach.

## A.2 Seabed resolution

Our seabed layering scheme is based on Warner et al. (2008), whose model includes a single, thin, active transport layer with thickness $z_a$, that represents the region of the seabed just below the sediment–water interface from which material can be entrained into the water column (Harris and Wiberg, 1997). This active transport layer, also called the surficial seabed layer, typically overlies a user-specified number of layers of uniform thickness, as well as a thick bottom layer that acts as a sediment repository. This scheme, however, can not resolve sub-millimeter scale changes in biogeochemical profiles near the seabed-water interface as well as cm-scale changes deeper in the seabed (e.g. Fig. 5), unless many seabed layers are used. Modifications to Warner et al. (2008)'s scheme therefore include incorporating both high-resolution and medium-resolution layers in the middle of the seabed.

Specifically, the layering scheme includes $N_{high\text{-}res}$ high-resolution layers with thickness $z_{high\text{-}res}$ immediately below the active transport layer, and then $N_{med\text{-}res}$ medium-resolution layers with thickness of $z_{med\text{-}res}$ in the middle of the seabed. After some experimentation, this study used 60 seabed layers, and $z_a$, $z_{high\text{-}res}$, $z_{med\text{-}res}$, $N_{high\text{-}res}$, and $N_{med\text{-}res}$ were set equal to 0.1 mm, 0.5 mm, 1 cm, 19 layers, and 39 layers (Table A.1). As in Warner et al. (2008), the bed layering scheme required that the number of layers remains constant; for this study, the number of "high" and "medium resolution" layers also remains constant, although their thicknesses may change slightly with erosion and deposition.

Incorporating multiple types of layers within the seabed and maintaining high resolution near the sediment–water interface affects how the layering scheme handles erosion and deposition. During depositional periods, new sediment is incorporated into surficial seabed layer(s) as described in Warner et al. (2008). When deposition increases the thickness of the surficial layer so that it exceeds ~$2*z_a$, the surficial layer is split into two, forming a thinner active transport layer and a new high-resolution layer, so that the surface layer remains thin. Similarly, if a high-resolution layer becomes thicker than $z_{high\text{-}res}$, this layer is also split into two layers. To maintain a constant number of layers, the bottommost high-resolution layer is then absorbed into the topmost medium-resolution layer. If adding material to the topmost medium-resolution layer causes it to exceed $z_{med\text{-}res}$ in thickness, the material from two medium-thick layers that are thinner than $z_{med\text{-}res}$ are combined or the bottommost medium-resolution layer is absorbed into the seabed repository. In contrast, during erosion, removal of high-resolution surface layers causes new high-resolution layers to split off from the topmost medium-resolution layer(s). When the topmost medium-resolution layer(s) is depleted, a new medium-resolution layer(s) is shaved off of the deep repository.

Additionally, the method of calculating the thickness of the surficial seabed layer, $z_a$, was changed to facilitate the representation of diffusive exchange across the seabed-water-column interface and to maintain high vertical resolution in the seabed. The CSTMS assumes that $z_a$ thickens with increasing bed shear stress, allowing sediment from deeper regions of the seabed to be entrained into the water column during energetic time periods (Harris and Wiberg, 1997; Warner et al., 2008).

During a resuspension event with bed shear stress of 2 Pa, this default parameterization would have thickened the surficial seabed layer to ~1.3 cm. Alternatively, some studies have constrained the active transport layer to smaller constant values, including 1 mm in the western Gulf of Lions (Law et al., 2008). For this biogeochemical-sediment transport model, it is important that the surface layer remain thin in order to represent the high gradients of oxygen observed at the seabed-water interface, and so $z_a$ is set equal to 0.1 mm to get reasonable oxygen penetration into the seabed. Overall, these adaptations from Warner et al. (2008) allow the seabed module to resolve mm-scale changes in seabed properties near the surface, while maintaining cm-scale resolution deeper in the seabed.

**Competing Interests**

Katja Fennel is a member of the editorial board of the journal.

**Acknowledgements**

Observations from the Rhône River Delta observatory (Mesurho) were provided by F. Toussaint (Laboratoire des Sciences du Climat et de l'Environnement). R. Wilson (formerly Dalhousie University) provided model code from Wilson et al. (2013). Feedback from E. Canuel, C. Friedrichs (Virginia Institute of Marine Science; VIMS), 2 anonymous reviewers, and Biogeosciences Associate Editor Jack Middelburg improved this paper. A. Miller, D. Weiss (VIMS), and E. Walters (the College of William & Mary; W&M) provided computational support and access to W&M's computing facilities, which are funded by the National Science Foundation, the Commonwealth of Virginia Equipment Trust Fund and the Office of Naval Research. Funding was provided by the U.S. National Oceanic and Atmospheric Administration Center for Sponsored Coastal Ocean Research (NA09NOS4780229, NA09NOS4780231, NGOMEX contribution 217) (Moriarty, Harris, Fennel, Xu), VIMS student fellowships (Moriarty), and MISTRALS/MERMEX-River and ANR-11-RSNR-0002/ AMORAD (Rabouille). This is contribution 3618 of the Virginia Institute of Marine Science.

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

**Table 1: Description of symbols used in this paper. Note that concentrations are porewater or bottom-water concentrations, not bulk concentrations, unless otherwise noted, but units of length and area (i.e., m and m$^2$) refer to the dimensions of the grid cell, and were not corrected for porosity.**

| Symbol | Description | Units |
|---|---|---|
| $Agg_{lab}$ | Concentration of labile aggregates | mmol N m$^{-3}$ |
| $Agg_{ref}$ | Concentration of refractory aggregates | mmol N m$^{-3}$ |
| $C_{ised}$ | Concentration of sediment from class *ised* | kg m$^{-2}$ |
| $C_{s\_tnew}$ | Concentration of dissolved tracer in the surficial seabed layer, for the new time step | mmol m$^{-3}$ |
| $C_{s\_told}$ | Concentration of dissolved tracer in the surficial seabed layer from the old time step | mmol m$^{-3}$ |
| $C_{w\_tnew}$ | Concentration of dissolved tracer in the bottom water-column layer, for the new time step | mmol m$^{-3}$ |
| $C_{w\_told}$ | Concentration of dissolved tracer in the bottom water-column layer from the old timestep | mmol m$^{-3}$ |
| $dO_2/dz_{OPD}$ | the slope of the vertical oxygen profile, averaged over the oxygen penetration depth, $z_{OPD}$ | [a]mmol O$_2$ m$^{-4}$ |
| $D_i$ | Coefficient for diffusion within the seabed for seabed constituent $i$ | m$^2$ s$^{-1}$ |
| $D_{ised}$ | Rate of deposition for sediment from class *ised* | kg m$^{-2}$ s$^{-1}$ |
| $D_{s-w}$ | Diffusion coefficient at the seabed water interface | m$^2$ s$^{-1}$ |
| dz | Grid cell thickness | m |
| $E_{ised}$ | Rate of erosion for sediment from class *ised* | kg m$^{-2}$ s$^{-1}$ |
| $f_{bur}$ | Fraction of organic matter that is buried in the seabed | --- |
| $f_{ised}$ | Fraction of the surficial seabed layer composed of sediment class *ised* | --- |
| $f_{lab}$ | Fraction of coagulated organic matter that is labile within the water column | --- |
| $f_{ldet}$ | Fraction of labile coagulated organic matter that is large detritus within the water column | --- |
| ised | Index used for different sediment classes. | --- |
| $k_{O2}$ | Half-saturation constant for O$_2$ limitation of aerobic remineralization | mmol O$_2$ m$^{-3}$ |
| $k_{O2\ nit}$ | Half-saturation constant for O$_2$ limitation of nitrification | mmol O$_2$ m$^{-3}$ |
| $k_{O2\ oduox}$ | Half-saturation constant for O$_2$ limitation of ODU oxidation | mmol O$_2$ m$^{-3}$ |
| $k_{NO3}$ | Half-saturation constant for NO$_3$ limitation of nitrate remineralization | mmol N m$^{-3}$ |
| $l_{O2}$ | Half-saturation constant for O$_2$ inhibition of nitrate remineralization | mmol O$_2$ m$^{-3}$ |
| $l_{O2\ anoxic}$ | Half-saturation constant for O$_2$ inhibition of anoxic remineralization | mmol O$_2$ m$^{-3}$ |
| $l_{NO3\ anoxic}$ | Half-saturation constant for NO$_3$ inhibition of anoxic remineralization | mmol N m$^{-3}$ |
| $L_{BO}$ | Limitation of seabed oxygen consumption due to bottom-water O$_2$ availability | --- |
| $L_{det}$ | Concentration of large detritus | mmol N m$^{-3}$ |
| $L_{tot}$ | Sum of the limitation factors on remineralization processes | --- |
| M | Erosion rate parameter representing seabed erodibility | kg m$^{-2}$ s$^{-1}$ |
| $NO_3$ | Nitrate concentration | mmol N m$^{-3}$ |
| $N_{high-res}$ | Number of high-resolution seabed layers | --- |
| $N_{med-res}$ | Number of medium-resolution seabed layers | --- |
| $NH_4$ | Ammonium concentration | mmol N m$^{-3}$ |
| $R_{anoxic}$ | Anaerobic Remineralization Rate in the seabed | [b]mmol C m$^{-3}$ d$^{-1}$ |
| $R_{aerobic}$ | Aerobic Remineralization Rate in the seabed | [b]mmol C m$^{-3}$ d$^{-1}$ |
| $R_{DNF}$ | Denitrification Rate in the seabed | [b]mmol C m$^{-3}$ d$^{-1}$ |
| $R_{nit}$ | Nitrification Rate in the seabed | mmol N m$^{-3}$ d$^{-1}$ |
| $R_{nit,max}$ | Maximum Nitrification Rate in the seabed | d$^{-1}$ |
| $R_{oduox}$ | Oxidation Rate of ODUs in the seabed | mmol O$_2$ m$^{-3}$ d$^{-1}$ |
| $R_{oduox,max}$ | Maximum Oxidation Rate of ODUs in the seabed | d$^{-1}$ |
| $R_{POC}$ | Remineralization rate constant for particulate organic matter in the seabed | d$^{-1}$ |
| $S_{inorganic}$ | Inorganic sedimentation rate | m y$^{-1}$, or kg m$^{-2}$ y$^{-1}$ |
| $S_{organic}$ | Particulate organic matter sedimentation rate | gC m$^{-2}$ y$^{-1}$ |
| $O_2$ | Dissolved oxygen concentration | mmol O$_2$ m$^{-3}$ |
| $O_{2,OPD}$ | Dissolved O$_2$ concentration at the oxygen penetration depth; equals zero by definition | mmol O$_2$ m$^{-3}$ |
| $O_{2,SWI}$ | Dissolved oxygen concentration at the seabed-water interface | mol O$_2$ m$^{-3}$ |
| ODU | Oxygen Demand Unit concentration | mmol O$_2$ m$^{-3}$ |
| POC | Particulate organic carbon concentration | [b]mmol C m$^{-3}$ |
| POM | Particulate organic matter concentration | [b]mmol N m$^{-3}$ |

| | | |
|---|---|---|
| $w_{s,ised}$ | Settling velocity of sediment from class *ised* | m s$^{-1}$ |
| $z_a$ | Thickness of seabed active transport layer | m |
| $z_{high-res}$ | Thickness of high-resolution seabed layers | m |
| $z_{med-res}$ | Thickness of medium-resolution seabed layers | m |
| $z_{newdep}$ | Thickness of new deposition | m |
| $z_{OPD}$ | Oxygen penetration depth into the seabed; this is negative in our coordinate system | m |
| $z_{SWI}$ | Depth at the seabed water interface (SWI); equals zero in our coordinate system | m |
| $z_{w1}$ | Thickness of bottom water-column grid cell | m |
| $\Phi$ | Seabed porosity | --- |
| $\tau_{bed}$ | Bed shear stress from waves and currents | Pa |
| $\tau_{crit}$ | Critical shear stress, assumed to be the same for all sediment classes. | Pa |
| $\tau_{crit,ised}$ | Critical shear stress for sediment class *ised* | Pa |

[a] m$^{-4}$ = m$^{-3}$ (of liquid) * m$^{-1}$(bulk distance)

[b] For this variable, m$^{-3}$ indicates volume of particulates in the grid cell, not water

**Table 2: Description of phrases, acronyms, and abbreviations, as used in this paper.**

| Acronym / Abbreviation | Description |
|---|---|
| Active transport layer | Region of the seabed from which material can be entrained into the water column; synonymous with the phrase 'active layer' in sediment transport papers (Harris and Wiberg, 1997; Warner et al., 2008). In the model, the active transport layer is the same as the surficial seabed layer. |
| Anoxic remineralization | Includes iron, manganese, and sulfur remineralization of organic matter, and methanogenesis, but not denitrification. |
| Bottom water | The region of the water column within 4 m of the seabed where suspended sediment concentrations were high during resuspension events |
| CSTMS | Community Sediment Transport Modeling System |
| Diagenesis | Within this paper, 'diagenesis' is used to refer to models that account for organic matter remineralization and associated biogeochemical processes within the seabed. We note, however, that diagenesis is commonly used to refer to any physical, chemical, geological, or biological changes in sediment or sediment rock following deposition, prior to metamorphism. |
| Diffusion at (or across) the seabed-water interface | Molecular diffusion of dissolved chemicals across the seabed-water interface. In the context of HydroBioSed, this refers to exchanges between the bottom water-column grid cell and surficial seabed layer so that they are in equilibrium (see Appendix). |
| Diffusion within the seabed | Molecular diffusion within the seabed; Referred to as 'biodiffusion' in other modeling papers when bioturbation is modeled as a diffusive process. |
| HydroBioSed | The coupled hydrodynamic–sediment transport– water-column and seabed biogeochemistry model developed and implemented in this study |
| Local resuspension | "One-dimensional" (vertical) resuspension, i.e. neglecting horizontal transport processes. |
| Module | Refers to a 'sub-model' within a model, e.g. the sediment transport module within ROMS |
| Nitrate remineralization | In this paper, synonymous with denitrification |
| Nutrient(s) | Refers to refer to nitrogen and/or phosphorus. Does not include ODUs |
| ODU | Oxygen Demand Unit; one ODU is the number of moles of reduced chemical species that react with one mole of $O_2$ when oxidized. |
| OPD | Oxygen Penetration Depth; Depth in the seabed at which oxygen decreased to zero. |
| POM | Particulate Organic Matter |
| Quiescent | Characterized by low-energy environmental conditions; i.e. used to refer non time periods with low waves and no resuspension in this paper |
| Re-deposition | Deposition of particulates previously resuspended from the same location |
| Resuspend, Resuspended | (verb, adjective) Refers to the entrainment of seabed material into the water column via erosion, or to the material that was eroded from the seabed |
| Resuspension (event) | (noun) Refers to cycle of erosion and deposition |
| ROMS | Regional Ocean Modeling System |
| Seabed | Region beneath the water column |
| Sediment | Inorganic particles |

Steady state                     Refers to models that do not change in time, e.g. due to wave-induced resuspension

**Table 3: Environmental conditions and parameters for the Standard Model implementation.**

| Model Input/Parameter | Modeled Value | Literature Source |
|---|---|---|
| **Hydrodynamic & Sediment Transport Parameter** | | |
| Water Depth | 24 m | Pastor et al. (2011a) |
| Wave Height | Observed time-series | Toussaint et al. (2014) |
| Wave Period | 10 s | Ulses et al. (2008), Palanques et al. (2006), Guillen et al. (2006) |
| Bottom-water Temperature | 15 $^{\circ}$C | Millot et al. (1990) |
| Surface Water Temperature | 20 $^{\circ}$C | Millot et al. (1990) |
| Bottom-water Salinity | 35 psu | Panlanques et al., 2006; Cruzado and Velasquez, 1990 |
| Surface Water Salinity | 33 psu | Panlanques et al., 2006; Cruzado and Velasquez, 1990 |
| Inorganic Sedimentation Rate | $S_{inorganic}$ = 10 cm $y^{-1}$ = 14 kg $m^{-2}$ $y^{-1}$ | Pastor et al. (2011a) |
| Fraction of Sediment that is Muddy Flocs | 80 % | Roussiez et al. (2006), Ferre et al. (2005), Radkovitch et al. (1999) |
| Fraction of Sediment that is Sand | 20 % | Roussiez et al. (2006), Ferre et al. (2005), Radkovitch et al. (1999) |
| Settling Velocity of Muddy Flocs | 0.19 mm $s^{-1}$ | Curran et al. (2007) |
| Settling Velocity of Sand | 30 mm $s^{-1}$ | Curran et al. (2007) |
| Critical Bed Shear Stress | $\tau_{crit}$ = 0.3 Pa | [a]Toussaint et al. (2014) |
| Erosion Rate Parameter | M = 0.01 kg $m^{-2}$ $s^{-1}$ | [a]Toussaint et al. (2014) |
| Porosity | $\Phi$ = 0.9 | Unpublished data |
| Sediment Density of Muddy Flocs | [b]1048 kg $m^{-3}$ | Curran et al. (2007) |
| Sediment Density of Sand | [b]2650 kg $m^{-3}$ | Curran et al. (2007) |
| **Water-column Biogeochemical Parameters** | | |
| Oxygen Concentration | 253 mmol $O_2$ $m^{-3}$ | Toussaint et al. (2014), Pastor et al. (2011a) |
| Nitrate Concentration | 0.5 mmol N $m^{-3}$ | Pastor et al. (2011a) |
| Ammonium Concentration | 5.8 mmol N $m^{-3}$ | Pastor et al. (2011a) |
| ODU Concentration | 0 mmol $O_2$ $m^{-3}$ | Pastor et al. (2011a) |
| Phytoplankton Concentration | 0.03 mmol N $m^{-3}$ | [c]Pastor et al. (2011a) |
| Zooplankton Concentration | 1.17 mmol N $m^{-3}$ | [c]Pastor et al. (2011a) |
| Small Detritus Concentrations | 0.03 mmol N $m^{-3}$ | [c]Pastor et al. (2011a) |
| Maximum Nitrification Rate | 0.7 $d^{-1}$ | Pinazo et al. (1996) |
| Coagulation Rate of Phytoplankton and Small Detritus | 182 $d^{-1}$ | [c]Pastor et al. (2011a) |
| Detritus & Aggregate Remineralization Rate Constant | 11 $y^{-1}$ | Pinazo et al. (1996) |
| Settling (Sinking) Velocity of Phytoplankton | 0.1 m $d^{-1}$ | [d]Fennel et al. (2006) |
| Settling (Sinking) Velocity of Large detritus | 1.0 m $d^{-1}$ | [d]Fennel et al. (2006) |
| Settling (Sinking) Velocity of Small detritus | 0.1 m $d^{-1}$ | [d]Fennel et al. (2006) |
| Settling (Sinking) Velocity of Labile Aggregates | 16.416 m $d^{-1}$ | Curran et al. (2007) |
| Settling (Sinking) Velocity of Refractory Aggregates | 16.416 m $d^{-1}$ | Curran et al. (2007) |
| Nudging Parameter for Large detritus, Aggregates, Sediment | 0 $d^{-1}$ | N/A |
| Nudging Parameter for $NO_3$, Phytoplankton, Small Detritus | 0.02 $d^{-1}$ | N/A |
| Nudging Parameter for $NH_4$, Oxygen, ODU, Zooplankton | 0.2 $d^{-1}$ | N/A |
| POM Sedimentation Rate | $S_{organic}$ = 657 gC $m^{-2}$ $y^{-1}$ | Pastor et al. (2011a) |
| Partitioning of Refractory vs. Labile Organic Matter | $f_{lab}$ = 0.5 | Pastor et al. (2011a), Tesi et al. |

| | | (2007) |
|---|---|---|
| Partitioning of Labile Aggregates vs. Large Detritus | $f_{ldet} = 0.5$ | Pastor et al. (2011a), Tesi et al. (2007) |

**Seabed Biogeochemical Parameters**

| | | |
|---|---|---|
| Labile Organic Matter Remineralization Rate Constant | 11 $y^{-1}$ | Pastor et al. (2011a) |
| Refractory Organic Matter Remineralization Rate Constant | 0.31 $y^{-1}$ | Pastor et al. (2011a) |
| Ratio of mol C: mol N in Labile Organic Matter | 7.10 | Pastor et al. (2011a) |
| Ratio of mol C: mol N in Refractory Organic Matter | 14.3 | Pastor et al. (2011a) |
| Half-Saturation Constant for $O_2$ Limitation of Aerobic Remineralization | $k_{O2} = 1$ mmol $O_2$ $m^{-3}$ | Pastor et al. (2011a) |
| Half-Saturation Constant for $NO_3$ Limitation of Nitrate Remineralization (Denitrification) | $k_{NO3} = 20$ mmol N $m^{-3}$ | Pastor et al. (2011a) |
| Half-Saturation Constant for $O_2$ Limitation of Nitrification | $k_{O2\ nit} = 10$ mmol $O_2$ $m^{-3}$ | Pastor et al. (2011a) |
| Half-Saturation Constant for $O_2$ Limitation in ODU Oxidation | $k_{O2\_oduox} = 1$ mmol $O_2$ $m^{-3}$ | Pastor et al. (2011a) |
| Half-Saturation Constant for $O_2$ Inhibition of Nitrate Remineralization (Denitrification) | $l_{O2} = 1$ mmol $O_2$ $m^{-3}$ | Pastor et al. (2011a) |
| Half-Saturation Constant for $O_2$ Inhibition of Anoxic Remineralization | $l_{O2\_anoxic} = 1$ mmol $O_2$ $m^{-3}$ | Pastor et al. (2011a) |
| Half-Saturation Constant for $NO_3$ Inhibition of Anoxic Remineralization | $l_{NO3\_anoxic} = 10$ mmol $NO_3$ $m^{-3}$ | Pastor et al. (2011a) |
| Maximum Nitrification Rate | $R_{nit,max} = 100$ $d^{-1}$ | Pastor et al. (2011a) |
| Maximum Oxidation Rate of Oxygen Demand Units | $R_{oduox,max} = 20$ $d^{-1}$ | Pastor et al. (2011a) |
| Fraction of ODUs Produced that are Solid and Inert | 99.5 % | Pastor et al. (2011a) |
| Diffusion Coefficient for Across Seabed-Water Interface | $D_{s-w} = 1.08 \cdot 10^{-9}$ $m^2$ $s^{-1}$ | Toussaint et al. (2014) |
| Coefficients for Diffusion Within the Seabed | $D_{particulates} = 2.55 \cdot 10^{-10}$ $m^2$ $s^{-1}$ $D_{O2} = 11.99 \cdot 10^{-10}$ $m^2$ $s^{-1}$ $D_{NO3} = 9.80 \cdot 10^{-10}$ $m^2$ $s^{-1}$ $D_{NH4} = 10.04 \cdot 10^{-10}$ $m^2$ $s^{-1}$ $D_{ODU} = 4.01 \cdot 10^{-10}$ $m^2$ $s^{-1}$ | [e]Pastor et al. (2011a) |

[a]Chosen based on time series of seabed elevation in Toussaint et al. (2014)
[b]Units are $m^3$ sediment, not $m^3$ water
[c]Chosen based on organic sedimentation rate
[d]No local data
[e]Derived from the molecular diffusion rates, but adjusted for the porosity and tortuosity of the seabed as described in Pastor et al., 2011a.

**Table 4: List of sensitivity tests. Additionally, for each simulation listed here, an identical model run was completed that neglected resuspension (i.e. with M = 0 kg/m²/s ; $\tau_{crit}$ = 10 Pa).**

| Sensitivity Test Abbreviation | Sensitivity Test Name | Changed Parameters and/or Parameterizations Relative to the Standard Model Run |
|---|---|---|
| R1 | Low Erosion Rate Parameter | M = 0.005 kg $m^{-2}$ $s^{-1}$ |
| R2 | High Erosion Rate Parameter | M = 0.02 kg $m^{-2}$ $s^{-1}$ |
| T1 | Low Critical Shear Stress | $\tau_{crit}$ = 0.15 Pa |
| T2 | High Critical Shear Stress | $\tau_{crit}$ = 0.6 Pa |
| S1 | Low Inorganic Sedimentation | $S_{inorganic}$ = 0.05 m $y^{-1}$ = 7 kg $m^{-2}$ $y^{-1}$ |
| S2 | High Inorganic Sedimentation | $S_{inorganic}$ = 0.20 m $y^{-1}$ = 28 kg $m^{-2}$ $y^{-1}$ |
| P1 | Low Particulate Organic Sedimentation | $S_{organic}$ = 328.5 gC $m^{-2}$ $y^{-1}$ |
| P2 | High Particulate Organic Sedimentation | $S_{inorganic}$ = 1314 gC $m^{-2}$ $y^{-1}$ |
| L1 | Low Lability | $f_{lab}$ = 0.20 |
| L2 | High Lability | $f_{lab}$ = 0.80 |
| B1 | Low Seabed Diffusion | $D_i$ = original values * 0.5 |
| B2 | High Seabed Diffusion | $D_i$ = original values * 2.0 |
| N1 | Low Nitrification Rate | $R_{nit,max}$ = 50 $d^{-1}$ |
| N2 | High Nitrification Rate | $R_{nit,max}$ = 200 $d^{-1}$ |
| C1 | No-Repartitioning | See Fig. 2c; Sect. 2.2.3 |

**Table 5: Statistics for model-observation comparison, including the root mean square difference (RMSD) and the correlation coefficient (R). The mean and standard deviation of estimates from both the model and observations are also shown.**

| | RMSD | R | Mean ± Standard Deviation | |
|---|---|---|---|---|
| | | | Model | Observations |
| **Seabed Height** | 1.39 cm | 0.21 | -0.52 ± 0.82 cm | -1.1 ± 1.2 cm |
| **O$_2$ Gradient** | 105 mol O$_2$ m$^{-4}$ | 0.48 | 180 ± 118 mol O$_2$ m$^{-4}$ | 173 ± 76 mol O$_2$ m$^{-4}$ |

**Table 6: O$_2$ Consumption (mmol O$_2$ m$^{-2}$ d$^{-1}$) in the seabed, bottom water, and combined seabed-bottom water due to various processes over the two-month model run, and during periods of deposition and erosion. Abbreviations include: POM Rem. (particulate organic matter remineralization); ODU Ox (Oxidation of ODUs); Nit (nitrification); and "Seabed + BW" (the combined seabed-bottom-water region).**

| | Seabed | | | | Bottom Waters | | | | Seabed + BW |
|---|---|---|---|---|---|---|---|---|---|
| | Total | POM Rem. | Nit. | ODU Ox. | Total | POM Rem. | Nit. | ODU Ox. | Total |
| **2-Month Average** | 19 | 5.0 | 14 | 0.20 | 56 | 31 | 24 | 0.30 | 74 |
| **Minimum Values over 2 Months** | 12 | 0.56 | 3.7 | 0.01 | 23 | 0.08 | 22 | 0 | 39 |
| **Maximum Values over 2 Months** | 35 | 18 | 33 | 0.64 | 200 | 170 | 34 | 10. | 220 |
| **Average During Depositional Periods** | 18 | 5.5 | 12 | 0.18 | 47 | 23 | 24 | 0.18 | 65 |
| **Average During Erosional Periods** | 21 | 3.3 | 18 | 0.26 | 90. | 63 | 26 | 0.78 | 110 |

**Table A.1: Parameters for new seabed layering scheme, as implemented for the Rhône study site. Dashed lines indicate that no symbol was assigned to that parameter.**

| Type of Layer | Symbol for Number of Layers | Number of Layers for Rhône model implementation | Symbol for Thickness of Each Layer | Thickness of Each Layer for Rhône model implementation (mm) |
|---|---|---|---|---|
| **Active Transport Layer** (i.e., the Surficial Layer) | -- | 1 | $z_a$ | 0.1 |
| **High-Resolution Layers** | $N_{high\text{-}res}$ | 19 | $z_{high\text{-}res}$ | 0.5 |
| **Medium-Resolution Layers** | $N_{med\text{-}res}$ | 39 | $z_{med\text{-}res}$ | 10 |
| **Repository** | -- | 1 | | Varies; 333 m at initialization |

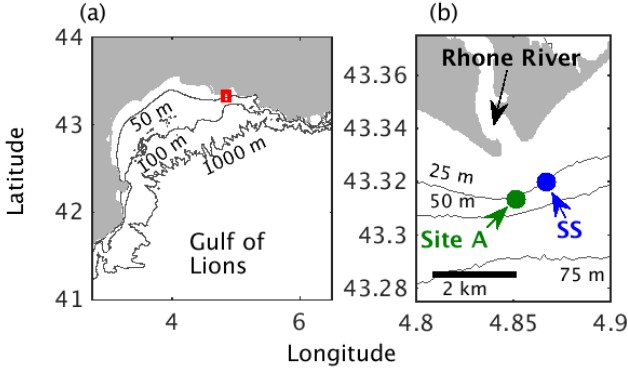

**Figure 1: a) Red box indicates location of panel (b) in the Gulf of Lions. b) Dots indicate our study site (SS; blue), i.e. the Mesurho station (Pairaud et al., 2016), and Pastor et al. (2014)'s Site A (green) offshore of the Rhône River. Bathymetric data (black lines)**

**were obtained from the European Marine Observation and Data Network. Coastline data were obtained from the U.S. National Oceanic and Atmospheric Administration.**

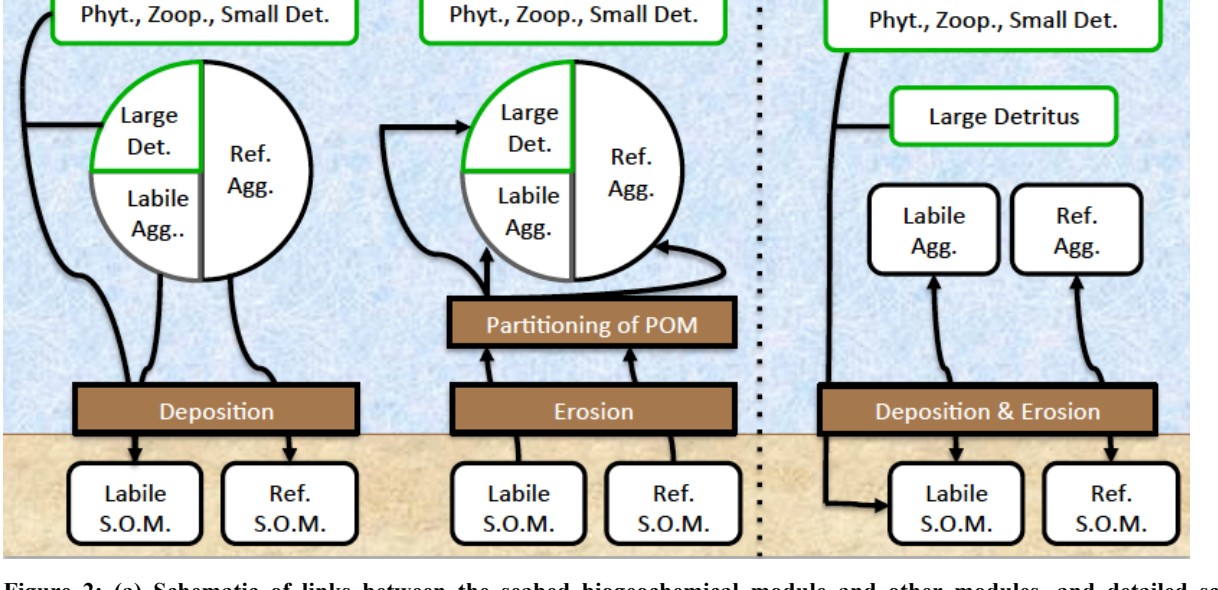

**Figure 2: (a) Schematic of links between the seabed biogeochemical module and other modules, and detailed schematics of particulate organic matter partitioning for the (b) standard model run and (c) no-repartitioning sensitivity test. The colors of the boxes and labels indicate processes associated with sediment transport (brown), water-column biogeochemistry (green) and seabed**

**biogeochemistry and model coupling (black). Abbreviations for this figure represent sediment (Sed.), biogeochemistry (Biogeochem.), phytoplankton (Phyt.), zooplankton (Zoop.), detritus (Det.), seabed organic matter (S.O.M.), aggregates (Agg.), labile (Lab.) and refractory (Ref.).**

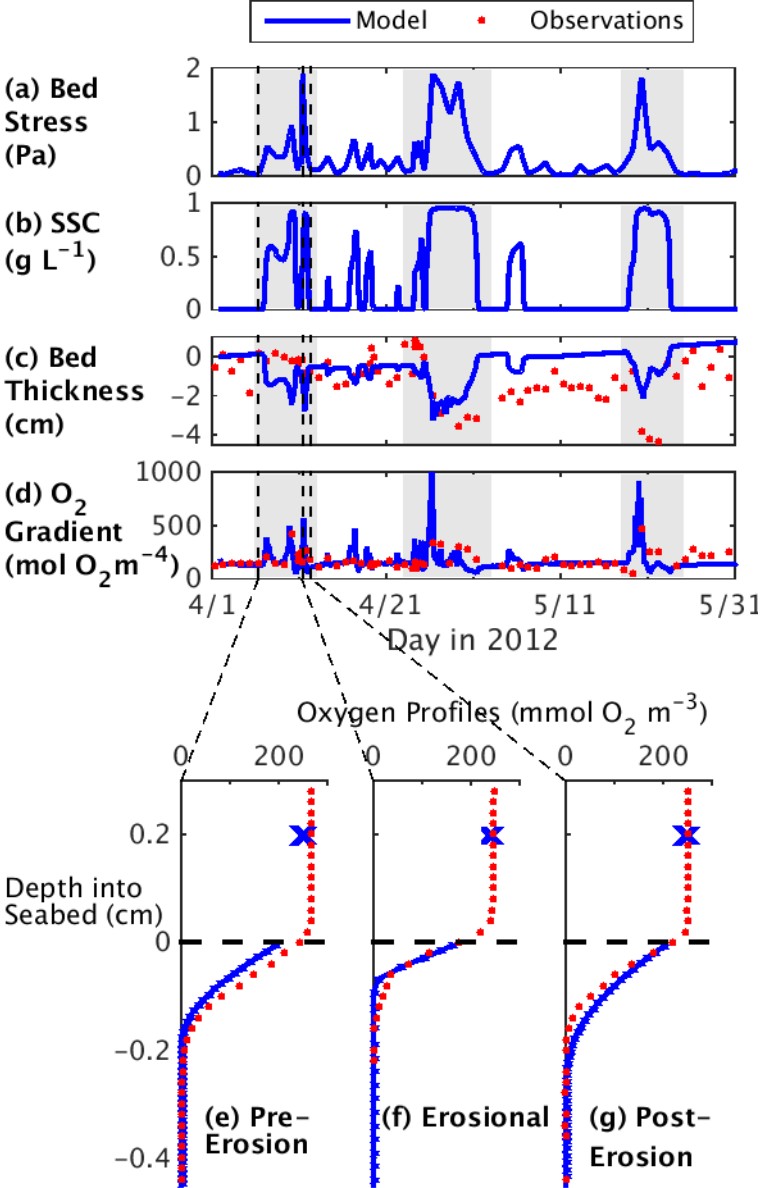

Figure 3: Time series of modeled (blue lines & x's) and observed (red dots; Toussaint et al., 2014) bed stress, near-bed suspended sediment concentrations (SSC), seabed height, and vertical oxygen gradient averaged over the oxic layer of the seabed (top 4 panels), and three examples of oxygen profiles before (6 April 2012), during (9 April 2012), and after (12 April 2012) an erosional event in early April (bottom panels). The dashed black lines in the bottom panels indicate the seabed-water interface. Shading in the top panels indicates resuspension events, i.e. cycles of erosion and re-deposition, including 6–13 April, 23 April–3 May, and 18–25 May 2012.

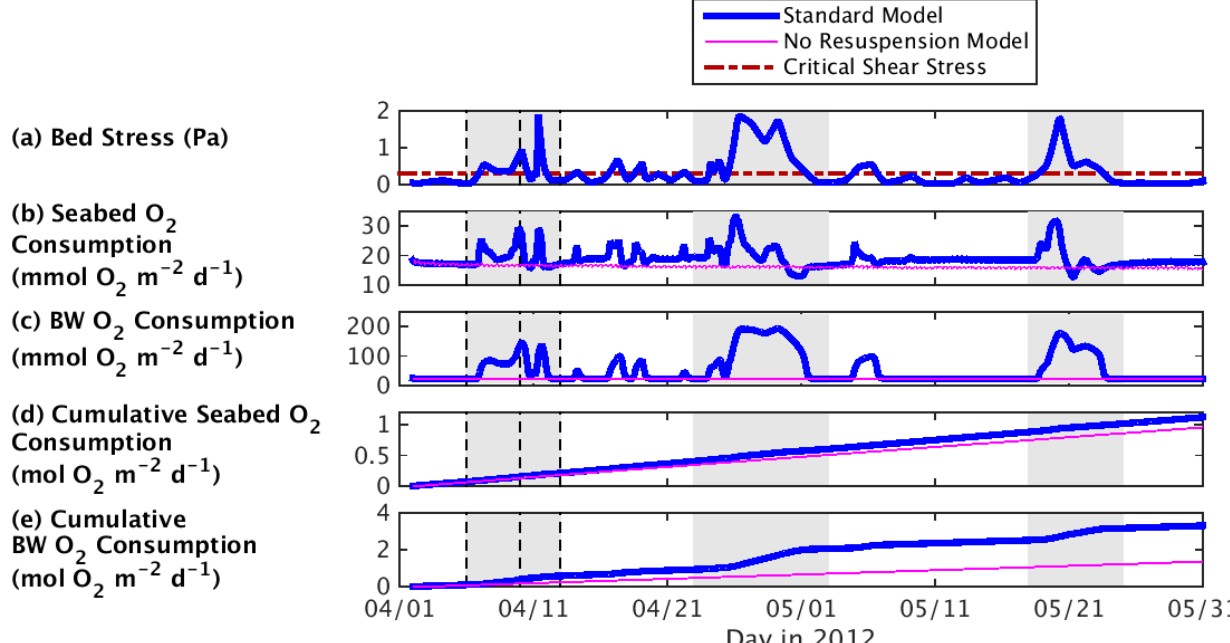

**Figure 4: Time series of bed stress and oxygen consumption in the seabed and bottom water (BW) for both the standard (blue solid line) and no-resuspension model runs (pink line). Shading indicates resuspension events, i.e. cycles of erosion and re-deposition, as listed in Fig. 3. The red dashed line indicates the critical shear stress for erosion, and the black dashed lines indicate the times at which profiles in Fig. 5 were estimated.**

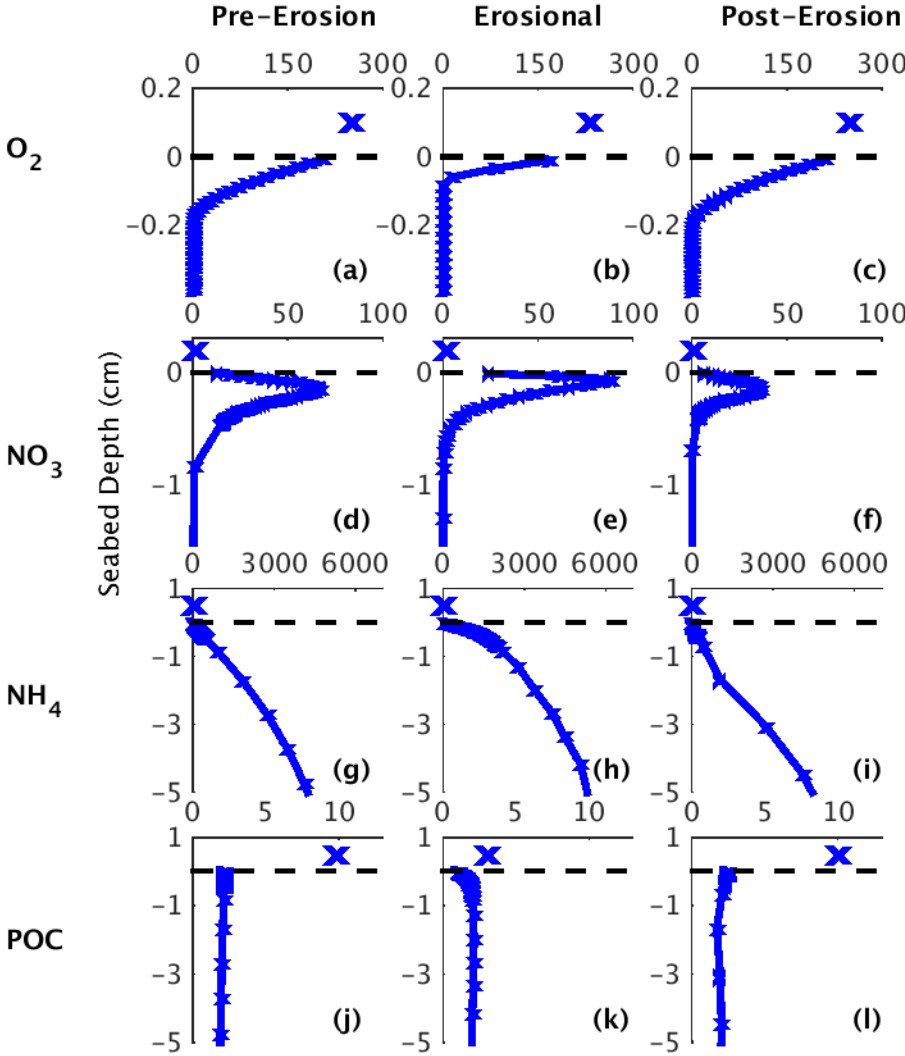

**Figure 5: Seabed profiles of oxygen (top row; mmol O$_2$ m$^{-3}$), nitrate (second row; mmol N m$^{-3}$), ammonium (third row; mmol N m$^{-3}$), and degradable particulate organic carbon (POC; bottom row; dry weight (%)) from the standard model run for times immediately preceding the mid-April resuspension event (6 April 2012, left column), during the erosional period (10 April 2012, center column), and during the depositional period (13 April 2012, right column). Fig. 4 shows the times at which the profiles were estimated. Tickmarks on the blue lines indicate the location of each seabed layer. The black dashed lines indicate the seabed water interface, and all seabed depths are given relative to this interface. The 'X's indicate near-bed values for the water column.**

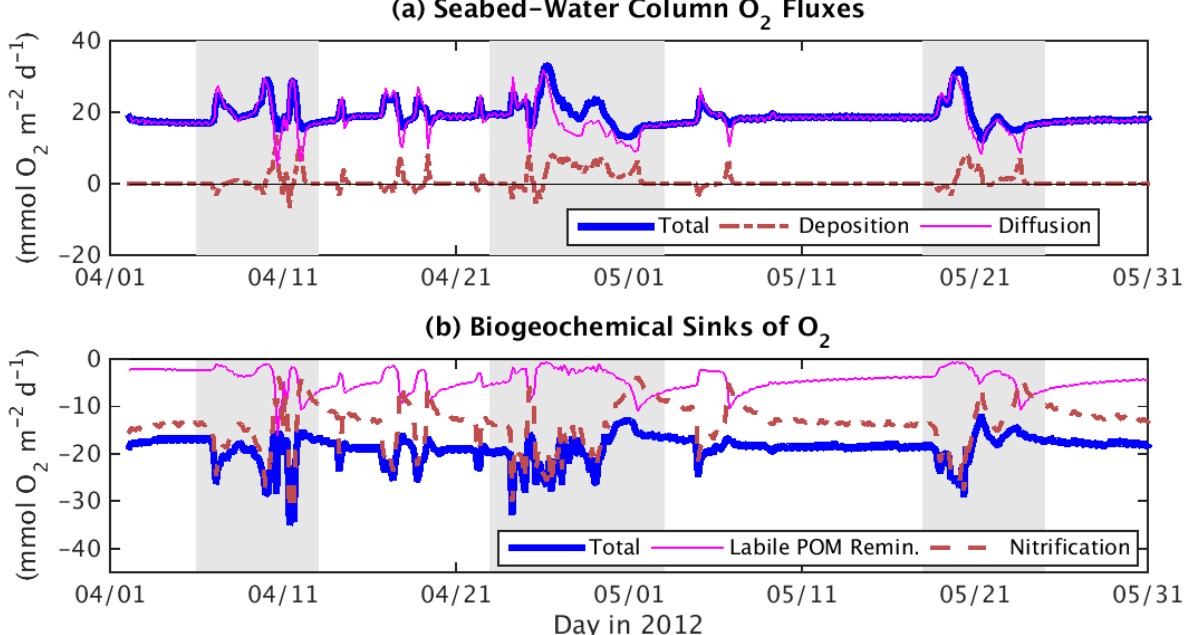

**Figure 6: Physical (top) and biogeochemical (bottom) sources and sinks of oxygen within the seabed for the standard model run. Sources and sinks of oxygen to the seabed are positive and negative, respectively. Small biogeochemical sinks <1 mmol O$_2$ m$^{-2}$ d$^{-1}$ (ODU oxidation and remineralization of refractory POM) are not shown. Shading indicates resuspension events, i.e. cycles of erosion & deposition, including 6–13 April, 23 April–3 May, and 18–25 May, 2012.**

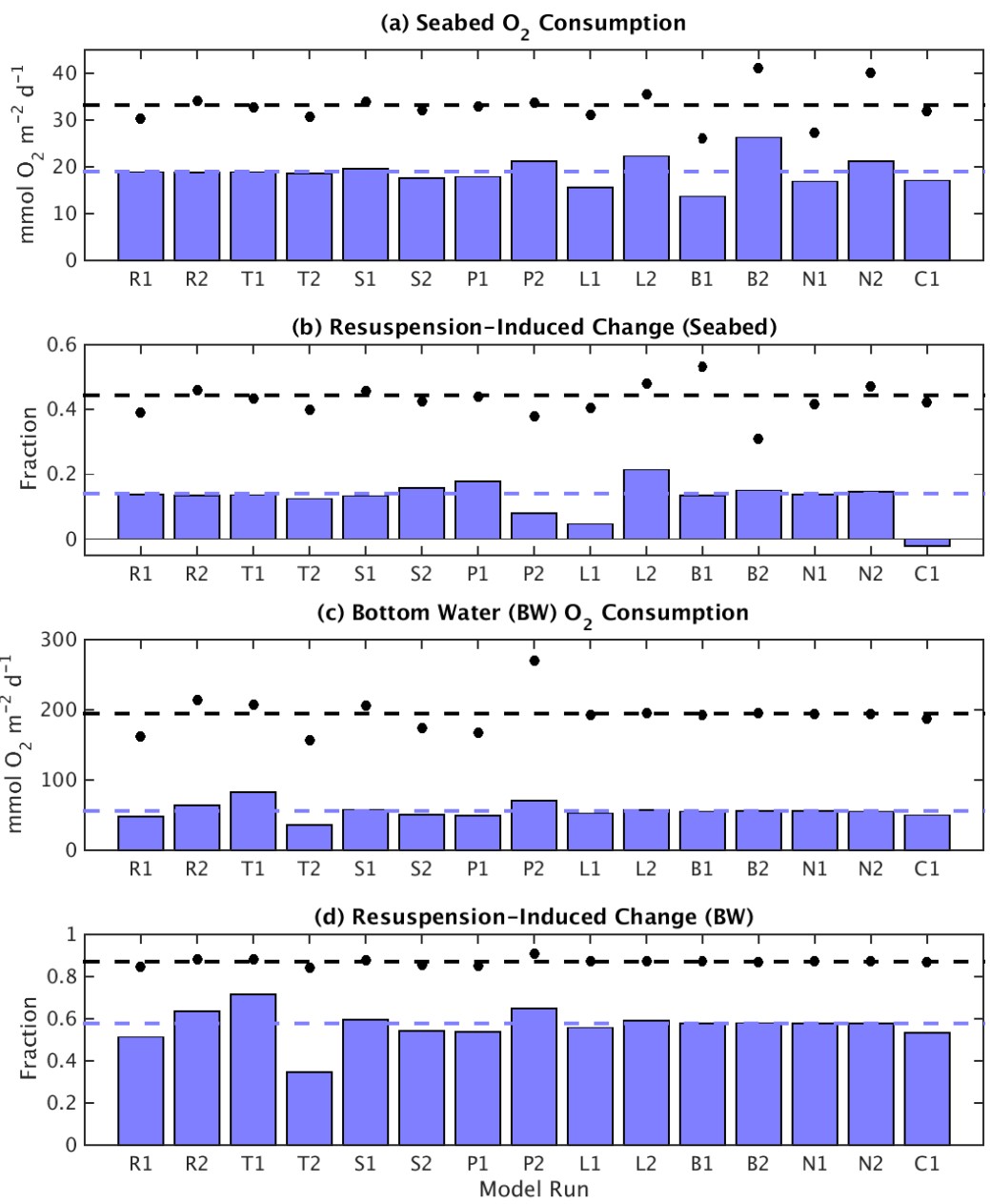

**Figure 7: Rate of oxygen consumption in the (a) seabed and (c) bottom waters for each sensitivity test listed in Table 4. Fraction of (b) seabed and (d) bottom-water oxygen consumption induced by resuspension, calculated by dividing the difference between each sensitivity test and its no-resuspension model run by the value from the sensitivity test. In both panels, bars represent averages over two months. Dots indicate the maximum values during this two-month period (which occurred during resuspension events). The dashed lines represent values from the standard model run, with the color of the line consistent with the type of data it represents (i.e. two-month average or maximum value).**

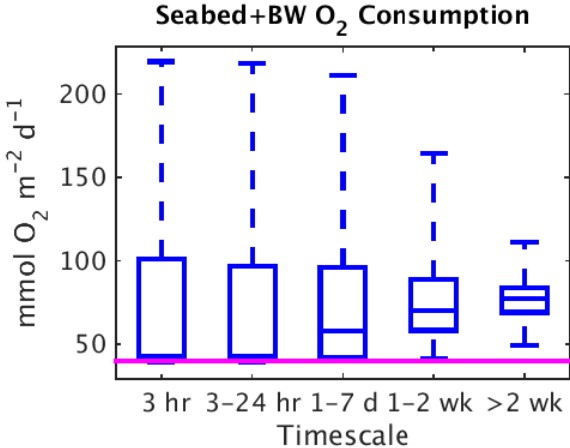

Figure 8: **Box and whisker plot indicating the 0$^{th}$, 25$^{th}$, 50$^{th}$, 75$^{th}$, and 100$^{th}$ percentiles of combined seabed-bottom-water (BW) oxygen consumption averaged over different timescales for the standard model run. The pink lines indicate estimates from the no-resuspension model run.**