# Peer review of "The roles of resuspension, diffusion and biogeochemical processes on oxygen dynamics offshore of the Rhône River, France: a numerical modeling study"

_Biogeosciences, 2016_

## Referee Comment (RC1) · Anonymous Referee #1 · 20 Dec 2016

General comments

This manuscript presents the development of the HydroBioSed module within the ROMS-CSTMS framework. This module couples existing hydrodynamic, sediment transport, and water-column biogeochemical modules with a seabed biogeochemistry module, with the aim of investigating the role of resuspension on biogeochemical dynamics, most notably oxygen. This novel module is then applied to the Rhone Delta, showing that local resuspension has an important impact on oxygen dynamics on at least seasonal timescales. The paper also classifies systems where the impact of intermittent local resuspension on oxygen dynamics may be substantial.

I believe that this work is a necessary addition to benthic-pelagic coupling of reactive transport models, and an advancement in the field. The manuscript is well written and clearly organized, actually even too organized to my liking, by announcing even every subsection in detail. I do however believe that sometimes a little more detail might be nice in the method section, such that the reader unfamiliar with ROMS and/or CSTMS can also well understand the model development. Specifically, I would add the supplement to the main paper, as it contains details that are necessary to fully understand the method development, which is a major aim of this paper. Moreover, I only became aware of my biggest concern with the paper after having read the supplement. This major concern is related to the way the model deals with the lability of resuspended organic matter. I do not believe that it is valid to reclassify refractory resuspended organic matter into the labile pools, and believe it is the main reason why the re-deposited organic matter is enriched in labile pools (p.10, l.31-32), which I consider an unexpected and also unrealistic result that may highly influence the estimated seabed oxygen consumption. I will explain this concern in more detail below, but I believe this issue needs to be resolved before the manuscript can be published. Aside from that, and a few other comments related to the methodology, my comments are mainly technical and deal with how results are presented.

Specific comments

- p.4, l.9-10: is there a way that these resuspension events can be predicted, e.g. from tides or wind speeds? I am asking this in the context of potential model applications.

- p.4, l.21-22: What is the estimated seabed oxygen consumption at the study site itself?

- p.5, l.15: when looking at Table 1, it becomes evident that this equation calculates an amount of sediment in areal units, i.e. in kg/m2. Do I have to see this as a depth-integrated concentration from the active transport layer (so a concentration multiplied

by za), or rather as sort of a flux across the sediment-water interface? What does the erosion rate parameter M represent exactly?

- p.5, l.18: In terms of mass, how much of POM is added relative to the number of inert particulates? I am asking this to see if adding POM as an additional particle source can lead to inconsistencies in e.g. estimated deposition rate compared to previous applications. Keeping this in mind, wouldn't it make more sense conceptually if the inert particulates pool would be split into a POM pool and an inert pool?

- p.5, l.22: The information of S.1 needs to be added to this section, especially since it is so important for the conceptual model. My main problem with the conceptual model is that the repartitioning of refractory resuspended organic matter into the labile pools does not seem a valid approach. I understand that resuspension may change the degree of aggregation, but I do not see why it would change the lability per se. The arguments given on p.2, l.5-8 of the supplement may be true, but can be implemented into the model in a conceptually better way. Faster OM remineralization under oxic conditions is already implemented in the model via the limitation factors and half-saturation constants. Higher remineralization rates in the water column can also be achieved by using different rate constants in the water column compared to the sediment, but in reality, I think this is due to the constant supply of fresh OM to the water column due to in-situ production and/or external loading. The former is already part of the model, whereas the latter might be implemented by an additional source term to the water column.

- p.6, l.1: to this section, the information of S.2 should be added. I was really missing this while reading this section, it is too important for the model description to be part of the supplement.

- p.6, l.17-21: I believe that these equations are currently incorrect, when following the units of Table 1. The right-hand side of Eqs. 2-6, is in mmol/m3/d, whereas the left-hand side is in mmol/m2/d. I would anyway stick with mmol/m3 and mmol/m3/d

everywhere throughout the manuscript, even though the model stores concentrations in a depth-integrated way, as this is the conventional way to present rates and concentrations.

- p.7, l.1-2: My first thought when reading this was: "shouldn't resuspension and re-deposition have a huge impact on porosity in the surficial centimetre of the seabed"? As far as I know, porosity can fluctuate a lot even in the top centimetre depending on the amount of deposition and/or resuspension. I wonder I the authors have tested the effect of a different porosity, even if kept constant with depth; perhaps this should be part of the sensitivity analysis.

- p.7, l.8-12: I might have missed this, but are sediment deposition and erosion mutually exclusive? I understand that erosion only occurs in case of high bed stress, but does this mean that there is no deposition at all during erosion, or that both processes co-occur but that erosion dominates?

- p.7, l.30: What are the units of the 5x6 model grid?

- p.9, l.4-6: It would be good to stress here that the bottom boundary layer is the layer below the pycnocline, as mentioned on p.7, l.32. Otherwise, this definition lacks a bit of context (i.e. why was 4 m chosen as thickness of the BBL?)

- p.9, l.7-9: Why are the rates of biogeochemical processes in the BBL not added to the code as output parameters?

- p.9, l.28: I It seems that, since zSWI and O2,OPD are by definition 0, O2,SWI is positive, and zOPD is negative (at least according to Figs. 3 and 5), Eq.8 would by default result in a negative value, which is not the case. So this needs to be changed to make it consistent.

- p.10, l.5-7: Have the authors considered implementing a loss term of the resuspended material, as a possible workaround to the uniform conditions in the horizontal?

- p.10, l.26-27: How were the erosion depths quantified? Is erosion depth more dependent on the duration of the event, or on the bed stress?

- p.10, l.31-32: This is a result I do not understand, and also do not believe to be an accurate representation of reality. I believe this is due to the parameterisation of the lability of the resuspended material, which I discussed above.

- p.11, l.22-27: I'm not sure if I understand this argument. Isn't this faster oxygen consumption during re-deposition driven by the newly-deposited OM being more labile, rather than additional oxygen availability?

- p.11, l.26: Wouldn't 'quasi-steady state' be a better term there than 'equilibrium'?

- p.12, l.1-2: Why would entrainment of nitrate increase the nitrification rate? Eq.5 does not show any NO3-limitation.

- p.12, l.10-12: This sentence seems to lack something. I was expecting to read "..com-pared to xxx" at the end.

- p.12, l.22-25: Fluxes of NH4 and ODU are presented here, it would be nice to also present them in a figure or table (similar to Table 6).

- p.14, l.14: all 3 resuspension events presented here have a duration of days (grey-shaded areas of Fig 4). I don't see any events with a duration of hours. Were the shorter events not captured by the model, or did they not take place during this two-month period?

- p.15, l.15: The Almroth-Rosell et al. (2011) paper shows most resuspension at other locations in the Baltic Sea, though it is high in certain parts of the Gulf of Finland. Perhaps the authors can elaborate a bit on this in their discussion.

- p.15, l.28-29: I am perfectly fine with keeping temperature constant in the model, but I do think that the impact of a higher temperature should be tested by increasing rate constants according to the Q10 rule. That would be a very valuable sensitivity analysis.

- p.16, l.3-5: Wouldn't OM loading also be lower in winter and thereby impacting oxygen

consumption?

- p.16, l.13-16: What is the spatial scale of these other models as compared to this model? Could differences therein be an argument for their different handling of oxygen consumption due to resuspension?

- p.30, Table 3: How has this sedimentation rate been estimated? If e.g. by 210Pb dating, wouldn't it be more correct to say that this is the total sedimentation rate, that then needs to be partitioned into an inorganic and an organic part? (see also comment on p.5, l.18)

- p.31, Table 3: What does the fraction of ODUs that are solid and inert represent? Since denitrification is modelled separately, does N2 also belong to the ODU pool here? If not, this number may be quite high.

- p.33, Figure 2: I do not find this figure very clear. What I would be interested to know is how suspended sediment, organic aggregates and organic particulates from the water column biogeochemical module are interlinked. I.e. a schematic of supplement S.1, showing that e.g. labile water-column organic matter is the sum of phytoplankton, detritus and labile aggregates, and that this is part of the total suspended matter.

Technical corrections

- A general comment: I get the impression that the terms 'resuspension event' and 'erosional period' seem to be used interchangeably (e.g. in the caption of Fig. 5), is this correct?

- p.3, l.24-26: I would leave out the "other factors" from the research question, it sounds too vague and results of other factors are not presented (only mentioned to be negligible in the case of ODU oxidation rate and the diffusion scheme)

- p.3, l.28: Section 1.1 fits better as the first section of the Methods.

- p.3, l.32: 'located at ~25m water depth'

- p.3, l.33: I would prefer 'model validation' over 'model evaluation' here

- p.8, l.28: the ODU oxidation rate sensitivity test is mentioned here as not being presented further, but it is part of Table 4. It should be removed from the table.

- p.9, l.10: 'it's' should be 'its'

- p.9, l.29: Table 5 does not show this; remove the reference here.

- p.12, l.21: The contribution of ODUs is virtually 0, as Table 6 shows, and negligible compared to the other processes, so remove it here from the text.

- p.16, l.10: change 'produce' to 'reproduce'

- p.32, Table 6: The total oxygen consumption is higher than the sum of the separate processes. Maybe this is because minimum and maximum values are presented, but also for the average, this is not the case.

- p.14, l.30: "reduced species for oxidation". This is not shown by the model, as the contribution of ODUs to oxygen consumption in the BBL is really minor. Remove this statement.

- p.15, l.25-26: Too many references here.

- p.16, l.19-23: More appropriate than what?

- p.17, l.12-17: This section might better fit in Section 4.1.2, when discussing temporal variations

- p.17, l.32-33: Something seems to be missing in this sentence.

- p.28/29, Table 1: carefully check the units, especially in relation to Eq. 2-5.

- p.32, Table 6: Add vertical lines between the three different categories as specified on the upper row. Also, why aren't averages presented of the depositional (white parts of Figs. 4 and 6) and erosional (grey parts of Figs. 4 and 6) periods presented, rather than only an average of the total two-month period? That seems clearer and more

meaningful to me. Finally, clearly specify whether the maximum and (especially) mini-
mum values presented are from the whole two-month period or from the erosional parts
only.

- p.34, Figure 3: Define SSC in the figure caption. Add the dates to the oxygen micro-
profiles.

- p.35, Figure 4: Add a horizontal line to panel a showing the critical shear stress. Panel
b could use a differently scaled y-axis, to make temporal variations clearer.

- p.36, Figure 5: Why are different times chosen here than in Figure 3? For clarity
and consistency, it would be nicer to study the same erosional event in both figures.
The depth scale of the O2 profile may be changed for clarity. Also for clarity, the cap-
tions 'pre-resuspension', 'during resuspension' and 'post-resuspension' (or something
equivalent) could be added to the three columns.

- p.37, Figure 6: I don't see any oxygen sources in panel b, only sinks. So I would
change the caption here. Also, I would replace 'fluxes' in the caption with 'sources and
sinks'.

- p.38, Figures 7 and 8: These plots would be clearer if the black and purple parts
were presented on different y-axis. Currently, it is quite difficult to observe differences
between different scenarios. Also, in the caption of Figure 7, 'it's' should be replaced
with 'its'.

---

## Short Comment (SC1) · 5 Jan 2017

Date: January 5, 2017

Dear Reviewer #1,

We appreciate the very constructive and thorough feedback within your review. Although we plan to wait until the end of the discussion period before we fully address your comments, we did want to respond to your primary concern, that the reclassification (or 'repartitioning') of refractory and labile particulate organic matter (POM) upon resuspension may affect the estimates of seabed oxygen consumption. Here, we elab-

orate on the rationale for this repartitioning, and show results from a model without this repartitioning.

As noted in the review, the model converts a fraction of resuspended refractory POM into labile POM upon entrainment from the seabed into the water column. This modeling approach is supported by laboratory experiments by Stahlberg et al. (2006) indicating that organic matter remineralization rates increased during and in the days following resuspension events, and that changes in remineralization rates were not only due to changes in oxygen availability. Literature pertaining to how resuspension affects the remineralization of particulate organic matter over days to weeks is limited, but we also considered related studies that focused on redox oscillations and remineralization (e.g. Gilbert et al., 2016; Sun et al., 2003; Caradec et al., 2004; Aller, 1994; Wakeham and Canuel, 2006; Arzayus and Canuel, 2004). Because guidance from this literature is inconclusive, we chose a parsimonious approach, i.e. 'repartitioning', for the partitioning of organic matter that mimics the changes in remineralization described in Stahlberg et al. (2006), and is consistent with field data from the Rhone River (see below).

Like the reviewer, we have admittedly been concerned about the model's sensitivity to this approach. With prompting from this review, we ran a "no-repartitioning" model run that was identical to the standard model run in the paper, except it did not repartition refractory and labile organic matter upon resuspension to the water column. Instead, any labile organic matter in the model was assumed to stay labile, and refractory organic matter in the model was assumed to stay refractory.

Overall, results from this no-repartitioning model run indicate that estimates of seabed oxygen consumption were sensitive to this repartitioning (see next paragraph), but estimates of water column oxygen consumption were not. In both the standard and the new "no-repartitioning" model runs, resuspension approximately doubled oxygen consumption in the bottom boundary layer when averaged over a two-month time period (see Figure R1 in supplement, please compare to Figure 4 in the submitted manuscript, reproduced here as Figure R2 in supplement). Since water column oxygen consumption is the dominant component of the total oxygen consumption, our overall results were insensitive to this parameterization.

However, as noted above, seabed oxygen consumption was sensitive to the repartitioning of organic matter. Compared to the standard model where resuspension increased seabed oxygen consumption by +20% (Fig. R2 in supplement), resuspension had a negligible effect on seabed oxygen consumption in the "no-repartitioning" model run over timescales of two-months (Fig. R1 in supplement). Note that this result from the "no-repartitioning" model is in conflict with observations from Toussaint et al. (2104; their Fig. 7), which do not show a decrease in seabed oxygen consumption following resuspension events like that seen in Figure R1 (see supplement). This implies that the model with repartitioning better describes the observations on the Rhone prodelta compared to the model without repartitioning.

We look forward to other input via the review process, after which a more complete response to reviews will be composed, including how the revision will incorporate these interesting new results. Thank you again for your review.

Best Regards,

Julia Moriarty

References:

Aller, R. C. (1994). Bioturbation and remineralization of sedimentary organic matter : effects of redox oscillation. Chemical Geology, 114, p. 331-345.

Arzayus, K. M., Canuel, E. A. (2004). Organic matter degradation in sediments of the York River estuary: Effects of biological vs. physical mixing. Geochimica et Cosmochimica Acta, 69 (2), p. 455-463.

Caradec, S., Grossi, V., Gilbert, F., Guigue, C., Goutx, M. (2004). Influence of various redox conditions on the degradation of microalgal triacylglycerols and fatty acids in marine sediments. Organic Geochemistry, 35, p. 277-287.

Gilbert, F., Hulth, S., Grossi, V., and Aller, R. C. (2016). Redox oscillation and benthic nitrogen mineralization within burrowed sediments: An experimental simulation at low frequency. Journal of Experimental Marine Biology and Ecology, 482, p. 75-84.

Stahlberg, C., Bastviken, D., Svensson, B. H., Rahm, L. (2006). Mineralisation of organic matter in coastal sediments at different frequency and duration of resuspension. Estuarine, Coastal and Shelf Science, 70, p. 317-325.

Sun, M.-Y., Aller, R. C., Lee, C., Wakeham, S. G. (2003). Effects of oxygen and redox oscillation on degradation of cell-associated lipids in surficial marine sediments. Geochimica et Cosmochimica Acta, 66, p. 2003-2012.

Toussaint, F., Rabouille, C. Cathalot, C., Bombled, B., Abchiche, A. Aouji, O., Buchholtz, G., Clemencon, A., Geyskens, N., Repecaud, M., Pairaud, I., Verney, R., Tisnerat-Laborde, N. (2014). A new device to follow temporal variations of oxygen demand in deltaic sediments: the LSCE benthic stations. Limnology and Oceanography: Methods, 12, p. 729-741.

Wakeham, S. G., Canuel, E. A. (2006). Degradation and preservation of organic matter in marine sediments. In: Handbook of Environmental Chemistry, 2. Published online on 2 December 2005.

Please also note the supplement to this comment:
http://www.biogeosciences-discuss.net/bg-2016-482/bg-2016-482-SC1-supplement.pdf

[Figure]

**Supplement:**

**Supplement for Short Comment Posted January 5, 2017 by J. M. Moriarty**

***Figure R1***: Time series of bed stress, and oxygen consumption in the seabed and bottom boundary layer for both the "no-repartitioning" (blue solid line) and "no-repartitioning, no-resuspension" model runs (pink dashed line). Shading indicates resuspension time periods, including 6–11 April, 23 April–3 May, and 18–25 May 2012. The black dashed lines indicate the times at which profiles in Fig. 5 were estimated.

[Figure]

***Figure R2***: Figure 4 from the submitted paper. Time series of bed stress, and oxygen consumption in the seabed and bottom boundary layer for both the standard (blue solid line) and no-resuspension model runs (pink dashed line). Shading indicates resuspension time periods, including 6–11 April, 23 April–3 May, and 18–25 May 2012. The black dashed lines indicate the times at which profiles in Fig. 5 were estimated.

[Figure]

---

## Referee Comment (RC2) · Anonymous Referee #2 · 17 Jan 2017

This is a very interesting paper on the effect of physical-induced sediment resuspension on the oxygen dynamics in the sediment and overlying water column in a shallow mediterranean area. As far as I am aware, this is the first time that such sediment dynamics are included in this detail into a mechanistic model. As clearly shown in this paper, such resuspension events may significantly alter sediment and bottom layer oxygen dynamics. Moreover, the correspondence of model and data suggest that resuspension in the area is very well represented by this model. The manuscript is very well written, and results are clearly explained.

[Figure]

I tend to have somewhat different views on why the model displays what it does, which might be considered. Also I would suggest to slightly rewrite the model equations and setup.

*Model results*

The sequence of events that are invoked to explain the differences in oxygen budget are strongly focused on the physics. The authors write that resuspension increases the vertical gradient of oxygen in the sediment, which in turn increases the diffusive flux, increasing oxygen consumption through nitrification. A biogeochemical view would be that resuspension brings ammonium from deep layers more towards the surface, in close contact with oxygen. This would increase nitrification and increase oxygen consumption, resulting in stronger vertical gradients and a higher flux. Probably the truth is in between both ?

One of the sediment characteristics that has a significant effect on resuspension-induced O2 dynamics in the sediment, is the abundance of labile organic material. This is quite surprising, considering that the increase in O2 consumption after resuspension is largely due to nitrification. Based on the nitrification effect, I would guess that high ammonium concentrations at depth would increase the effect of resuspension of O2 dynamics. And deep concentrations of ammonium are usually linked to deposition of refractory OM rather than of reactive organic matter. I do not understand what is causing this effect of labile OM.

I guess that the 20% increase in oxygen consumption of the seabed in case of resuspension results from the fact that the initial state of the diagenetic model has been estimated in the absence of resuspension events (although I am not sure this is how the model was initialized). Perhaps, for sediments where these resuspension events occur regularly, a better initial profile would be generated as a dynamic equilibrium that is established including these resuspension events? In this case, the average oxygen consumption would not be higher, but the variability would be increased due to

resuspension.

*Model description:*

Several ways to describe the benthic model are quite confusing, not standard and sometimes inconsistent.

The formula for Eised is confusing as Eised is described differently in equation (1) (page 5 line 13-14) compared to its description in table 1. It is not standard as it features the delta t (timestep). Timesteps only determine how the model is solved, and should not feature in model equations. I suggest to give the equation for the erosion *rate* instead of the eroded mass, as is usual in sediment modelling studies. This removes the delta t in the equation, and changes the units of Eised.

A related question concerns the sedimentation rate. Does this only apply when there is no erosion?

In diagenetic models, the units of dissolved substances are typically expressed in mmol / m3 *liquid*, and porosity features in the diagenetic equation because diffusion takes place in the porewater, while the mass balance needs to be written for bulk sediment. In this model, the units of O2, and NO3, (and of the parameters kO2 and kNO3, etc. . .) are said to be mmol / m2. But what is this: mmol/m2 bulk sediment or mmol/m2 liquid? (my guess is that it is per m2 bulk, but I am not sure). Moreover, the units are not always consistent. For instance, the dO2/dz is said to be in units of mmol O2/m4 which suggests that O2 is in mmol/m3. Finally, the units of the monod constants are said to be mmol/m2 in table 1, but in mmol/m3 in table 3. I suggest to represent the equations using concentrations per m3 liquid as is custom in diagenetic modelling. How this is actually implemented in the model is less relevant.

Perhaps related to the previous comment: the equation for the diffusion across the sediment-water interface is very strange (in the supplement). I suspect -but could be wrong- that this is necessary to assure mass conservation and correct for using the

wrong units (i.e. if units are per bulk sediment and not per liquid)?. Also, I could not see that this is how Soetaert et al. implemented sediment-water exchange in their model.

It only becomes clear how resuspension is effectively included in this model based on the supplement. As this is the truly distinguishing feature, it should be included in the main paper. Related to that: how is this model integrated-i.e. which integration method is used? Many integration methods assume smooth dynamics, and one cannot just alter state variables directly.

There is no need for figures 7, 8 and 9.

––––––––––––––––––––––––––––––––

---

## Author Comment (AC1) · 15 Feb 2017

**General comments**
This manuscript presents the development of the HydroBioSed module within the ROMS-CSTMS framework. This module couples existing hydrodynamic, sediment transport, and water-column biogeochemical modules with a seabed biogeochemistry module, with the aim of investigating the role of resuspension on biogeochemical dynamics, most notably oxygen. This novel module is then applied to the Rhone Delta, showing that local resuspension has an important impact on oxygen dynamics on at least seasonal timescales. The paper also classifies systems where the impact of intermittent local resuspension on oxygen dynamics may be substantial.

I believe that this work is a necessary addition to benthic-pelagic coupling of reactive transport models, and an advancement in the field. The manuscript is well written and clearly organized, actually even too organized to my liking, by announcing even every subsection in detail. I do however believe that sometimes a little more detail might be nice in the method section, such that the reader unfamiliar with ROMS and/or CSTMS can also well understand the model development. Specifically, I would add the supplement to the main paper, as it contains details that are necessary to fully understand the method development, which is a major aim of this paper. Moreover, I only became aware of my biggest concern with the paper after having read the supplement. This major concern is related to the way the model deals with the lability of resuspended organic matter. I do not believe that it is valid to reclassify refractory resuspended organic matter into the labile pools, and believe it is the main reason why the re-deposited organic matter is enriched in labile pools (p.10, l.31-32), which I consider an unexpected and also unrealistic result that may highly influence the estimated seabed oxygen consumption. I will explain this concern in more detail below, but I believe this issue needs to be resolved before the manuscript can be published. Aside from that, and a few other comments related to the methodology, my comments are mainly technical and deal with how results are presented.

Dear Reviewer #1,

Thank you for your supportive and constructive feedback, which we believe has improved the paper. The most substantial changes to the manuscript based on your review are summarized here:

1. **Attention to the re-partitioning of particulate organic matter:** We added a new sensitivity test that did not re-partition the organic matter in the water column. Results from this sensitivity test showed that even when resuspended organic matter was not repartitioned in the water column,

oxygen consumption still increased due to resuspension events over timescales of days to two months (see item 5 below).  Additionally, we improved the justification for our approach.
2.  **Clarification of the model equations and methods**: We moved the supplement into the main manuscript, and more attention was given to the presentation of equations and variables.

For further details, please see our response below.  All page and line numbers refer to the original submitted manuscript.

Thank you again for your review.

Best Regards,

Julia Moriarty, Courtney Harris, Christophe Rabouille, Katja Fennel, Marjorie Friedrichs, and Kevin Xu

**Specific comments**
1.  p.4, l.9-10: is there a way that these resuspension events can be predicted, e.g. from tides or wind speeds? I am asking this in the context of potential model applications.
    It is possible to predict resuspension events based on wave and current shear stresses, but simple parameterizations often have a large amount of uncertainty. We have added text to Section 4.2, line 22 to suggest that such parameterizations predicting erosion and deposition could be combined with biogeochemical parameterizations for seabed-water column fluxes to estimate the effect of resuspension on oxygen dynamics in larger-scale numerical models in future research efforts.

2.  p.4, l.21-22: What is the estimated seabed oxygen consumption at the study site itself?
    We have added text to Section 1.1 (pg. 4, lines 21-22) to clarify this.

3.  p.5, l.15: when looking at Table 1, it becomes evident that this equation calculates an amount of sediment in areal units, i.e. in kg/m2. Do I have to see this as a depth-integrated concentration from the active transport layer (so a concentration multiplied by za), or rather as sort of a flux across the sediment-water interface? What does the erosion rate parameter M represent exactly?
    This equation estimates a flux of sediment across the seabed-water interface. This is now clarified in Section 2.1.1 (pg. 5, lines 13-14).

    The erosion rate parameter, M, is a rate constant representing how much sediment is entrained from the seabed into the water column per unit time as bed stress increases, i.e. the erodibility of the seabed. We now note this in Table

1, and repeat the reference to Warner et al., (2008) in Section 2.1.1 (page 5, line 14) for further reference.

4. p.5, l.18: In terms of mass, how much of POM is added relative to the number of inert particulates? I am asking this to see if adding POM as an additional particle source can lead to inconsistencies in e.g. estimated deposition rate compared to previous applications. Keeping this in mind, wouldn't it make more sense conceptually if the inert particulates pool would be split into a POM pool and an inert pool?

   In this location POM only accounts for 3% of the combined inorganic + organic particulates in the seabed by weight (Pastor et al., 2011a). Because 3% is small compared to the uncertainty in estimating sedimentation rates and the uncertainty introduced by imposing a long-term sedimentation rate on a two-month time period, we chose to develop the model such that POM deposition does not contribute to the thickness of eroded or deposited seabed layers. Thus, the instantaneous inorganic sedimentation rate equals the rate at which seabed layers are shifted downwards (or upwards, during erosional periods). Also, our inorganic sedimentation rate of 10 cm $y^{-1}$ (or 14 kg $m^{-2}$ $yr^{-1}$) is equal to the total sedimentation rate used in other models of this site (i.e. Pastor et al., 2011a). In response to this question, we have clarified that we neglect POM's contribution to the total sedimentation rate in Section 2.1.1, p. 5, line 18.

   Note that neglecting POM's contribution to the thickness of seabed layers is also justified by sensitivity tests that indicated that even a 50% change in total sedimentation rate caused relatively small variations in seabed and bottom water oxygen dynamics (Figures 7, 8). We therefore expect a 3% change to be relatively unimportant.

5. p.5, l.22: The information of S.1 needs to be added to this section, especially since it is so important for the conceptual model. My main problem with the conceptual model is that the repartitioning of refractory resuspended organic matter into the labile pools does not seem a valid approach. I understand that resuspension may change the degree of aggregation, but I do not see why it would change the lability per se. The arguments given on p.2, l.5-8 of the supplement may be true, but can be implemented into the model in a conceptually better way. Faster OM remineralization under oxic conditions is already implemented in the model via the limitation factors and half-saturation constants. Higher remineralization rates in the water column can also be achieved by using different rate constants in the water column compared to the sediment, but in reality, I think this is due to the constant supply of fresh OM to the water column due to in-situ production and/or external loading. The former is already part of the model, whereas the latter might be implemented by an additional source term to the water column.

   First, thank you for this, and other related, comments, which prompted us to include an additional sensitivity test and the inclusion of additional supporting

information from the literature.  Specific responses and changes to the manuscript are listed below:

(1) We added the information from Supplement S.1 to Sections 2.1.2 and 2.1.3 (pages 5-7).

(2) We significantly improved the justification for our approach in the Methods (in Section 2.1.3, pg. 7, near lines 5-21). Because it was a major concern for the reviewer, we also justify our approach here. As the reviewer notes, the model converts a fraction of resuspended refractory POM into labile POM upon entrainment from the seabed into the water column. This modeling approach is supported by laboratory experiments by Stahlberg et al. (2006) indicating that organic matter remineralization rates increased during and in the days following resuspension events, and that changes in remineralization rates were not only due to changes in oxygen availability. Literature pertaining to how resuspension affects the remineralization of particulate organic matter over days to weeks is limited, so we also considered related studies that focused on redox oscillations and remineralization (e.g. Gilbert et al., 2016; Sun et al., 2003; Caradec et al., 2004; Aller, 1994; Wakeham and Canuel, 2006; Arzayus and Canuel, 2004). Because guidance from this literature is inconclusive, we chose "repartitioning" for the organic matter that mimics the changes in remineralization described in Stahlberg et al. (2006), and is consistent with field data from the Rhone River (see bullet (3)).

(3) We added a 'no-repartitioning' sensitivity test that was the same as the standard model, but did not re-partition the resuspended POM in the water column. Instead, any labile organic matter in the model was assumed to stay labile, and refractory organic matter in the model was assumed to stay refractory.

Overall, results from this no-repartitioning model run indicate that estimates of *seabed oxygen consumption* were sensitive to this repartitioning, but estimates of *water column oxygen consumption* were not. **Since water column oxygen consumption is the dominant component of the total oxygen consumption, our overall results were insensitive to the re-partitioning parameterization.** However, as noted above, *seabed oxygen consumption* was sensitive to the repartitioning of organic matter. Compared to the standard model where resuspension increased seabed oxygen consumption by +20%, resuspension had a negligible effect on seabed oxygen consumption in the "no-repartitioning" model run over timescales of two-months.

However, note that results from the "no-repartitioning" model conflict with observations from Toussaint et al. (2104; their Fig. 7), who observed no significant change in seabed oxygen consumption (i.e. no reduction of diffusive oxygen fluxes into the seabed) following erosional periods like that estimated by the 'no-repartitioning' sensitivity test. This implies that the model with

repartitioning better describes the observations on the Rhone prodelta compared to the model without repartitioning.

Results from the no-repartitioning sensitivity test are presented in Section 3.3.1, Figures 7 & 8, and are discussed in Section 4.2 (first paragraph) and Section 4.3 (p. 17, lines 26-28 & p. 18, line 2).

6. p.6, l.1: to this section, the information of S.2 should be added. I was really missing this while reading this section, it is too important for the model description to be part of the supplement.
We agree that this information should be in the main manuscript, as opposed to a separate document. We have therefore moved this material from the Supplement to an Appendix (located below the Conclusions). We think this information is better suited for the Appendix, rather than the Methods section, because although this technical information will be informative to numerical modelers, it may not be as interesting or relevant to other readers.

7. p.6, l.17-21: I believe that these equations are currently incorrect, when following the units of Table 1. The right-hand side of Eqs. 2-6, is in mmol/m3/d, whereas the left-hand side is in mmol/m2/d. I would anyway stick with mmol/m3 and mmol/m3/d everywhere throughout the manuscript, even though the model stores concentrations in a depth-integrated way, as this is the conventional way to present rates and concentrations.
Thank you very much for pointing this out. We have changed all units to mmol $m^{-3}$ $d^{-1}$ throughout the paper, including on p. 6, lines 17-21.

8. p.7, l.1-2: My first thought when reading this was: "shouldn't resuspension and redeposition have a huge impact on porosity in the surficial centimetre of the seabed"? As far as I know, porosity can fluctuate a lot even in the top centimetre depending on the amount of deposition and/or resuspension. I wonder I the authors have tested the effect of a different porosity, even if kept constant with depth; perhaps this should be part of the sensitivity analysis.
Yes, we agree that porosity can fluctuate in surficial seabed sediments. Given that we expect the most significant effect of changes in porosity to be changes in diffusion, and that the manuscript already includes sensitivity tests to diffusion within the seabed, however, we suggest that this sensitivity test should be included in a future paper that more thoroughly investigates how seabed and bottom water dynamics are affected by different combinations of environmental conditions.

9. p.7, l.8-12: I might have missed this, but are sediment deposition and erosion mutually exclusive? I understand that erosion only occurs in case of high bed stress, but does this mean that there is no deposition at all during erosion, or that both processes co-occur but that erosion dominates?
Sediment fluxes across the seabed-water interface are assumed to equal the difference between deposition and erosion as estimated for each time-step, so

they are not mutually exclusive and may co-occur. This is now noted in the text, and we have added the equation for sediment deposition (Section 2.1.1, pg 5, line 15).

10. p.7, l.30: What are the units of the 5x6 model grid?
Units are 'grid cells'. This is now noted in the text (Section 2.1, pg. 7, line 30).

11. p.9, l.4-6: It would be good to stress here that the bottom boundary layer is the layer below the pycnocline, as mentioned on p.7, l.32. Otherwise, this definition lacks a bit of context (i.e. why was 4 m chosen as thickness of the BBL?)
We have edited the text per the reviewer's suggestion (p. 9, lines 4-6). To reduce confusion about our definition of the bottom boundary layer (BBL) versus other definitions, we have also revised the entire manuscript to refer to this region as "bottom water", rather than the bottom boundary layer.

12. p.9, l.7-9: Why are the rates of biogeochemical processes in the BBL not added to the code as output parameters?
We have revised the model code so that this information is saved. This has simplified the manuscript and affected text in Section 2.2 (p. 9, lines 7-9), and slightly changed the model estimates given in the Results.

13. p.9, l.28: I It seems that, since zSWI and O2,OPD are by definition 0, O2,SWI is positive, and zOPD is negative (at least according to Figs. 3 and 5), Eq.8 would by default result in a negative value, which is not the case. So this needs to be changed to make it consistent.
Thank you for noticing this discrepancy. We have added a factor of (-1) to equation (8) (p. 9, l. 28) so that increases in $dO_2/dz_{OPD}$ correlate with sharper oxygen gradients, and have clarified the definition of $z_{OPD}$ in Table 1 (pg. 28, last line).

14. p.10, l.5-7: Have the authors considered implementing a loss term of the resuspended material, as a possible workaround to the uniform conditions in the horizontal?
We also believe the fate of eroded sediment may be important to quantify to better understand seabed & BBL biogeochemical dynamics. However, for this paper we wanted to focus on the vertical exchange processes. Instead, we believe the issue of resuspended sediment that is transferred in to, or out of, the local area is better addressed by implementing the model in three dimensions. We are planning to implement the model for the Gulf of Mexico, and expect to address this issue in that publication.

15. p.10, l.26-27: How were the erosion depths quantified? Is erosion depth more dependent on the duration of the event, or on the bed stress?
We have now clarified how erosion depths were calculated in the Methods (Section 2.2, pg. 9, line 12). Depths of erosion depend on both bed stress and duration of event.,.

16. p.10, l.31-32: This is a result I do not understand, and also do not believe to be an accurate representation of reality. I believe this is due to the parameterisation of the lability of the resuspended material, which I discussed above.
Please see bullet (5) of this response to reviews.

17. p.11, l.22-27: I'm not sure if I understand this argument. Isn't this faster oxygen consumption during re-deposition driven by the newly-deposited OM being more labile, rather than additional oxygen availability?
Yes, we agree that the rapid consumption of oxygen during depositional periods occurs due to (1) re-deposition of POM, and the conversion of some POM from refractory to labile, as the reviewer suggests, as well as (2) resuspension-induced rates of nitrification (see Figure 6). During erosional periods, process (2) dominates oxygen consumption, and therefore the maintenance of 'erosional oxygen profiles'. Both (2) and (1) assist the destruction of 'depositional oxygen profiles'. We have edited the text to clarify that both processes were important.

18. p.11, l.26: Wouldn't 'quasi-steady state' be a better term there than 'equilibrium'?
This has been changed here (p. 11, line 26), as well as on p. 9, line 22, where it is also used.

19. p.12, l.1-2: Why would entrainment of nitrate increase the nitrification rate? Eq.5 does not show any NO3-limitation.
Thank you for catching this mistake; we have removed this.

20. p.12, l.10-12: This sentence seems to lack something. I was expecting to read "..compared to xxx" at the end.
We agree, and have rephrased this sentence.

21. p.12, l.22-25: Fluxes of NH4 and ODU are presented here, it would be nice to also present them in a figure or table (similar to Table 6).
While we agree that this is interesting information for some researchers, the focus of this paper is on oxygen dynamics and introducing the coupled model. As the paper is already quite long, and the proposed table/figure does not directly support the main conclusions, we respectfully believe that including such a table/figure is beyond the scope of our paper.

22. p.14, l.14: all 3 resuspension events presented here have a duration of days (grey shaded areas of Fig 4). I don't see any events with a duration of hours. Were the shorter events not captured by the model, or did they not take place during this two-month period?
Yes, we agree that this is confusing, and have clarified the text (Section 4.1,p. 14, l. 14) to state that we mean short time periods, from hours-to-days, not just the entire resuspension event as a whole, which does occur over a period of days.

23. p.15, l.15: The Almroth-Rosell et al. (2011) paper shows most resuspension at other locations in the Baltic Sea, though it is high in certain parts of the Gulf of Finland. Perhaps the authors can elaborate a bit on this in their discussion.
We thank the reviewer for directing us to this paper. The text on p. 15, lines 12-15 focuses on the criteria needed for **local** resuspension, i.e. when material is resuspended and then re-deposited, to affect oxygen dynamics in various areas. In contrast, the modeling study in Almroth-Rosell et al. (2011) primarily focused on the role of redistribution of resuspended POM, and its affect on oxygen dynamics, and not the role of local resuspension. This reference is therefore especially pertinent to our future work section, and we have added it to Section 4.3 (p. 18, lines 10-12). We also replaced 'resuspension' with 'local resuspension' to make the text clearer in Section 4.1.1 (p. 15, lines 9-17), and other places within the manuscript.

24. p.15, l.28-29: I am perfectly fine with keeping temperature constant in the model, but I do think that the impact of a higher temperature should be tested by increasing rate constants according to the Q10 rule. That would be a very valuable sensitivity analysis.
We agree that this analysis would be valuable, but believe it should be incorporated into a future paper that more thoroughly investigates how seabed and bottom water dynamics are affected by different combinations of environmental conditions. Here, especially because the paper is already quite long, we have chosen to focus specifically on a single time period at a single site, i.e. the Rhone subaqueous delta in Spring 2012.

25. p.16, l.3-5: Wouldn't OM loading also be lower in winter and thereby impacting oxygen consumption?
We agree that changes in organic matter loading can affect oxygen consumption, and we have added a paragraph about organic matter loading to Section 4.1.2 (p. 16, after line 7).

26. p.16, l.13-16: What is the spatial scale of these other models as compared to this model? Could differences therein be an argument for their different handling of oxygen consumption due to resuspension?
We do not believe that the spatial scale of these regional models should affect the scientific results, but acknowledge that inclusion of a full sediment model in larger-scale water quality models may be inefficient. This is now noted in Section 4.2, pg. 16, lines 19-22.

27. p.30, Table 3: How has this sedimentation rate been estimated? If e.g. by 210Pb dating, wouldn't it be more correct to say that this is the total sedimentation rate, that then needs to be partitioned into an inorganic and an organic part? (see also comment on p.5, l.18)
We estimated the sedimentation rate based on Pastor et al. (2011a), as noted in Table 3. Pastor et al. (2011a) based their estimate on the radioisotope observations of Zuo et al. (1997, 1998), Radakovitch et al. (1999) and Miralles et

al. (2005).  As you note, these observations provide indications of the total sedimentation rate, which introduces about a 3% error in our estimate of 10 cm/yr.  As noted above, in the response to comment #4, however, (1) this error is small compared to other sources of uncertainty, and (2) sensitivity tests indicate that even a 50% error in the sedimentation rate does not produce large changes in the results.

28. p.31, Table 3: What does the fraction of ODUs that are solid and inert represent? Since denitrification is modelled separately, does N2 also belong to the ODU pool here? If not, this number may be quite high.
This fraction represents pyrite and other materials that are assumed to be oxidized sufficiently slowly such that they do not affect model estimates over the timescale of our study (2 months), as noted in Soetaert et al. (1996a,b), which is cited by this paper.  It does not include $N_2$.  In our study, we assumed that 99.5% of ODUs were solid and inert.  This value, 99.5%, is on the high end of the range of values in the scientific literature, but has already been justified for this study site by Pastor et al. (2011; see Section 4.3.1), as noted in our Table 3.

29. p.33, Figure 2: I do not find this figure very clear. What I would be interested to know is how suspended sediment, organic aggregates and organic particulates from the water column biogeochemical module are interlinked. I.e. a schematic of supplement S.1, showing that e.g. labile water-column organic matter is the sum of phytoplankton, detritus and labile aggregates, and that this is part of the total suspended matter.
Thank you for this suggestion.  We have added schematics showing how particulate organic matter is partitioned in the model during resuspension and deposition for the standard model run and no-repartitioning sensitivity test.

**Technical corrections**
30. A general comment: I get the impression that the terms 'resuspension event' and 'erosional period' seem to be used interchangeably (e.g. in the caption of Fig. 5), is this correct?
We have clarified the definition of resuspension (event) by adding it to Table 2A, and have confirmed that we now use it consistently in the manuscript. Note that although the verb *resuspend* and adjective *resuspended* are synonymous with erode and eroded, the noun *resuspension (event)* refers to the entire cycle of erosion and deposition.

31. p.3, l.24-26: I would leave out the "other factors" from the research question, it sounds too vague and results of other factors are not presented (only mentioned to be negligible in the case of ODU oxidation rate and the diffusion scheme).
We have deleted this phrase from p. 3, line 26, as suggested.

32. p.3, l.28: Section 1.1 fits better as the first section of the Methods.
We have reorganized the paper, as suggested.

33. p.3, l.32: 'located at _25m water depth'
   We have changed 'in' to 'at' as suggested.

34. p.3, l.33: I would prefer 'model validation' over 'model evaluation' here
   This sentence refers to the use of data from both (1) Pastor et al. (2011a), which was used to validate the model (i.e. to confirm that the model was implemented correctly based on its ability to reproduce model estimates from Pastor et al. (2011)'s previously implemented Soetaert model), as well as data from (2) Toussaint et al. (2014), whose data was used to evaluate the model (i.e. to compare our model estimates to observations).  To clarify this sentence, we therefore replaced "model input and evaluation" with "model input, validation and evaluation" (p. 3, line 33).

35. p.8, l.28: the ODU oxidation rate sensitivity test is mentioned here as not being presented further, but it is part of Table 4. It should be removed from the table.
   We agree, and have removed the last two rows from Table 4 (pg. 32).

36. p.9, l.10: 'it's' should be 'its'
   Thank you, this was changed.

37. p.9, l.29: Table 5 does not show this; remove the reference here.
   Thank you for this correction, we have removed this reference to Table 5, and added it to the correct location (p. 9, l. 26).

38. p.12, l.21: The contribution of ODUs is virtually 0, as Table 6 shows, and negligible compared to the other processes, so remove it here from the text.
   Thank you for this suggestion.  We have removed the text describing the contribution of ODUs (p. 12, lines 19-29).

39. p.16, l.10: change 'produce' to 'reproduce'
   We have changed this.

40. p.32, Table 6: The total oxygen consumption is higher than the sum of the separate processes. Maybe this is because minimum and maximum values are presented, but also for the average, this is not the case.
   Yes, we do not expect the maximum seabed oxygen consumption and the maximum bottom water (i.e. BBL) oxygen consumption to equal the maximum seabed+BBL oxygen consumption.  This is because the maxima of the seabed and BBL oxygen consumption may not coincide with each other.  The same logic applies for the minima of oxygen consumption.

   For the averaged values and value on June 1, the sum of the total seabed oxygen consumption and total BBL oxygen consumption does equal the total seabed+BBL oxygen consumption, except that these values are sometimes off by one unit due to rounding error when using two significant digits.

41. p.14, l.30: "reduced species for oxidation". This is not shown by the model, as the contribution of ODUs to oxygen consumption in the BBL is really minor. Remove this statement.
We have removed this statement (p. 14, line 30).

42. p.15, l.25-26: Too many references here.
We reduced the references to only the most essential (from 5 to 2 references; p. 15, l. 25-26).

43. p.16, l.19-23: More appropriate than what?
We have clarified this sentence so that it focuses on which types of parameterizations may be most applicable for the Rhone delta and similar environments (i.e. compared to other types of parameterizations).

44. p.17, l.12-17: This section might better fit in Section 4.1.2, when discussing temporal variations
We agree, and moved this (Section 4.3, p. 17, lines 19-23) to a new paragraph at the end of Section 4.1.2. As a result, we combined the first and fourth paragraphs of Section 4.3 to suggest that future work could include applying the model (in 1D or 3D) for different environments (Section 4.3, p. 17-18).

45. p.17, l.32-33: Something seems to be missing in this sentence.
This was a typo. We deleted the word 'may'.

46. p.28/29, Table 1: carefully check the units, especially in relation to Eq. 2-5.
Thank you, and we have done so.

47. p.32, Table 6: Add vertical lines between the three different categories as specified on the upper row. Also, why aren't averages presented of the depositional (white parts of Figs. 4 and 6) and erosional (grey parts of Figs. 4 and 6) periods presented, rather than only an average of the total two-month period? That seems clearer and more meaningful to me. Finally, clearly specify whether the maximum and (especially) minimum values presented are from the whole two-month period or from the erosional parts only.
We have added the vertical lines to Table 6.

We added model averages for depositional and erosional periods to Table 6. However, note that the grey shading indicates resuspension events, which includes the cycle of erosion and re-deposition, not just the erosional time period.  This has been clarified in Table 2, and in the captions of Figures 4 and 6.

We have clarified the caption of Table 6 so that it indicates that the maximum and minimum values were calculated based on the entire model run.

48. p.34, Figure 3: Define SSC in the figure caption. Add the dates to the oxygen microprofiles.

We have defined SSC in the caption, and added the dates to the caption (p. 34, Fig. 3). Adding the dates to the profiles made the figure look too crowded.

49. p.35, Figure 4: Add a horizontal line to panel a showing the critical shear stress. Panel b could use a differently scaled y-axis, to make temporal variations clearer.
We have made the suggested changes (p. 35, fig. 4).

50. p.36, Figure 5: Why are different times chosen here than in Figure 3? For clarity and consistency, it would be nicer to study the same erosional event in both figures. The depth scale of the O2 profile may be changed for clarity. Also for clarity, the captions 'pre-resuspension', 'during resuspension' and 'post-resuspension' (or something equivalent) could be added to the three columns.
We have changed Fig. 5 so that it plots profiles for the same event that is the focus of Figure 3. We have also changed the depth scale for the O2 profiles and added the column-headers (p. 36, Fig. 5).

51. p.37, Figure 6: I don't see any oxygen sources in panel b, only sinks. So I would change the caption here. Also, I would replace 'fluxes' in the caption with 'sources and sinks'.
We have made the suggested changes to Fig. 6 and the caption (p. 37, lines 2-5).

52. p.38, Figures 7 and 8: These plots would be clearer if the black and purple parts were presented on different y-axis. Currently, it is quite difficult to observe differences between different scenarios. Also, in the caption of Figure 7, 'it's' should be replaced with 'its'.
To make the scenarios easier to compare, we made the lines and dots thinner. We also combined the two figures into one figure so that the panels could be taller, and thus, easier to read. We did not divide these 4 panels (2 in the original Figure 7 & 2 in the original Figure 8) into 8 panels because we tried to balance your comments with suggestions from the other reviewer, who suggested we remove a few figures, including Figs. 7 & 8.

Also, one of the main points of the Figures 7 & 8 is that changing the parameters in the model generally does not have a large effect on the results, suggesting that our results are relatively robust. We now cite Figures 7 & 8 when we make this point (Section 3.3., p. 13, lines 31-32; p. 14, line 1).

The apostrophe from "it's" was removed (pg. 38, line 3).

---

## Author Comment (AC2) · 15 Feb 2017

**General comments**
This is a very interesting paper on the effect of physical-induced sediment resuspension on the oxygen dynamics in the sediment and overlying water column in a shallow mediterranean area. As far as I am aware, this is the first time that such sediment dynamics are included in this detail into a mechanistic model. As clearly shown in this paper, such resuspension events may significantly alter sediment and bottom layer oxygen dynamics. Moreover, the correspondence of model and data suggest that resuspension in the area is very well represented by this model. The manuscript is very well written, and results are clearly explained.

I tend to have somewhat different views on why the model displays what it does, which might be considered. Also I would suggest to slightly rewrite the model equations and setup.

Dear Reviewer #2,

Thank you for your supportive and constructive feedback, which has helped us improve the manuscript. As a result of this review, we propose changing the manuscript by including a more refined analysis, removing Figures 9a and 9b, and emphasizing resuspension's effect on ammonium, and how this affects seabed oxygen consumption. Additionally, the model equations and methods were clarified based on the reviewer's comments.

For further details, please see our response below. All page and line numbers refer to the original submitted manuscript.

Thank you again for the review.

Best Regards,

Julia Moriarty, Courtney Harris, Christophe Rabouille, Katja Fennel, Marjorie Friedrichs, and Kevin Xu

**Specific comments**
*Model results*
1.  The sequence of events that are invoked to explain the differences in oxygen budget are strongly focused on the physics. The authors write that resuspension increases the vertical gradient of oxygen in the sediment, which in turn increases the diffusive flux, increasing oxygen consumption through nitrification. A biogeochemical view would be that resuspension brings ammonium from deep layers more towards the surface, in close contact with oxygen. This would

increase nitrification and increase oxygen consumption, resulting in stronger vertical gradients and a higher flux. Probably the truth is in between both ?

Thank you for this suggestion. We agree that resuspension-induced increases in ammonium concentrations and the related increase in nitrification is important for maintaining the 'erosional oxygen profiles' and increasing seabed oxygen consumption during erosional periods. We have added text to emphasize this (Section 3.2, p. 10, line 29; Section 3.2.1, p. 11 lines 27-28 & p. 12, lines 1-4).

2. One of the sediment characteristics that has a significant effect on resuspension-induced O2 dynamics in the sediment, is the abundance of labile organic material.

This is quite surprising, considering that the increase in O2 consumption after resuspension is largely due to nitrification. Based on the nitrification effect, I would guess that high ammonium concentrations at depth would increase the effect of resuspension of O2 dynamics. And deep concentrations of ammonium are usually linked to deposition of refractory OM rather than of reactive organic matter. I do not understand what is causing this effect of labile OM.

Yes, we agree that this was somewhat confusing. We have clarified why this occurs in the manuscript (Section 3.3.1, p. 13, lines 18-19), and note it below.

Please recall that resuspension increases the amount of labile organic matter on the seabed due to the partitioning of organic matter in the water column (Supplement S.1, now in Section 2.1.3, and addressed in the other review), which increases rates of oxygen consumption due to aerobic remineralization, especially *following resuspension events*. Please recall that resuspension-induced oxygen consumption equals oxygen consumption from the standard model run minus oxygen consumption from the no-resuspension model run. Thus, although this effect occurs during the depositional portion of the resuspension event, it still increases the overall resuspension-induced oxygen consumption. Overall, the increase in aerobic remineralization offsets the decrease in nitrification estimated by the model during depositional periods.

3. I guess that the 20% increase in oxygen consumption of the seabed in case of resuspension results from the fact that the initial state of the diagenetic model has been estimated in the absence of resuspension events (although I am not sure this is how the model was initialized). Perhaps, for sediments where these resuspension events occur regularly, a better initial profile would be generated as a dynamic equilibrium that is established including these resuspension events? In this case, the average oxygen consumption would not be higher, but the variability would be increased due to resuspension.

Yes, the initial state of seabed biogeochemical profiles was based on results from a no-resuspension model run that was run until the model reached a steady state. This is noted in Section 2.2 (pg. 8, line 20).

For the Rhone delta implementation, the seabed profiles in surficial sediments (which have a largest effect on our results) adjusted within days to new environmental conditions.  As a test case, we re-ran the model with initial conditions that equaled the model results from the last timestep of the standard model.  These new profiles caused resuspension-induced seabed oxygen consumption to decrease slightly, from +20% to +15%, but primarily affected the first resuspension event.  Because this 5% change was small compared to other sensitivity tests, and did not change our main conclusion, we did not run additional tests.  This is now noted in Section 2.2 (pg. 8, line 20).

*Model description:*
4.  Several ways to describe the benthic model are quite confusing, not standard and sometimes inconsistent.

    a. The formula for Eised is confusing as Eised is described differently in equation (1) (page 5 line 13-14) compared to its description in table 1. It is not standard as it features the delta t (timestep). Timesteps only determine how the model is solved, and should not feature in model equations. I suggest to give the equation for the erosion *rate* instead of the eroded mass, as is usual in sediment modelling studies. This removes the delta t in the equation, and changes the units of Eised.
    Thank you for catching this error. We have corrected the discrepancy between equation (1) and table 1, and have switched to the 'standard' formula for erosion. (See Section 2.2.1, pg 5, lines 13-15).

    b. A related question concerns the sedimentation rate. Does this only apply when there is no erosion?
    Both reviewers noted that this was confusing, and so we have clarified this by adding text to Section 2.2.1 (pg. 5, lines 13-14), and by including the equations for deposition and net fluxes across the seabed-water interface, in addition to the formula for erosion (Section 2.2.1, pg. 5, lines 15).

    The background sedimentation rate of 10 cm $yr^{-1}$ (equivalent to 14 kg $m^{-2}$ $yr^{-1}$) was applied throughout the model run, but instantaneous rates of erosion and deposition varied, depending on hydrodynamic conditions. For example, when bed stresses were high, more sediment was eroded from the seabed than was deposited, causing net deposition rates to be negative.  Similarly, as bed stresses decreased, less sediment was eroded and more sediment settled to the seabed, causing an increase of the net deposition rate to >10 cm $yr^{-1}$.

5.  In diagenetic models, the units of dissolved substances are typically expressed in mmol/ m3 *liquid*, and porosity features in the diagenetic equation because diffusion takes place in the porewater, while the mass balance needs to be written for bulk sediment. In this model, the units of O2, and NO3, (and of the parameters kO2 and kNO3, etc…) are said to be mmol / m2. But what is this: mmol/m2 bulk sediment or mmol/m2 liquid? (my guess is that it is per m2 bulk,

but I am not sure). Moreover, the units are not always consistent. For instance, the dO2/dz is said to be in units of mmol O2/m4 which suggests that O2 is in mmol/m3. Finally, the units of the monod constants are said to be mmol/m2 in table 1, but in mmol/m3 in table 3. I suggest to represent the equations using concentrations per m3 liquid as is custom in diagenetic modelling. How this is actually implemented in the model is less relevant.

We changed units for seabed tracers to mmol m$^{-3}$ *liquid* for dissolved seabed tracers as is customary for seabed biogeochemical models. We note this in the caption of Table 1, and have edited the manuscript and equations accordingly. We also clarified units for the other variables that use m, m$^2$, m$^3$, and m$^4$.

6. Perhaps related to the previous comment: the equation for the diffusion across the sediment-water interface is very strange (in the supplement). I suspect -but could be wrong- that this is necessary to assure mass conservation and correct for using the wrong units (i.e. if units are per bulk sediment and not per liquid)?. Also, I could not see that this is how Soetaert et al. implemented sediment-water exchange in their model.

We chose this formulation because we felt it was a parsimonious, efficient method, not because it was necessary to assure mass conservation or to correct for units. Because the surficial sediment layer was so thin (0.1 mm), this parameterization produced the same results as a model sensitivity test run that used a Fickian Diffusion Law (noted in the Supplement, pg. 3, line 1).

However, we thank the reviewer for helping us realize that we mis-interpreted the description of the seabed-water column boundary condition in Soetaert et al. (1996). We removed this reference to Soetaert et al. (1996a,b) in the Supplement, pg. 2, line 24. (Note that although I cite the page and line numbers of the supplement from the original submission, in revision, the supplements were moved to the main body of the manuscript, as requested by both reviewers.)

7. It only becomes clear how resuspension is effectively included in this model based on the supplement. As this is the truly distinguishing feature, it should be included in the main paper. Related to that: how is this model integrated-i.e. which integration method is used? Many integration methods assume smooth dynamics, and one cannot just alter state variables directly.

Following both reviewers' advice, we have moved the supplement to the main paper, adding the information from S.1 to Sections 2.1.2 and 2.1.3 (pages 5-7). The material from S.2 was added as an Appendix to the paper because, although it is less crucial to understanding the results of the paper, we agreed that it should be in the same document as the rest of the manuscript.

The ROMS framework was written to handle additional tracers using a variety of integration schemes. Specific numerical schemes for our model are listed below and are now referenced in the revised manuscript:

i. For advection of water column tracers, we used the MPDATA second-order scheme (Smolarkiewicz and Margolin, 1998), which has been used in several sediment transport and biogeochemical models (e.g., Fennel et al., 2013; Feng et al., 2015; Fall et al., 2014; Xu et al., 2013). This is now referenced in Section 2.2 (pg. 8, lines 20-22).

ii. For vertical mixing, we used a GLS (generic length scale) approach (Umlauf and Burchard, 2009), which is also commonly used (e.g., Bever et al., 2013, McSweeney et al., 2016; Feng et al., 2015) and referenced in the manuscript (pg. 8 lines 20-22).

iii. Transport within the seabed due to diffusion was calculated based on Sherwood et al. (2016), who implemented a scheme based on Thomas' algorithm (Anderson, Tannehill and Pletcher, 1984; p. 549-550) that was mass-conserving, and appropriate for non-spatially uniform diffusion rates and seabed layer thickness. It was first-order accurate when implemented for typical regional modeling spatial and temporal scales. The manuscript now references this method in Section 2.1.1 (pg. 5, lines 20-21).

iv. Changes in concentrations due to biogeochemical reactions were estimated using first-order methods, consistent with other biogeochemical models included in ROMS (Fennel et al., 2006; Franks et al., 1986; Powell et al., 2006) and Soetaert et al. (1996a,b). This is now noted in 2.1.3 (pg. 6, line 7).

v. Changes in the seabed biogeochemical profiles due to advection (i.e., resuspension and deposition) were not estimated using numerical integration, but were estimated using the 'bookkeeping' methods used in sediment transport models, as described in Warner et al. (2008) and Supplement S.2. These references are noted in the text.

8. There is no need for figures 7, 8 and 9.
We agree that figure 9a-b could be removed, and have done so. We kept 9c as we believe it is useful to examine how accounting for resuspension affects estimates of seabed oxygen consumption over time. Given the reviewers' interest in the sensitivity tests, we retained Figures 7 and 8.

---

## Author Response (AR1)

Dear Dr. Jack Middelburg, Associate Editor

Thank you for your feedback. As requested, we have revised the manuscript as outlined in our previous response. Please note that when saving this marked-up version of the manuscript as a PDF, Microsoft word saved some of the changes in a separate document that we then attached to the end of this document.

In addition to the changes proposed by the reviewers and discussed in our response to reviews (dated 15 February 2017), we made a few minor changes to the manuscript. For example, we found a small discrepancy between a coefficient listed in the manuscript ($f_{lab}$ = 0.5) and the value used in the code ($f_{lab}$ = 0.515). In revision, we have re-run the model with $f_{lab}$=0.5. This did not affect any of the main messages of the paper, but sometimes changed numbers by couple percentages. We also edited Section 4.2 in response to Reviewer 1's Comment #5, but this was not noted in the original response to reviewers.

Overall, the constructive feedback provided by the review process has significantly improved the manuscript. Thank you for considering it for publication.

Best Regards,

Julia Moriarty, Courtney Harris, Christophe Rabouille, Katja Fennel, Marjorie Friedrichs, and Kevin Xu

[revised manuscript text omitted]

JMM 3/3/2017 10:41 AM

JMM 3/3/2017 10:41 AM

JMM 3/3/2017 10:41 AM

JMM 3/3/2017 10:41 AM

JMM 3/3/2017 10:41 AM

JMM 3/3/2017 10:41 AM

JMM 3/3/2017 10:41 AM

JMM 3/3/2017 10:41 AM

JMM 3/3/2017 10:41 AM
Deleted: Note that POC, $O_2$, $NO_3$, $NH_4$, and ODU are stored in units of mmol m$^{-2}$ in the model, so they are divided by dz to provide a concentration and to be consistent with Soetaert et al. (1996a, 199 ... [11]

JMM 3/3/2017 10:41 AM
Deleted: erosion and deposition, as well as diffusion. In the CSTMS module, when bed shear stress exceeds the critical shear stress of the seabed, sediment may be entrained from the seabed into the water column (Eq. 1). In HydroBioSed, any POM or dissolved chemical species in the porewater within the resuspended seabed layer(s) is also entrained into the bottom water column layer. Similarly, during depositional periods, phytoplankton, detritus and aggregates settling to the seabed are incorporated into the surficial seabed layer. Upon deposition, the model adds phytoplankton, detritus and labile aggregates to the pool of labile seabed organic matter, whereas refractory aggregates are added to refractory organic matter. Porewater in newly deposited sediments is assumed to initially have concentrations of nutrients and oxygen equal to those in the overlying water column. In addition to resuspension, dissolved oxygen and nutrients may diffuse across the seabed-water interface as described in Soetaert et al. (1996a, 1996b) and Supplement S.2. Overall, HydroBioSed can represent

[revised manuscript text omitted]

Formatted ... [28]
JMM 3/3/2017 10:41 AM
Formatted ... [32]
JMM 3/3/2017 10:41 AM
Formatted Table ... [29]
JMM 3/3/2017 10:41 AM
Formatted ... [30]
JMM 3/3/2017 10:41 AM
Formatted ... [31]
JMM 3/3/2017 10:41 AM
JMM 3/3/2017 10:41 AM
JMM 3/3/2017 10:41 AM
Formatted Table ... [33]
JMM 3/3/2017 10:41 AM
JMM 3/3/2017 10:41 AM
JMM 3/3/2017 10:41 AM
JMM 3/3/2017 10:41 AM
JMM 3/3/2017 10:41 AM
JMM 3/3/2017 10:41 AM
JMM 3/3/2017 10:41 AM
JMM 3/3/2017 10:41 AM
Formatted ... [34]
JMM 3/3/2017 10:41 AM
JMM 3/3/2017 10:41 AM
JMM 3/3/2017 10:41 AM
JMM 3/3/2017 10:41 AM
JMM 3/3/2017 10:41 AM
JMM 3/3/2017 10:41 AM
JMM 3/3/2017 10:41 AM
JMM 3/3/2017 10:41 AM
JMM 3/3/2017 10:41 AM
JMM 3/3/2017 10:41 AM
... [37]
JMM 3/3/2017 10:41 AM
... [38]
JMM 3/3/2017 10:41 AM
JMM 3/3/2017 10:41 AM
... [39]
JMM 3/3/2017 10:41 AM
... [40]
JMM 3/3/2017 10:41 AM
... [41]
JMM 3/3/2017 10:41 AM
JMM 3/3/2017 10:41 AM
JMM 3/3/2017 10:41 AM
JMM 3/3/2017 10:41 AM
... [42]
JMM 3/3/2017 10:41 AM
... [43]
JMM 3/3/2017 10:41 AM
... [44]
JMM 3/3/2017 10:41 AM

[revised manuscript text omitted]

JMM 3/3/2017 10:41 AM
Formatted [50]
JMM 3/3/2017 10:41 AM
Formatted [51]
JMM 3/3/2017 10:41 AM
Formatted Table [52]
JMM 3/3/2017 10:41 AM
JMM 3/3/2017 10:41 AM
Formatted Table [53]
JMM 3/3/2017 10:41 AM
JMM 3/3/2017 10:41 AM
JMM 3/3/2017 10:41 AM
JMM 3/3/2017 10:41 AM
JMM 3/3/2017 10:41 AM
JMM 3/3/2017 10:41 AM
JMM 3/3/2017 10:41 AM
Formatted Table [54]
JMM 3/3/2017 10:41 AM
JMM 3/3/2017 10:41 AM
JMM 3/3/2017 10:41 AM
JMM 3/3/2017 10:41 AM
JMM 3/3/2017 10:41 AM
JMM 3/3/2017 10:41 AM
JMM 3/3/2017 10:41 AM
JMM 3/3/2017 10:41 AM
JMM 3/3/2017 10:41 AM
JMM 3/3/2017 10:41 AM
JMM 3/3/2017 10:41 AM
JMM 3/3/2017 10:41 AM
JMM 3/3/2017 10:41 AM
JMM 3/3/2017 10:41 AM
JMM 3/3/2017 10:41 AM

| | | |
|---|---|---|
| Partitioning of Refractory vs. Labile Organic Matter | $f_{lab} = 0.5$ | Pastor et al. (2011a), Tesi et al. (2007) |
| Partitioning of Labile Aggregates vs. Large Detritus | $f_{ldet} = 0.5$ | Pastor et al. (2011a), Tesi et al. (2007) |
| **Seabed Biogeochemical Parameters** | | |
| Labile Organic Matter Remineralization Rate Constant | $11 \text{ y}^{-1}$ | Pastor et al. (2011a) |
| Refractory Organic Matter Remineralization Rate Constant | $0.31 \text{ y}^{-1}$ | Pastor et al. (2011a) |
| Ratio of mol C: mol N in Labile Organic Matter | 7.10 | Pastor et al. (2011a) |
| Ratio of mol C: mol N in Refractory Organic Matter | 14.3 | Pastor et al. (2011a) |
| Half-Saturation Constant for $O_2$ Limitation of Aerobic Remineralization | $k_{O2} = 1 \text{ mmol } O_2 \text{ m}^{-3}$ | Pastor et al. (2011a) |
| Half-Saturation Constant for $NO_3$ Limitation of Nitrate Remineralization (Denitrification) | $k_{NO3} = 20 \text{ mmol N m}^{-3}$ | Pastor et al. (2011a) |
| Half-Saturation Constant for $O_2$ Limitation of Nitrification | $k_{O2\_nit} = 10 \text{ mmol } O_2 \text{ m}^{-3}$ | Pastor et al. (2011a) |
| Half-Saturation Constant for $O_2$ Limitation in ODU Oxidation | $k_{O2\_oduox} = 1 \text{ mmol } O_2 \text{ m}^{-3}$ | Pastor et al. (2011a) |
| Half-Saturation Constant for $O_2$ Inhibition of Nitrate Remineralization (Denitrification) | $l_{O2} = 1 \text{ mmol } O_2 \text{ m}^{-3}$ | Pastor et al. (2011a) |
| Half-Saturation Constant for $O_2$ Inhibition of Anoxic Remineralization | $l_{O2\_anoxic} = 1 \text{ mmol } O_2 \text{ m}^{-3}$ | Pastor et al. (2011a) |
| Half-Saturation Constant for $NO_3$ Inhibition of Anoxic Remineralization | $l_{NO3\_anoxic} = 10 \text{ mmol } NO_3 \text{ m}^{-3}$ | Pastor et al. (2011a) |
| Maximum Nitrification Rate | $R_{nit,max} = 100 \text{ d}^{-1}$ | Pastor et al. (2011a) |
| Maximum Oxidation Rate of Oxygen Demand Units | $R_{oduox,max} = 20 \text{ d}^{-1}$ | Pastor et al. (2011a) |
| Fraction of ODUs Produced that are Solid and Inert | 99.5 % | Pastor et al. (2011a) |
| Diffusion Coefficient for Across Seabed-Water Interface | $D_{s-w} = 1.08 \cdot 10^{-9} \text{ m}^2 \text{ s}^{-1}$ | Toussaint et al. (2014) |
| Coefficients for Diffusion Within the Seabed | $D_{particulates} = 2.55 \cdot 10^{-10} \text{ m}^2 \text{ s}^{-1}$ $D_{O2} = 11.99 \cdot 10^{-10} \text{ m}^2 \text{ s}^{-1}$ $D_{NO3} = 9.80 \cdot 10^{-10} \text{ m}^2 \text{ s}^{-1}$ $D_{NH4} = 10.04 \cdot 10^{-10} \text{ m}^2 \text{ s}^{-1}$ $D_{ODU} = 4.01 \cdot 10^{-10} \text{ m}^2 \text{ s}^{-1}$ | [c]Pastor et al. (2011a) |

[a]Chosen based on time series of seabed elevation in Toussaint et al. (2014)
[b]Units are m$^3$ sediment, not m$^3$ water
[c]Chosen based on organic sedimentation rate
[d]No local data
[e]Derived from the molecular diffusion rates, but adjusted for the porosity and tortuosity of the seabed as described in Pastor et al., 2011a.

**Table 4: List of sensitivity tests. Additionally, for each simulation listed here, an identical model run was completed that neglected resuspension (i.e. with M = 0 kg/m$^2$/s ; $\tau_{crit}$ = 10 Pa).**

| Sensitivity Test Abbreviation | Sensitivity Test Name | Changed Parameters and/or Parameterizations Relative to the Standard Model Run |
|---|---|---|
| R1 | Low Erosion Rate Parameter | $M = 0.005 \text{ kg m}^{-2} \text{ s}^{-1}$ |
| R2 | High Erosion Rate Parameter | $M = 0.02 \text{ kg m}^{-2} \text{ s}^{-1}$ |
| T1 | Low Critical Shear Stress | $\tau_{crit} = 0.15 \text{ Pa}$ |
| T2 | High Critical Shear Stress | $\tau_{crit} = 0.6 \text{ Pa}$ |
| S1 | Low Inorganic Sedimentation | $S_{inorganic} = 0.05 \text{ m y}^{-1} = 7 \text{ kg m}^{-2} \text{ y}^{-1}$ |
| S2 | High Inorganic Sedimentation | $S_{inorganic} = 0.20 \text{ m y}^{-1} = 28 \text{ kg m}^{-2} \text{ y}^{-1}$ |
| P1 | Low Particulate Organic Sedimentation | $S_{organic} = 328.5 \text{ gC m}^{-2} \text{ y}^{-1}$ |
| P2 | High Particulate Organic Sedimentation | $S_{inorganic} = 1314 \text{ gC m}^{-2} \text{ y}^{-1}$ |
| L1 | Low Lability | $f_{lab} = 0.20$ |
| L2 | High Lability | $f_{lab} = 0.80$ |
| B1 | Low Seabed Diffusion | $D_i = \text{original values} * 0.5$ |
| B2 | High Seabed Diffusion | $D_i = \text{original values} * 2.0$ |
| N1 | Low Nitrification Rate | $R_{nit,max} = 50 \text{ d}^{-1}$ |

JMM 3/3/2017 10:41 AM
JMM 3/3/2017 10:41 AM
JMM 3/3/2017 10:41 AM
JMM 3/3/2017 10:41 AM
Formatted Table ... [55]
JMM 3/3/2017 10:41 AM
JMM 3/3/2017 10:41 AM
JMM 3/3/2017 10:41 AM
JMM 3/3/2017 10:41 AM
JMM 3/3/2017 10:41 AM
JMM 3/3/2017 10:41 AM
JMM 3/3/2017 10:41 AM
JMM 3/3/2017 10:41 AM
JMM 3/3/2017 10:41 AM
JMM 3/3/2017 10:41 AM
JMM 3/3/2017 10:41 AM
JMM 3/3/2017 10:41 AM
JMM 3/3/2017 10:41 AM
JMM 3/3/2017 10:41 AM
JMM 3/3/2017 10:41 AM
JMM 3/3/2017 10:41 AM
Formatted ... [63]
JMM 3/3/2017 10:41 AM
JMM 3/3/2017 10:41 AM
JMM 3/3/2017 10:41 AM
JMM 3/3/2017 10:41 AM
Formatted ... [64]
JMM 3/3/2017 10:41 AM
JMM 3/3/2017 10:41 AM
Formatted ... [65]
JMM 3/3/2017 10:41 AM
JMM 3/3/2017 10:41 AM
Formatted ... [66]
JMM 3/3/2017 10:41 AM
JMM 3/3/2017 10:41 AM
Formatted ... [67]
JMM 3/3/2017 10:41 AM
JMM 3/3/2017 10:41 AM
JMM 3/3/2017 10:41 AM
JMM 3/3/2017 10:41 AM
... [68]
JMM 3/3/2017 10:41 AM

| N2 | High Nitrification Rate | $R_{nit,max} = 200$ $d^{-1}$ |
| C1 | No-Repartitioning | See Fig. 2c; Sect. 2.2.3 |

**Table 5:** Statistics for model-observation comparison, including the root mean square difference (RMSD) and the correlation coefficient (R). The mean and standard deviation of estimates from both the model and observations are also shown.

| | RMSD | R | Mean ± Standard Deviation | |
| --- | --- | --- | --- | --- |
| | | | Model | Observations |
| **Seabed Height** | 1.39 cm | 0.21 | -0.52 ± 0.82 cm | -1.1 ± 1.2 cm |
| **O₂ Gradient** | 105 mol O$_2$ m$^{-4}$ | 0.48 | 180 ± 118 mol O$_2$ m$^{-4}$ | 173 ± 76 mol O$_2$ m$^{-4}$ |

**Table 6:** O$_2$ Consumption (mmol O$_2$ m$^{-2}$ d$^{-1}$) in the seabed, bottom water, and combined seabed-bottom water due to various processes over the two-month model run, and during periods of deposition and erosion. Abbreviations include: POM Rem. (particulate organic matter remineralization); ODU Ox (Oxidation of ODUs); Nit (nitrification); and "Seabed + BW" (the combined seabed-bottom water region).

| | Seabed | | | | Bottom Waters | | | | Seabed + BW |
| --- | --- | --- | --- | --- | --- | --- | --- | --- | --- |
| | Total | POM Rem. | Nit. | ODU Ox. | Total | POM Rem. | Nit. | ODU Ox. | Total |
| **2-Month Average** | 19 | 5.0 | 14 | 0.20 | 56 | 31 | 24 | 0.30 | 74 |
| **Minimum Values over 2 Months** | 12 | 0.56 | 3.7 | 0.01 | 23 | 0.08 | 22 | 0 | 39 |
| **Maximum Values over 2 Months** | 35 | 18 | 33 | 0.64 | 200 | 170 | 34 | 10. | 220 |
| **Average During Depositional Periods** | 18 | 5.5 | 12 | 0.18 | 47 | 23 | 24 | 0.18 | 65 |
| **Average During Erosional Periods** | 21 | 3.3 | 18 | 0.26 | 90. | 63 | 26 | 0.78 | 110 |

**Table A.1:** Parameters for new seabed layering scheme, as implemented for the Rhône study site. Dashed lines indicate that no symbol was assigned to that parameter.

| Type of Layer | Symbol for Number of Layers | Number of Layers for Rhône model implementation | Symbol for Thickness of Each Layer | Thickness of Each Layer for Rhône model implementation (mm) |
| --- | --- | --- | --- | --- |
| **Active Transport Layer** (i.e., the Surficial Layer) | - | 1 | $z_a$ | 0.1 |
| **High-Resolution Layers** | $N_{high-res}$ | 19 | $z_{high-res}$ | 0.5 |
| **Medium-Resolution Layers** | $N_{med-res}$ | 39 | $z_{med-res}$ | 10 |
| **Repository** | -- | 1 | | Varies; 333 m at initialization |

JMM 3/3/2017 10:41 AM
JMM 3/3/2017 10:41 AM
JMM 3/3/2017 10:41 AM
JMM 3/3/2017 10:41 AM
Split Cells ... [70]
JMM 3/3/2017 10:41 AM
Split Cells ... [72]
JMM 3/3/2017 10:41 AM
Split Cells ... [73]
JMM 3/3/2017 10:41 AM
Formatted Table ... [71]
JMM 3/3/2017 10:41 AM
JMM 3/3/2017 10:41 AM
JMM 3/3/2017 10:41 AM
Formatted Table ... [74]
JMM 3/3/2017 10:41 AM
JMM 3/3/2017 10:41 AM
Merged Cells ... [76]
JMM 3/3/2017 10:41 AM
Formatted Table ... [77]
JMM 3/3/2017 10:41 AM
JMM 3/3/2017 10:41 AM
JMM 3/3/2017 10:41 AM
Formatted ... [78]
JMM 3/3/2017 10:41 AM
JMM 3/3/2017 10:41 AM
JMM 3/3/2017 10:41 AM
Deleted Cells ... [84]
JMM 3/3/2017 10:41 AM
Formatted Table ... [79]
JMM 3/3/2017 10:41 AM
JMM 3/3/2017 10:41 AM
JMM 3/3/2017 10:41 AM
JMM 3/3/2017 10:41 AM
JMM 3/3/2017 10:41 AM
JMM 3/3/2017 10:41 AM
Deleted Cells ... [80]
JMM 3/3/2017 10:41 AM
JMM 3/3/2017 10:41 AM
Deleted Cells ... [81]
JMM 3/3/2017 10:41 AM
JMM 3/3/2017 10:41 AM
Deleted Cells ... [82]
JMM 3/3/2017 10:41 AM
JMM 3/3/2017 10:41 AM
Deleted Cells ... [83]
JMM 3/3/2017 10:41 AM
JMM 3/3/2017 10:41 AM
JMM 3/3/2017 10:41 AM
JMM 3/3/2017 10:41 AM
JMM 3/3/2017 10:41 AM
JMM 3/3/2017 10:41 AM

[Figure]

Figure 1: a) Red box indicates location of panel (b) in the Gulf of Lions. b) Dots indicate our study site (SS; blue), i.e. the Mesurho station (Pairaud et al., 2016), and Pastor et al. (2014)'s Site A (green) offshore of the Rhône River. Bathymetric data (black lines) were obtained from the European Marine Observation and Data Network. Coastline data were obtained from the U.S. National Oceanic and Atmospheric Administration.

JMM 3/3/2017 10:41 AM

JMM 3/3/2017 10:41 AM
JMM 3/3/2017 10:41 AM
JMM 3/3/2017 10:41 AM
JMM 3/3/2017 10:41 AM

[Figure]

[Figure]

JMM 3/3/2017 10:41 AM

[revised manuscript text omitted]

JMM 3/3/2017 10:41 AM

Unknown

JMM 3/3/2017 10:41 AM

JMM 3/3/2017 10:41 AM

JMM 3/3/2017 10:41 AM

JMM 3/3/2017 10:41 AM

JMM 3/3/2017 10:41 AM

| Page 5: [1] Deleted | JMM | 3/3/17 10:41 AM |
|---|---|---|
| Rhone | | |

| Page 5: [1] Deleted | JMM | 3/3/17 10:41 AM |
|---|---|---|
| Rhone | | |

| Page 5: [1] Deleted | JMM | 3/3/17 10:41 AM |
|---|---|---|
| Rhone | | |

| Page 5: [1] Deleted | JMM | 3/3/17 10:41 AM |
|---|---|---|
| Rhone | | |

| Page 5: [1] Deleted | JMM | 3/3/17 10:41 AM |
|---|---|---|
| Rhone | | |

| Page 5: [1] Deleted | JMM | 3/3/17 10:41 AM |
|---|---|---|
| Rhone | | |

| Page 5: [1] Deleted | JMM | 3/3/17 10:41 AM |
|---|---|---|
| Rhone | | |

| Page 5: [1] Deleted | JMM | 3/3/17 10:41 AM |
|---|---|---|
| Rhone | | |

| Page 5: [1] Deleted | JMM | 3/3/17 10:41 AM |
|---|---|---|
| Rhone | | |

| Page 5: [1] Deleted | JMM | 3/3/17 10:41 AM |
|---|---|---|
| Rhone | | |

| Page 5: [1] Deleted | JMM | 3/3/17 10:41 AM |
|---|---|---|
| Rhone | | |

| Page 5: [1] Deleted | JMM | 3/3/17 10:41 AM |
|---|---|---|
| Rhone | | |

| Page 5: [1] Deleted | JMM | 3/3/17 10:41 AM |
|---|---|---|
| Rhone | | |

| Page 5: [2] Deleted | JMM | 3/3/17 10:41 AM |
|---|---|---|
| resuspension | | |

| Page 5: [2] Deleted | JMM | 3/3/17 10:41 AM |
|---|---|---|
| resuspension | | |

| Page 5: [2] Deleted | JMM | 3/3/17 10:41 AM |
|---|---|---|
| resuspension | | |

| Page 5: [2] Deleted | JMM | 3/3/17 10:41 AM |

resuspension

| Page 5: [3] Deleted | JMM | 3/3/17 10:41 AM |

rates

| Page 8: [4] Deleted | JMM | 3/3/17 10:41 AM |

Wilson et al. (2013

| Page 8: [5] Deleted | JMM | 3/3/17 10:41 AM |

molecule

| Page 8: [5] Deleted | JMM | 3/3/17 10:41 AM |

molecule

| Page 8: [6] Deleted | JMM | 3/3/17 10:41 AM |

$$\frac{POC}{dz} \times R_{POC}$$

| Page 8: [7] Deleted | JMM | 3/3/17 10:41 AM |

$$\frac{POC}{dz} \times R_{POC}$$

| Page 8: [8] Deleted | JMM | 3/3/17 10:41 AM |

$$\frac{POC}{dz} \times R_{POC}$$

| Page 8: [9] Deleted | JMM | 3/3/17 10:41 AM |

$$\frac{NH_4}{dz}$$

| Page 8: [10] Deleted | JMM | 3/3/17 10:41 AM |

$$\frac{ODU}{dz} \times R_{odu,max}$$

| Page 8: [11] Deleted | JMM | 3/3/17 10:41 AM |

Note that POC, $O_2$, $NO_3$, $NH_4$, and ODU are stored in units of mmol m$^{-2}$ in the model, so they are divided by dz to provide a concentration and to be consistent with Soetaert et al. (1996a, 1996b).

**Page 8: [11] Deleted**            **JMM**            **3/3/17 10:41 AM**

Note that POC, $O_2$, $NO_3$, $NH_4$, and ODU are stored in units of mmol m$^{-2}$ in the model, so they are divided by dz to provide a concentration and to be consistent with Soetaert et al. (1996a, 1996b).

**Page 15: [12] Deleted**            **JMM**            **3/3/17 10:41 AM**

to the

| **Page 15: [12] Deleted** | **JMM** | **3/3/17 10:41 AM** |
|---|---|---|

to the

| **Page 15: [12] Deleted** | **JMM** | **3/3/17 10:41 AM** |
|---|---|---|

to the

| **Page 15: [12] Deleted** | **JMM** | **3/3/17 10:41 AM** |
|---|---|---|

to the

| **Page 15: [12] Deleted** | **JMM** | **3/3/17 10:41 AM** |
|---|---|---|

to the

| **Page 15: [12] Deleted** | **JMM** | **3/3/17 10:41 AM** |
|---|---|---|

to the

| **Page 15: [12] Deleted** | **JMM** | **3/3/17 10:41 AM** |
|---|---|---|

to the

| **Page 15: [12] Deleted** | **JMM** | **3/3/17 10:41 AM** |
|---|---|---|

to the

| **Page 15: [12] Deleted** | **JMM** | **3/3/17 10:41 AM** |
|---|---|---|

to the

| **Page 15: [12] Deleted** | **JMM** | **3/3/17 10:41 AM** |
|---|---|---|

to the

| **Page 15: [12] Deleted** | **JMM** | **3/3/17 10:41 AM** |
|---|---|---|

to the

| **Page 15: [12] Deleted** | **JMM** | **3/3/17 10:41 AM** |
|---|---|---|

to the

| **Page 15: [12] Deleted** | **JMM** | **3/3/17 10:41 AM** |
|---|---|---|

to the

| **Page 15: [12] Deleted** | **JMM** | **3/3/17 10:41 AM** |
|---|---|---|

to the

| **Page 15: [12] Deleted** | **JMM** | **3/3/17 10:41 AM** |
|---|---|---|

to the

| **Page 15: [13] Deleted** | **JMM** | **3/3/17 10:41 AM** |
|---|---|---|

seabed and

| Page 15: [13] Deleted | JMM | 3/3/17 10:41 AM |
|---|---|---|

seabed and

| Page 15: [13] Deleted | JMM | 3/3/17 10:41 AM |
|---|---|---|

seabed and

| Page 15: [13] Deleted | JMM | 3/3/17 10:41 AM |
|---|---|---|

seabed and

| Page 15: [13] Deleted | JMM | 3/3/17 10:41 AM |
|---|---|---|

seabed and

| Page 15: [13] Deleted | JMM | 3/3/17 10:41 AM |
|---|---|---|

seabed and

| Page 15: [13] Deleted | JMM | 3/3/17 10:41 AM |
|---|---|---|

seabed and

| Page 15: [13] Deleted | JMM | 3/3/17 10:41 AM |
|---|---|---|

seabed and

| Page 15: [13] Deleted | JMM | 3/3/17 10:41 AM |
|---|---|---|

seabed and

| Page 15: [13] Deleted | JMM | 3/3/17 10:41 AM |
|---|---|---|

seabed and

| Page 15: [13] Deleted | JMM | 3/3/17 10:41 AM |
|---|---|---|

seabed and

| Page 15: [14] Deleted | JMM | 3/3/17 10:41 AM |
|---|---|---|

supply

| Page 15: [14] Deleted | JMM | 3/3/17 10:41 AM |
|---|---|---|

supply

| Page 15: [14] Deleted | JMM | 3/3/17 10:41 AM |
|---|---|---|

supply

| Page 15: [14] Deleted | JMM | 3/3/17 10:41 AM |
|---|---|---|

supply

| Page 15: [14] Deleted | JMM | 3/3/17 10:41 AM |
|---|---|---|

supply

| Page 17: [15] Deleted | JMM | 3/3/17 10:41 AM |
|---|---|---|

-lived

| Page 17: [15] Deleted | JMM | 3/3/17 10:41 AM |
|---|---|---|

-lived

| Page 17: [16] Deleted | JMM | 3/3/17 10:41 AM |
|---|---|---|

Rhone Subaqueous Delta

**Page 17: [16] Deleted**                    **JMM**                    **3/3/17 10:41 AM**

Rhone Subaqueous Delta

**Page 17: [16] Deleted**                    **JMM**                    **3/3/17 10:41 AM**

Rhone Subaqueous Delta

**Page 17: [16] Deleted**                    **JMM**                    **3/3/17 10:41 AM**

Rhone Subaqueous Delta

**Page 17: [16] Deleted**                    **JMM**                    **3/3/17 10:41 AM**

Rhone Subaqueous Delta

**Page 17: [16] Deleted**                    **JMM**                    **3/3/17 10:41 AM**

Rhone Subaqueous Delta

**Page 17: [16] Deleted**                    **JMM**                    **3/3/17 10:41 AM**

Rhone Subaqueous Delta

**Page 17: [16] Deleted**                    **JMM**                    **3/3/17 10:41 AM**

Rhone Subaqueous Delta

**Page 17: [16] Deleted**                    **JMM**                    **3/3/17 10:41 AM**

Rhone Subaqueous Delta

**Page 17: [16] Deleted**                    **JMM**                    **3/3/17 10:41 AM**

Rhone Subaqueous Delta

**Page 17: [16] Deleted**                    **JMM**                    **3/3/17 10:41 AM**

Rhone Subaqueous Delta

**Page 17: [16] Deleted**                    **JMM**                    **3/3/17 10:41 AM**

Rhone Subaqueous Delta

**Page 17: [16] Deleted**                    **JMM**                    **3/3/17 10:41 AM**

Rhone Subaqueous Delta

**Page 20: [17] Deleted**                    **JMM**                    **3/3/17 10:41 AM**

The remainder of this section explores what sediment processes were most critical for modeling the effect of resuspension on Rhone Delta oxygen dynamics. First, resuspension increased the importance of the bottom boundary layer relative to the seabed. During quiescent conditions, the bottom boundary layer and seabed each accounted for similar rates of oxygen consumption. However, when POM and porewater were entrained into the water column via resuspension, bottom boundary layer oxygen consumption increased by a factor of eight, while seabed oxygen consumption only doubled. This disproportionate increase of oxygen consumption within the bottom boundary layer affirmed the importance of observing and modeling oxygen dynamics within the bottom boundary layer during resuspension events. Also, only accounting for

quiescent time periods would underestimate the role of the bottom boundary layer, which accounted for 75% of the total oxygen consumption over the two-month model run for the Rhone Delta site, but only accounted for about 50% when resuspension was neglected.

**Diffusion of oxygen across the sediment-water interface dominated the supply of oxygen to the seabed in the model, regardless of the timescale or time period considered. The other transport mechanism, the "pumping" of oxygen into and out of the seabed when sediments were deposited or eroded, provided at most a third of the flux to the seabed (during depositional time periods; Fig. 5). Also, "pumping" contributed much less to seabed oxygen supply over time, primarily because the entrainment of porewater from the seabed into the water column during erosional periods partially offset the depositional flux of oxygen (Fig. 5). Over the two-month model simulation, diffusion across the seabed-water interface accounted for 96% of the seabed oxygen supply, whereas "pumping" due to erosion and deposition accounted for only 4% of seabed oxygen fluxes. Thus, for environments like the Rhone**

| Page 20: [18] Deleted | JMM | 3/3/17 10:41 AM |
|---|---|---|

, as well as water column concentrations of nutrients and oxygen, constant in time. Yet, many coastal environments experience daily, tidal, seasonal, or inter-annual variations in their supply. On the Rhone Delta, for example, flood events may deliver relatively refractory organic matter to the site, lowering seabed oxygen consumption in spite of the large quantity of riverine organic matter deposited on the shelf (e.g. Cathalot et al., 2010). Investigating how these temporal variations affect the relative importance of resuspension for oxygen dynamics could be useful for further extrapolating our results to different environments.

| Page 20: [19] Moved to page 21 (Move #6) | JMM | 3/3/17 10:41 AM |
|---|---|---|

Our analysis focused on oxygen, but resuspension also affected model estimates of nitrogen dynamics. For example, during quiescent periods, nitrification roughly balanced production of ammonium from remineralization of organic matter in the seabed. However, during erosional periods, the exposure of ammonium-rich porewater to oxygen increased seabed nitrification, enhancing fluxes of nitrate out of the seabed, consistent with observations from other systems (e.g. Fanning et al., 1982; Sloth et al., 1996; Tengberg et al., 2003). Overall, resuspension increased nitrate fluxes out of the seabed by about a factor of 2 during resuspension, which led to about a 10

| Page 20: [20] Deleted | JMM | 3/3/17 10:41 AM |
|---|---|---|

% increase overall for the two-month model run.

HydroBioSed did not represent all processes that occur near the seabed-water column interface, and so future work could include accounting for turbulence-induced changes in diffusion, advective fluxes through the seabed, and porosity variations in the seabed. Within HydroBioSed, for example, the steepening of the oxygen gradient at the seabed-water interface occurred because of changes in oxygen concentrations within the seabed and bottom boundary layer (Fig.

3). HydroBioSed did not account for the thinning of the viscous layer at the seabed-water interface in response to wave-induced turbulence, which would act to further increase the oxygen gradient during erosional time periods (Gundersen and Jorgensen, 1990; Chatelain and Guizien, 2010; Wang et al., 2013).

 estimates of oxygen diffusion into the seabed during resuspension events are conservative. Additionally, the model could be adapted for locations where waves and currents drive flows of water through non-cohesive seabeds, stimulating biogeochemical reactions (Huettel et al., 2014), or to account for vertical gradients in seabed porosity (Soetaert et al., 1996a, 1996b).

Applying HydroBioSed for a three-dimensional system would facilitate its application to more scientific and water quality concerns.

seabed and bottom boundary

seabed and bottom boundary

seabed and bottom boundary

seabed and bottom boundary

seabed and bottom boundary

seabed and bottom boundary

seabed and bottom boundary

seabed and bottom boundary

seabed and bottom boundary

seabed and bottom boundary

| Page 22: [24] Deleted | JMM | 3/3/17 10:41 AM |
|---|---|---|

seabed and bottom boundary

| Page 22: [24] Deleted | JMM | 3/3/17 10:41 AM |
|---|---|---|

seabed and bottom boundary

| Page 22: [24] Deleted | JMM | 3/3/17 10:41 AM |
|---|---|---|

seabed and bottom boundary

| Page 22: [24] Deleted | JMM | 3/3/17 10:41 AM |
|---|---|---|

seabed and bottom boundary

| Page 22: [24] Deleted | JMM | 3/3/17 10:41 AM |
|---|---|---|

seabed and bottom boundary

| Page 22: [24] Deleted | JMM | 3/3/17 10:41 AM |
|---|---|---|

seabed and bottom boundary

| Page 22: [24] Deleted | JMM | 3/3/17 10:41 AM |
|---|---|---|

seabed and bottom boundary

| Page 22: [24] Deleted | JMM | 3/3/17 10:41 AM |
|---|---|---|

seabed and bottom boundary

| Page 22: [24] Deleted | JMM | 3/3/17 10:41 AM |
|---|---|---|

seabed and bottom boundary

| Page 22: [25] Deleted | JMM | 3/3/17 10:41 AM |
|---|---|---|

Rhone Subaqueous Delta

| Page 22: [25] Deleted | JMM | 3/3/17 10:41 AM |
|---|---|---|

Rhone Subaqueous Delta

| Page 22: [25] Deleted | JMM | 3/3/17 10:41 AM |
|---|---|---|

Rhone Subaqueous Delta

| Page 26: [26] Deleted | JMM | 3/3/17 10:41 AM |
|---|---|---|

Mobile deltaic and continental shelf muds as suboxic, fluidized bed reactors, Mar. Chem., 61, 143–155,

1998.

Aller, R.C.:

| Page 29: [27] Deleted | JMM | 3/3/17 10:41 AM |
|---|---|---|

Haidvogel, D.B., Arango, H

| Page 35: [28] Formatted | JMM | 3/3/17 10:41 AM |
|---|---|---|

Line spacing:  single

| Page 35: [29] Formatted Table | JMM | 3/3/17 10:41 AM |

Formatted Table

| Page 35: [30] Formatted | JMM | 3/3/17 10:41 AM |

Font:Not Bold

| Page 35: [31] Formatted | JMM | 3/3/17 10:41 AM |

Font:Not Bold

| Page 35: [32] Formatted | JMM | 3/3/17 10:41 AM |

Font:Not Bold

| Page 35: [33] Formatted Table | JMM | 3/3/17 10:41 AM |

Formatted Table

| Page 35: [34] Formatted | JMM | 3/3/17 10:41 AM |

Font:Italic

| Page 35: [35] Deleted | JMM | 3/3/17 10:41 AM |

$f_{ldet}$      Fraction of labile coagulated organic matter that is large detritus within the water column     ---

| Page 35: [36] Deleted | JMM | 3/3/17 10:41 AM |

$^1$mmol

| Page 35: [36] Deleted | JMM | 3/3/17 10:41 AM |

$^1$mmol

| Page 35: [37] Deleted | JMM | 3/3/17 10:41 AM |

$^1$mmol

| Page 35: [37] Deleted | JMM | 3/3/17 10:41 AM |

$^1$mmol

| Page 35: [38] Deleted | JMM | 3/3/17 10:41 AM |

$^1$mmol

| Page 35: [38] Deleted | JMM | 3/3/17 10:41 AM |

$^1$mmol

| Page 35: [39] Deleted | JMM | 3/3/17 10:41 AM |

mmol

| Page 35: [39] Deleted | JMM | 3/3/17 10:41 AM |

mmol

| Page 35: [40] Deleted | JMM | 3/3/17 10:41 AM |

mmol

| Page 35: [40] Deleted | JMM | 3/3/17 10:41 AM |

mmol

| Page 35: [41] Deleted | JMM | 3/3/17 10:41 AM |

mmol

| Page 35: [41] Deleted | JMM | 3/3/17 10:41 AM |

mmol

| Page 35: [42] Deleted | JMM | 3/3/17 10:41 AM |
|---|---|---|

[1]mmol

| Page 35: [42] Deleted | JMM | 3/3/17 10:41 AM |
|---|---|---|

[1]mmol

| Page 35: [43] Deleted | JMM | 3/3/17 10:41 AM |
|---|---|---|

[1]mmol

| Page 35: [43] Deleted | JMM | 3/3/17 10:41 AM |
|---|---|---|

[1]mmol

| Page 35: [44] Deleted | JMM | 3/3/17 10:41 AM |
|---|---|---|

mmol

| Page 35: [44] Deleted | JMM | 3/3/17 10:41 AM |
|---|---|---|

mmol

| Page 35: [45] Deleted | JMM | 3/3/17 10:41 AM |
|---|---|---|

[1]mmol ODU

| Page 35: [45] Deleted | JMM | 3/3/17 10:41 AM |
|---|---|---|

[1]mmol ODU

| Page 35: [46] Deleted | JMM | 3/3/17 10:41 AM |
|---|---|---|

[1]mmol

| Page 35: [46] Deleted | JMM | 3/3/17 10:41 AM |
|---|---|---|

[1]mmol

| Page 35: [47] Deleted | JMM | 3/3/17 10:41 AM |
|---|---|---|

[1]mmol

| Page 35: [47] Deleted | JMM | 3/3/17 10:41 AM |
|---|---|---|

[1]mmol

| Page 36: [48] Deleted | JMM | 3/3/17 10:41 AM |
|---|---|---|

[1]unless otherwise noted

| Page 37: [49] Deleted | JMM | 3/3/17 10:41 AM |
|---|---|---|

| Page 37: [50] Formatted | JMM | 3/3/17 10:41 AM |
|---|---|---|

Font:Times New Roman

| Page 37: [51] Formatted | JMM | 3/3/17 10:41 AM |
|---|---|---|

Indent: Left:  0", First line:  0"

| Page 37: [52] Formatted Table | JMM | 3/3/17 10:41 AM |
|---|---|---|

Formatted Table

| Page 37: [53] Formatted Table | JMM | 3/3/17 10:41 AM |
|---|---|---|

Formatted Table

| Page 37: [54] Formatted Table | JMM | 3/3/17 10:41 AM |
|---|---|---|

Formatted Table

| Page 38: [55] Formatted Table | JMM | 3/3/17 10:41 AM |
|---|---|---|

Formatted Table

| Page 38: [56] Deleted | JMM | 3/3/17 10:41 AM |
|---|---|---|

µmol

| Page 38: [56] Deleted | JMM | 3/3/17 10:41 AM |
|---|---|---|

µmol

| Page 38: [57] Deleted | JMM | 3/3/17 10:41 AM |
|---|---|---|

100 µmol

| Page 38: [57] Deleted | JMM | 3/3/17 10:41 AM |
|---|---|---|

100 µmol

| Page 38: [58] Deleted | JMM | 3/3/17 10:41 AM |
|---|---|---|

µmol

| Page 38: [58] Deleted | JMM | 3/3/17 10:41 AM |
|---|---|---|

µmol

| Page 38: [59] Deleted | JMM | 3/3/17 10:41 AM |
|---|---|---|

2 µmol

| Page 38: [59] Deleted | JMM | 3/3/17 10:41 AM |
|---|---|---|

2 µmol

| Page 38: [60] Deleted | JMM | 3/3/17 10:41 AM |
|---|---|---|

µmol

| Page 38: [60] Deleted | JMM | 3/3/17 10:41 AM |
|---|---|---|

µmol

| Page 38: [61] Deleted | JMM | 3/3/17 10:41 AM |
|---|---|---|

µmol

| Page 38: [61] Deleted | JMM | 3/3/17 10:41 AM |
|---|---|---|

µmol

| Page 38: [62] Deleted | JMM | 3/3/17 10:41 AM |
|---|---|---|

µmol

**Page 38: [62] Deleted**      JMM      3/3/17 10:41 AM

μmol

**Page 38: [63] Formatted**      JMM      3/3/17 10:41 AM

Superscript

**Page 38: [64] Formatted**      JMM      3/3/17 10:41 AM

Superscript

**Page 38: [65] Formatted**      JMM      3/3/17 10:41 AM

Superscript

**Page 38: [66] Formatted**      JMM      3/3/17 10:41 AM

Superscript

**Page 38: [67] Formatted**      JMM      3/3/17 10:41 AM

Superscript

**Page 38: [68] Deleted**      JMM      3/3/17 10:41 AM

[4]These rates derive

**Page 38: [68] Deleted**      JMM      3/3/17 10:41 AM

[4]These rates derive

**Page 39: [69] Deleted**      JMM      3/3/17 10:41 AM

O2      High ODU Oxidation Rate      $R_{oduox,max} = 40$

**Page 39: [70] Split Cells**      JMM      3/3/17 10:41 AM

Split Cells

**Page 39: [71] Formatted Table**      JMM      3/3/17 10:41 AM

Formatted Table

**Page 39: [72] Split Cells**      JMM      3/3/17 10:41 AM

Split Cells

**Page 39: [73] Split Cells**      JMM      3/3/17 10:41 AM

Split Cells

**Page 39: [74] Formatted Table**      JMM      3/3/17 10:41 AM

Formatted Table

**Page 39: [75] Deleted**      JMM      3/3/17 10:41 AM

**boundary layer**

**Page 39: [75] Deleted**      JMM      3/3/17 10:41 AM

**boundary layer**

**Page 39: [75] Deleted**      JMM      3/3/17 10:41 AM

**boundary layer**

**Page 39: [75] Deleted**     JMM     3/3/17 10:41 AM

**boundary layer**

**Page 39: [76] Merged Cells**     JMM     3/3/17 10:41 AM

Merged Cells

**Page 39: [77] Formatted Table**     JMM     3/3/17 10:41 AM

Formatted Table

**Page 39: [78] Formatted**     JMM     3/3/17 10:41 AM

Font:Bold

**Page 39: [79] Formatted Table**     JMM     3/3/17 10:41 AM

Formatted Table

**Page 39: [80] Deleted Cells**     JMM     3/3/17 10:41 AM

Deleted Cells

**Page 39: [81] Deleted Cells**     JMM     3/3/17 10:41 AM

Deleted Cells

**Page 39: [82] Deleted Cells**     JMM     3/3/17 10:41 AM

Deleted Cells

**Page 39: [83] Deleted Cells**     JMM     3/3/17 10:41 AM

Deleted Cells

**Page 39: [84] Deleted Cells**     JMM     3/3/17 10:41 AM

Deleted Cells

**Page 39: [85] Deleted**     JMM     3/3/17 10:41 AM

| | | | | | | | | | |
|---|---|---|---|---|---|---|---|---|---|
| **Maximum Values** | 35 | 18 | 32 | 0.63 | 190 | 160 | 20 | 0.003 | 217 |
| **Quiescent Conditions, i.e. value on June 1High-Resolution Layers** | $18N_{high\text{-}res}$ | 4.6 | 13 | 0.19 | $23z_{high\text{-}res}$ | 0.735 | 13 | 0 | 41 |

**Page 39: [86] Formatted Table**     JMM     3/3/17 10:41 AM

Formatted Table

**Page 39: [87] Deleted Cells**     JMM     3/3/17 10:41 AM

Deleted Cells

**Page 39: [88] Deleted Cells**     JMM     3/3/17 10:41 AM

Deleted Cells

| Page 39: [89] Deleted Cells | JMM | 3/3/17 10:41 AM |
|---|---|---|

Deleted Cells

| Page 39: [90] Deleted Cells | JMM | 3/3/17 10:41 AM |
|---|---|---|

Deleted Cells

| Page 39: [91] Deleted Cells | JMM | 3/3/17 10:41 AM |
|---|---|---|

Deleted Cells

| Page 40: [92] Deleted | JMM | 3/3/17 10:41 AM |
|---|---|---|

[Figure]

| Page 44: [93] Deleted | JMM | 3/3/17 10:41 AM |
|---|---|---|

2$^{nd}$

| Page 44: [93] Deleted | JMM | 3/3/17 10:41 AM |
|---|---|---|

2$^{nd}$

| Page 44: [93] Deleted | JMM | 3/3/17 10:41 AM |
|---|---|---|

2$^{nd}$

| Page 44: [93] Deleted | JMM | 3/3/17 10:41 AM |
|---|---|---|

2$^{nd}$

| Page 44: [93] Deleted | JMM | 3/3/17 10:41 AM |
|---|---|---|

2$^{nd}$

| Page 47: [94] Deleted | JMM | 3/3/17 10:41 AM |
|---|---|---|

**Same as Fig. 7, but for oxygen consumption in the bottom boundary layer (i.e. the bottom 4 m of the water column).**

[Figure]

[Figure]

[Figure]

**Figure 9:**

---

## Author Response (AR2)

Dear Dr. Jack Middelburg, Associate Editor

5 We appreciate your careful review of the manuscript. We have addressed all of your comments below using red text, and all changes to the manuscript are indicated using Microsoft Word's Track Changes feature. Additionally, we made a few wording changes to keep the manuscript at 44 pages, and added information to the acknowledgements based on new information from our funding sources.

10 Thank you for considering our manuscript for publication.

Best Regards,
Julia Moriarty, Courtney Harris, Christophe Rabouille, Katja Fennel, Marjorie Friedrichs and Kehui Xu

15 **Associate Editor Decision: Publish subject to minor revisions (Editor review)** (08 Mar 2017) by Dr. Jack Middelburg
Comments to the Author:
Dear Dr. Moriarty:

Thank you for submitting this nice and interesting paper to Biogeosciences. I have read the revised paper and your rebuttal
20 and believe that your paper is almost ready for publication.
There are a few minor technical issues to be addressed, see below, before going into print.

- All through, if you use bottom-water or water-column as adjectives, please hyphen.
**1. Response & Changes to the Text:** Thank you for catching this. We have made these corrections throughout the
25 manuscript.

- P.5, title section 2.2. add space
**2. Response & Changes to the Text:** Thank you for catching this. We have made this correction.

30 - P.6, I presume it is Sherwood et al. and not Sherwood. Moreover, this paper has not yet been submitted.
**3. Response & Changes to the Text:** Thank you for catching this. We have changed the citation in the text to Sherwood et al. (in prep). Based on the "Manuscript Preparation" portion of Biogeosciences website, we believe we can cite the in-prep paper, but if the editor prefers, we can change this to Sherwood (pers. comm.). We did email Sherwood this week to confirm that he plans to submit the paper soon.

- P. 21, l. 20: I presume it is Huettel and not Huettle.
- Same on p. 25
**4. Response & Changes to the Text:** Thank you for catching this. We have made these corrections.

40 **- P. 31: Sherwood et al. status?**
**5. Response & Changes to the Text:** See our third response.

Besides these minor technical issues, there is a potential problem, not identified by the referees, and perhaps I might have misunderstood it. In lines 5-21 of page 13, and related Table 6 and Fig. 6, you report that 60 to 85 % of oxygen consumption
45 is due to nitrification. I have problems with these numbers, it suggests to me that a calculation error (or more likely unit conversion mess up from model code output to publication units) has been made. Why? This is the rationale. If we assume a 1:1 ratio for oxygen consumption to aerobic respiration and a C:N ratio of about 7, one would expect that nitrification consumes about 15% of the oxygen. The only way you can increase this fraction substantially is by supplying additional ammonium from elsewhere, i.e. ammonium without additional oxygen consumption due to carbon oxidation or re-oxidation
50 of other reduced substances. ODU do not solve your problem based on my modelling experience in the past. Indeed I would

expect a somewhat higher fraction following erosion because you can eat into the pore-water ammonium stock, but never these high numbers. Moreover, all data I am aware are consistent with the 10-20% range of oxygen consumption due to nitrification in marine sediments. Please check carefully, because this type of back-of-the-envelope calculation I just made a quite robust and usually right.

**Response:** We agree that attributing 60-70% of oxygen consumption to nitrification is somewhat unusual, but it is consistent with a previous study at this location. Specifically, our results are consistent with Pastor et al. (2011a), who implemented a steady-state version of the Soetaert et al. (1996a,b) model for various locations on the shelf offshore of the Rhone delta. Although Pastor et al. (2011a) estimated that ~10-30% of seabed oxygen consumption was due to nitrification over most of

10    the shelf, they estimated a percentage of ~54% for their Site A (see their Table 4), which is very similar to our study site (Rassmann et al., 2016) and located only a few km away. Because our implementation of HydroBioSed used the same forcing, as well as the same biogeochemical rate constants and parameters, that Pastor et al. (2011a) used for their model of Site A, we expect results from the two models to be similar. The difference in percentages between Pastor et al. (2011a) (54%) and HydroBioSed (60-70%) is likely due to our inclusion of resuspension, and our method of repartitioning organic

15    matter, which increased seabed remineralization rates in HydroBioSed relative to Pastor et al. (2011a).

Nitrification accounts for such a large proportion of seabed oxygen consumption because of this site's high rates of sub-oxic and anoxic remineralization. With a sediment accumulation rate of ~10 cm $y^{-1}$ and an organic matter deposition rate of 657 g Carbon $m^{-2} y^{-1}$, large amounts of organic matter are rapidly transported through the 1-2 millimeter-thick oxic region of the

20    seabed to the underlying anoxic region. Remineralization of organic matter in this anoxic environment produces ammonium (i.e. ~3000 mmol $m^{-3}$; see observed seabed profiles in Pastor et al. (2011a)'s Fig. 2) that then diffuses upwards where it can be oxidized. Assuming a diffusion rate of ~$10^{-9}$ $m^2 s^{-1}$ (see Table 3), and a change in ammonium concentration of ~1500 mmol $m^{-3}$ over a vertical distance of ~0.5 cm (Fig. 5), ammonium may diffuse upwards at a rate of ~5,000 mmol $m^{-3} d^{-1}$ in surficial sediments. Nitrification of this ammonium would therefore consume ~10,000 mmol $m^{-3} d^{-1}$ of $O_2$. This calculation is

25    consistent with nitrification rates presented in Fig. 6 of ~20 mmol $O_2 m^{-2} d^{-1}$, which is equivalent to ~10,000 mmol $O_2 m^{-3} d^{-1}$ because nitrification in the model occurs primarily in the top 2 mm, i.e. the oxic region, of the seabed. In contrast, "back-of-the-envelope" estimates imply that aerobic respiration consumes ~5,000 mmol $O_2 m^{-3} d^{-1}$. This number was estimated by assuming organic matter concentrations of 2 DW% (~=3.5 $10^5$ mmol C $m^{-3}$) and a remineralization rate of 5.5 $y^{-1}$ (i.e., the average of 0.31 and 11 $y^{-1}$). Overall, these calculations imply that nitrification accounts for ~2/3 of seabed oxygen

30    consumption, whereas ~1/3 is from aerobic respiration, consistent with Table 6, Fig. 6, and the text on pg. 15, lines 5-21 (pg. 13 in the previously submitted manuscript).

**Changes to the Text:** To provide more context for the reader, we have added text to the Study Site section, i.e. Section 2.1, pg. 6, lines 27-31, so that it includes Pastor et al. (2011a)'s result that large percentages of organic matter are respired

35    anaerobically and that nitrification accounts for an unusually large amount of seabed oxygen consumption on the Rhone prodelta. We also added text to the Discussion (Section 4.3, pg. 22, line 3) indicating that our result that both nitrification and aerobic respiration were large components of seabed oxygen consumption is consistent with Pastor et al. (2011a).

Thank you for submitting this paper to Biogeosciences,

Jack Middelburg, Associate Editor

[revised manuscript text omitted]

Julia Moriarty 3/8/2017 5:31 PM

---

## Author Response (AR3)

Dear Dr. Jack Middelburg, Associate Editor,

Thank you for handling our paper, BG-2016-482. Recently, our co-author Dr. Christophe Rabouille requested that he be listed as the last author, as data from his studies was instrumental for the project. My PhD advisor, Dr. Courtney Harris,
5  wishes to remain second author. For that reason, if possible, we would like to modify the order of the authorship to be:

Moriarty, Harris, Fennel, Friedrichs, Xu, and Rabouille.

Is it possible to make the change at this time, and if so, is this the appropriate way to go about it?
10
Thank you,
Julia Moriarty

[revised manuscript text omitted]

**(a) Bed Stress (Pa)**

**(b) Seabed O$_2$ Consumption (mmol O$_2$ m$^{-2}$ d$^{-1}$)**

**(c) BW O$_2$ Consumption (mmol O$_2$ m$^{-2}$ d$^{-1}$)**

**(d) Cumulative Seabed O$_2$ Consumption (mol O$_2$ m$^{-2}$ d$^{-1}$)**

**(e) Cumulative BW O$_2$ Consumption (mol O$_2$ m$^{-2}$ d$^{-1}$)**

**Figure 4: Time series of bed stress and oxygen consumption in the seabed and bottom water (BW) for both the standard (blue solid line) and no-resuspension model runs (pink line). Shading indicates resuspension events, i.e. cycles of erosion and re-deposition, as listed in Fig. 3. The red dashed line indicates the critical shear stress for erosion, and the black dashed lines indicate the times at which profiles in Fig. 5 were estimated.**

[Figure]

**Figure 5: Seabed profiles of oxygen (top row; mmol $O_2$ m$^{-3}$), nitrate (second row; mmol N m$^{-3}$), ammonium (third row; mmol N m$^{-3}$), and degradable particulate organic carbon (POC; bottom row; dry weight (%)) from the standard model run for times immediately preceding the mid-April resuspension event (6 April 2012, left column), during the erosional period (10 April 2012, center column), and during the depositional period (13 April 2012, right column). Fig. 4 shows the times at which the profiles were estimated. Tickmarks on the blue lines indicate the location of each seabed layer. The black dashed lines indicate the seabed water interface, and all seabed depths are given relative to this interface. The 'X's indicate near-bed values for the water column.**

[Figure]

**Figure 6: Physical (top) and biogeochemical (bottom) sources and sinks of oxygen within the seabed for the standard model run. Sources and sinks of oxygen to the seabed are positive and negative, respectively. Small biogeochemical sinks <1 mmol O$_2$ m$^{-2}$ d$^{-1}$ (ODU oxidation and remineralization of refractory POM) are not shown. Shading indicates resuspension events, i.e. cycles of erosion & deposition, including 6–13 April, 23 April–3 May, and 18–25 May, 2012.**

[Figure]

**Figure 7: Rate of oxygen consumption in the (a) seabed and (c) bottom waters for each sensitivity test listed in Table 4. Fraction of (b) seabed and (d) bottom-water oxygen consumption induced by resuspension, calculated by dividing the difference between each sensitivity test and its no-resuspension model run by the value from the sensitivity test. In both panels, bars represent averages over two months. Dots indicate the maximum values during this two-month period (which occurred during resuspension events). The dashed lines represent values from the standard model run, with the color of the line consistent with the type of data it represents (i.e. two-month average or maximum value).**

[Figure]

Figure 8: **Box and whisker plot indicating the $0^{th}$, $25^{th}$, $50^{th}$, $75^{th}$, and $100^{th}$ percentiles of combined seabed-bottom-water (BW) oxygen consumption averaged over different timescales for the standard model run. The pink lines indicate estimates from the no-resuspension model run.**